# Physics-Informed Interpolator Generalizes Well in Fixed Dimension: Inductive Bias and Benign Overfitting

## Abstract

Recent advances in machine learning have inspired a surge of research into reconstructing specific quantities of interest from measurements that comply with certain physical laws. These efforts focus on inverse problems that are governed by partial differential equations (PDEs). In this work, we develop an asymptotic Sobolev norm learning curve for kernel ridge(less) regression when addressing (elliptical) linear inverse problems. Our results show that the PDE operators in the inverse problem can stabilize the variance and even behave benign overfitting for fixed-dimensional problems, exhibiting different behaviors from regression problems. Besides, our investigation also demonstrates the impact of various inductive biases introduced by minimizing different Sobolev norms as a form of implicit regularization. For the regularized least squares estimator, we find that all considered inductive biases can achieve the optimal convergence rate, provided the regularization parameter is appropriately chosen. The convergence rate is actually independent to the choice of (smooth enough) inductive bias for both ridge and ridgeless regression. Surprisingly, our smoothness requirement recovers the condition found in Bayesian setting and extends the conclusion to the minimum norm interpolation estimators.

## 1 Introduction

Inverse problems are widespread across science, medicine, and engineering, with research in this field yielding significant real-world impacts in medical image reconstruction (Ronneberger et al., 2015), inverse scattering (Khoo et al., 2017) and 3D reconstruction (Sitzmann et al., 2020). One typical way to solve (elliptical) inverse problems is conducted by statistical machine learning methods (Kaipio & Somersalo, 2006; Knapik et al., 2011; Lu et al., 2022). To be specific, we consider the problem of reconstructing a function $f^*$ from random sampled observations $D = \{(x_i, y_i)\}_{i=1}^n$ from an unknown distribution $P$ on $\mathcal{X} \times \mathcal{Y}$, where $y_i$ is the noisy measurement of $f^*$ through a measurement procedure $\mathcal{A}$, $i.e.$ $\mathbb{E}[y|X = x] = (\mathcal{A}f)(x)$. For simplicity, we assume $\mathcal{A}$ is self-adjoint (elliptic) linear operator in this paper (Knapik et al., 2011; de Hoop et al., 2021; Lu et al., 2022). When the observations are the direct observations of the function, the problem is a classical non-parametric function estimation (De Vito et al., 2005; Tsybakov, 2004). Nevertheless, the observations may also come from certain physical laws described by a partial differential equation (PDE) (Stuart, 2010; Benning & Burger, 2018). Since the most challenging linear inverse problems $\mathcal{A}^{-1}$ are ill-posed, where a small noise in the observation can result in much larger errors in the solution. Further analysis (Knapik et al., 2011; Nickl et al., 2020; Lu et al., 2021b; 2022; Nickl, 2023; Randrianarisoa & Szabo, 2023) of how the structure of the ill-posed inverse problem would change the information-theoretical analysis is always needed.

To handle such ill-posed inverse problem, over-parameterized machine learning models (Raissi et al., 2019; Han et al., 2018; Sirignano & Spiliopoulos, 2018) and interpolated estimators (Yang et al., 2021; Chen et al., 2021a) become successful solutions to linear inverse problems and they can generalize well under noisy observation, $i.e.$, benign overfitting (Bartlett et al., 2020a; Frei et al., 2022; Cao et al., 2022; Zhu et al., 2023). Nevertheless, statistical mechanism and inherent properties of these estimators for inverse problems are still unclear in terms of the following question:

*What are the conditions inherent to inverse problems that facilitate or impede benign overfitting?*
*How to achieve it by selecting the appropriate inductive bias?*

To understand this question, we investigate physics-informed kernel methods (Chen et al., 2021a; Yang et al., 2021) as a theoretical model to model the over-parameterization behaviours. We found that the PDE operator in the inverse problem stabilizes the variance, leading to benign overfitting even in fixed-dimension settings. This contrasts with function fitting, where benign overfitting typically occurs only in high-dimensional settings, while fixed-dimension settings tend to exhibit catastrophic/temper overfitting Mallinar et al. (2022); Buchholz (2022); Rakhlin & Zhai (2019a). We also observed that inductive bias needs to focus enough on the low frequency component to achieve best possible convergence rate. To this end, we consider a general class of norm, known as Kernel Sobolev space (KSS) (Steinwart & Christmann, 2008; Fischer & Steinwart, 2020; Lu et al., 2022; Zhang et al., 2023; Li et al., 2024), to quantize inductive bias in a certain space, *i.e.* the amount of support that the estimator is allowed to have on the tail of the spectrum. The KSS is a spectral transformed space with polynomial transformation (Steinwart & Christmann, 2008; Steinwart & Scovel, 2012; Fischer & Steinwart, 2020; Zhai et al., 2024b) which is a spectral characterization of Sobolev spaces (Fischer & Steinwart, 2020; Adams & Fournier, 2003), which is widely used in characterizing the stability of (elliptic) inverse problems. Mathematically, given a non-negative real number $\beta > 0$, the $\beta$-power Sobolev space $\mathcal{H}^\beta$ associated with a kernel $K$ (see Definition 2.1 for details). The parameter $\beta \in [0, 1]$ characterizes how much we are biased towards low frequency functions. Regarding the learned model, we consider both regularized least square and minimum norm interpolation in this paper for solving the abstract inverse problem:

**Regularized Least Square** (Knapik et al., 2011; Nickl et al., 2020; Lu et al., 2022)

$$\hat{f}_\gamma := \arg\min_f \gamma_n \|f\|_{\mathcal{H}^\beta} + \frac{1}{n}\sum_{i=1}^n \|\mathcal{A}f(x_i) - y_i\|^2 \qquad (1)$$

**Minimum Norm Interpolation** (Wang & Wang, 2018; Yang et al., 2021; Chen et al., 2021a)

$$\hat{f} := \arg\min_f \|f\|_{\mathcal{H}^\beta} \quad \text{s.t.} \mathcal{A}f(x_i) = y_i \qquad (2)$$

Accordingly, we have developed the generalization guarantees of Sobolev norm learning for both (Sobolev norm)-regularized least squares and minimum (Sobolev) norm interpolation in the context of elliptical linear inverse problems. Based on the derived results, we investigate the effects of various inductive biases (*i.e.* $\beta$) that arise when minimizing different Sobolev norms. Minimizing these norms imposes an inductive bias from the machine learning algorithms. In the case of the regularized least squares estimator, we demonstrate that all the smooth enough inductive biases are capable of achieving the optimal convergence rate, assuming the regularization parameter is selected correctly. Additionally, the choice of inductive bias does not influence the convergence rate for interpolators, e.g., the overparameterized/ridgeless estimators. This suggests that with a perfect spectrally transformed kernel, the convergent behavior of regression will not change. The only difference may occur when using empirical data to estimate the kernel, *i.e.* under the semi-supervised learning setting (Zhou & Burges, 2008; Zhai et al., 2024b). The contributions and technical challenges are summarized as below.

### 1.1 Contribution and Technical Challenges

- Instead of considering regularizing RKHS norm (Lu et al., 2022; Randrianarisoa & Szabo, 2023) or interpolation while minimizing RKHS norm (Barzilai & Shamir, 2023; Cheng et al., 2024), we consider (implicit) regularization using a Kernel Sobolev norm (Fischer & Steinwart, 2020) or spectrally transformed kernel (Zhai et al., 2024b). Under such setting, we aim to study how different inductive bias will change the statistical properties of estimators. To this end, we derived the closed form solution for spectrally transformed kernel (Zhai et al., 2024b) estimators for linear inverse problem via a generalized Represener theorem for inverse problem (Unser, 2021) and extend previous non-asymptotic benign overfitting bounds (Bartlett et al., 2020a; Cheng et al., 2024; Barzilai & Shamir, 2023) to operator and inverse problem setting.

- Our non-asymptotic bound can cover both regularized and minimum norm interpolation kernel estimators for solving (linear) inverse problems. For the regularized case, we recovered the minimax optimal rate for linear inverse problem presented in (Lu et al., 2022). We provide the first rigorous upper bound for the excess risk of the min-norm kernel interpolator in the fixed dimensional setting from benign overfitting to tempered overfitting, and catastrophic overfitting

in Physics-informed machine learning. ***Our results show that the PDE operators in inverse problems possess the capability to stabilize variance and remarkably behave benign overfitting, even for problems with a fixed number of dimensions, a trait that distinguishes them from regression problems.***

- Our target is to examine the effects of various inductive biases that arise from minimizing different Sobolev norms, which serve as a form of inductive bias imposed by the machine learning algorithms. For regularized regression in fixed dimension, traditional research (Fischer & Steinwart, 2020; Lu et al., 2022; Guastavino & Benvenuto, 2020) show that proper regularized least square regression can achieve minimax optimal excess risk with ***smooth enough*** implicit regularization of arbitrary spectral decay. Our bound concrete the similar phenomenon happens in the overparamterized / interpolating kernel estimators where ***the choice of smooth enough inductive bias also does not affect convergence speed***. The smoothness requirement of implicit bias $\beta$ should satisfies $\lambda\beta \geq \frac{\lambda r}{2} - p$, where $r$ is the smoothness of the target function (characterized by the source condition), $\lambda$ is the spectral decay of the kernel operator and $p$ is the order of the elliptical inverse problem, see Table 1 for details. Under the function estimation setting, the selection matches the empirical understanding in semi-supervised learning (Zhou & Burges, 2008; Zhou & Belkin, 2011; Smola & Kondor, 2003; Chapelle et al., 2002; Dong et al., 2020; Zhai et al., 2024b) and ***theoretically surprisingly matches the smoothness threshold determined for the Bayesian Inverse problems*** (Knapik et al., 2011; Szabó et al., 2013).

## 1.2 RELATED WORK

**Physics-informed Machine Learning:** Partial differential equations (PDEs) are widely used in many disciplines of science and engineering and play a prominent role in modeling and forecasting the dynamics of multiphysics and multiscale systems. The recent machine learning revolution transforming the computational sciences by enabling flexible, universal approximations for high-dimensional functions and functionals. This inspires researcher to tackle traditionally intractable high-dimensional partial differential equations via machine learning methods (Long et al., 2018; 2019; Raissi et al., 2019; Han et al., 2018; Sirignano & Spiliopoulos, 2018; Khoo et al., 2017; Liu et al., 2020). Theoretical convergence results for deep learning based PDE solvers has also received considerable attention recently. Specifically, Lu et al. (2021a); Grohs & Herrmann (2020); Marwah et al. (2021); Wojtowytsch et al. (2020); Xu (2020); Shin et al. (2020); Bai et al. (2021) investigated the regularity of PDEs approximated by a neural network and Lu et al. (2021a); Luo & Yang (2020); Duan et al. (2021); Jiao et al. (2021a;b); Jin et al. (2022); Doumèche et al. (2024) further provided generalization analyses. Nickl et al. (2020); Lu et al. (2021b); Hütter & Rigollet (2019); Manole et al. (2021); Huang et al. (2021); Wang et al. (2023) provided information theoretical optimal lower and upper bounds for solving PDEs from random samples. However, previous analyses have concentrated on under-parameterized models, which do not accurately characterize large neural networks (Raissi et al., 2019; E & Yu, 2018) and interpolating estimators (Yang et al., 2021; Chen et al., 2021a). Our analysis addresses this gap in theoretical research and provide the first unified upper bound from regularized least square estimators to benign overfitting minimum norm interpolators under fixed dimensions. It is important to point out that concurrent work by Haas et al. (2024) also constructed a kernel interpolator exhibiting benign overfitting in a fixed dimension, using a spiked kernel. In our work, we do not modify the kernel but demonstrate benign overfitting through physics-informed learning.

**Learning with kernel:** Supervised least square regression in RKHS has a long history and its generalization ability and mini-max optimality has been thoroughly studied (Caponnetto & De Vito, 2007; Smale & Zhou, 2007; De Vito et al., 2005; Rosasco et al., 2010; Mendelson & Neeman, 2010). The convergence of least square regression in Sobolev norm has been discussed recently in (Fischer & Steinwart, 2020; Liu & Li, 2020; Zhang et al., 2023). Recently, training neural networks with stochastic gradient descent in certain regimes has been found to be equivalent to kernel regression (Daniely, 2017; Lee et al., 2017; Jacot et al., 2018). Recently Lu et al. (2022); Randrianarisoa & Szabo (2023); Doumèche et al. (2024); Randrianarisoa & Szabo (2023) use kernel based analysis to theoretically understand physics-informed machine learning. Our work is different from this line of researches in two perspective. Firstly, we consider the family of spectrally transformed kernels (Zhai et al., 2024b) to study how different inductive bias on smoothness would affect the efficiency of machine learning estimators. Secondly, We aim to analyze the statistical behavior of kernel interpolators, e.g., overparameterized estimators. Thus we build the first rigorous upper bound for the

| Param. | $\lambda > 1$ | $r \in (0, 1]$ | $p < 0$ | $\mathcal{H}_\beta$ | $\mathcal{H}_{\beta'}$ |
|--------|---------------|----------------|---------|---------------------|------------------------|
| | Eigendecay of Kernel Matrix *(Capacity Condition)* | Smoothness of the ground truth solution *(Source Condition)* | Order of the Inverse Problem *(Capacity Condition on $\mathcal{A}$)* | norm used for regularization $\beta \in [0, 1]$ | norm used for evaluation $\beta' \in [0, \beta]$ |

Table 1: The parameters $\lambda$, $r$, $p$, $\mathcal{H}_\beta$ and $\mathcal{H}_{\beta'}$ are used to describe our problem. The blue-shaded blocks, $\lambda, r, p$ and $\beta'$, represent the parameters that are employed to characterize the inverse problem task, which should influence the minimax optimal risk.

excess risk of the min-norm interpolator in the fixed dimensional setting from benign overfitting to tempered overfitting in physics-informed machine learning.

## 2 PRELIMINARIES, NOTATIONS, AND ASSUMPTIONS

In this section, we introduce the necessary notations and preliminaries for Reproducing kernel Hilbert space (RKHS), including Mercer's decomposition, the integral operator techniques (Smale & Zhou, 2007; De Vito et al., 2005; Caponnetto & De Vito, 2007; Fischer & Steinwart, 2020; Rosasco et al., 2010) and the relationship between RKHS and the Sobolev space (Adams & Fournier, 2003). The required assumptions are also introduced in this section.

We consider a Hilbert space $\mathcal{H}$ with inner product $\langle \cdot, \cdot \rangle_{\mathcal{H}}$ is a separable Hilbert space of functions $\mathcal{H} \subset \mathbb{R}^{\mathcal{X}}$. We call this space a Reproducing Kernel Hilbert space if $f(x) = \langle f, K_x \rangle_{\mathcal{H}}$ for all $K_x \in \mathcal{H} : t \to K(x, t), x \in \mathcal{X}$. Now we consider a distribution $\rho$ on $\mathcal{X} \times \mathcal{Y}(\mathcal{Y} \subset \mathbb{R})$ and denote $\rho_X$ as the marginal distribution of $\rho$ on $\mathcal{X}$. We further assume $\mathbb{E}[K(x, x)] < \infty$ and $\mathbb{E}[Y^2] < \infty$. We define $g \otimes h = gh^\top$ is an operator from $\mathcal{H}$ to $\mathcal{H}$ defined as $g \otimes h : f \to \langle f, h \rangle_{\mathcal{H}} g$. The integral operator technique (Smale & Zhou, 2007; Caponnetto & De Vito, 2007) consider the covariance operator on the Hilbert space $\mathcal{H}$ defined as $\Sigma = \mathbb{E}_{\rho_{\mathcal{X}}} K_x \otimes K_x$. Then for all $f \in \mathcal{H}$, using the reproducing property, we know that $(\Sigma f)(z) = \langle K_z, \Sigma f \rangle_{\mathcal{H}} = \mathbb{E}[f(X)K(X, z)] = \mathbb{E}[f(X)K_z(X)]$. If we consider the mapping $S : \mathcal{H} \to L_2(\rho_{\mathcal{X}})$ defined as a parameterization of a vast class of functions in $\mathbb{R}^{\mathcal{X}}$ via $\mathcal{H}$ through the mapping $(Sg)(x) = \langle g, K_x \rangle$ Its adjoint operator $S^*$ then can be defined as $S^* : \mathcal{L}_2 \to \mathcal{H} : g \to \int_{\mathcal{X}} g(x)K_x \rho_X(dx)$. We further define the empirical sampling operator $\hat{S}_n : \mathcal{H} \to \mathbb{R}^n$ as $\hat{S}_n f := (\langle f, K_{x_1} \rangle, \cdots, \langle f, K_{x_n} \rangle)$ and $\hat{S}_n^* : \mathbb{R}^n \to \mathcal{H}$ as $\hat{S}_n^* \theta = \sum_{i=1}^n \theta_i K_{x_i}$, then we know $\hat{S}_n \hat{S}_n^* : \mathbb{R}^n \to \mathbb{R}^n$ is the Kernel Matrix we denote it as $\hat{K}$, and $\frac{1}{n}\hat{S}_n^* \hat{S}_n : \mathcal{H} \to \mathcal{H}$ is the empirical covariance operator $\hat{\Sigma}$.

Next we consider the eigen-decomposition of the integral operator $\mathcal{L}$ to construct the feature map mapping via Mercer's Theorem. There exists an orthogonal basis $\{\psi_i\}$ of $\mathcal{L}_2(\rho_{\mathcal{X}})$ consisting of eigenfunctions of kernel integral operator $\mathcal{L}$. The kernel function have the following representation $K(s, t) = \sum_{i=1}^\infty \lambda_i \psi_i(s) \psi_i(t)$. where $\psi_i$ are orthogonal basis of $\mathcal{L}_2(\rho_{\mathcal{X}})$. Then $\psi_i$ is also the eigenvector of the covariance operator $\Sigma$ with eigenvalue $\lambda_i > 0$, *i.e.* $\Sigma \psi_i = \lambda_i \psi_i$.

Following the (Bartlett et al., 2020a; Cheng et al., 2024; Barzilai & Shamir, 2023; Tsigler & Bartlett, 2023), we conduct the theoretical analysis using spectral decomposition. Thus, in this paper, we define the spectral feature map $\phi : \mathcal{H} \to \mathbb{R}^\infty$ via $\phi f := (\langle f, \phi_i \rangle_{\mathcal{H}})_{i=1}^\infty$ where $\phi_i = \sqrt{\lambda_i}\psi_i$ which forms an orthogonal basis of the reproducing Kernel Hilbert space. Then $\phi^* : \mathbb{R}^\infty \to \mathcal{H}$ takes $\theta$ to $\sum_{i=1}^\infty \theta_i \phi_i$. Then $\phi^* \phi = id : \mathcal{H} \to \mathcal{H}$, $\phi\phi^* = id : \ell_2^\infty \to \ell_2^\infty$. $\phi$ is an isometry i.e. for any function $f$ in $\mathcal{H}$ we have $\|f\|_{\mathcal{H}}^2 = \|\phi f\|_{\ell_2^\infty}^2$ and $\ell_2^\infty$ denotes the space of sequences of real numbers $\{x_i\}_{i=1}^\infty$ such that the $\ell_2$ norm $\|\mathbf{x}\|_{\ell_2^\infty} = \sqrt{\sum_{i=1}^\infty x_i^2}$ is bounded. Similarly we also define $\psi : \mathcal{H} \to \ell_2^\infty$ via $\psi f := (\langle f, \psi_i \rangle_{\mathcal{H}})_{i=1}^\infty$, the motivation of defining this is this can simplify our computation in the lemmas, we define $\psi^* : \mathbb{R}^\infty \to \mathcal{H}$ takes $\theta$ to $\sum_{i=1}^\infty \theta_i \psi_i$. We then define the operator $\Lambda_{\mathcal{X}} : \mathbb{R}^\infty \to \mathbb{R}^\infty$ corresponding to $\mathcal{X}$ is the operator such that $\mathcal{X} = \phi^* \Lambda_{\mathcal{X}} \phi$, which implies $\Lambda_{\mathcal{X}\mathcal{Y}} = \Lambda_{\mathcal{X}} \Lambda_{\mathcal{Y}}$. Followed by our notation, we can simplify the relationship between $\phi$ and $\psi$ as $\phi = \Lambda_\Sigma^{1/2}\psi$ and $\phi^* = \psi^* \Lambda_\Sigma^{1/2}$.

**Definition 2.1** (Sobolev Norm (Steinwart & Christmann, 2008; Steinwart & Scovel, 2012; Pillaud–Vivien et al., 2018; Fischer & Steinwart, 2020; Zhang et al., 2023)). For $\beta > 0$, the $\beta$-power Kernel Sobolev Space (KSS) is

$$\mathcal{H}^\beta := \{\sum_{i \geq 1} a_i \lambda_i^{\beta/2} \psi_i : \sum_{i \geq 1} a_i^2 < \infty\} \subset L^2(\rho_{\mathcal{X}}),$$

equipped with the $\beta$-power norm via $\|\sum_{i \geq 1} a_i \lambda_i^{\beta/2} \psi_i\|_\beta := (\sum_{i \geq 1} a_i^2)^{1/2}$.

*Remark* 1. We follows the definition of Sobolev space in (Steinwart & Christmann, 2008; Pillaud-Vivien et al., 2018; Fischer & Steinwart, 2020; Zhang et al., 2023) which is introduced to characterize the misspecification in kernel regression (Zhang et al., 2023; Kanagawa et al., 2016; Pillaud-Vivien et al., 2018; Steinwart et al., 2009). The parameter $\beta$ in the source condition controls the amount of support that is allowed to have on the tail of the spectrum. As shown in Steinwart & Scovel (2012); Steinwart & Christmann (2008); Fischer & Steinwart (2020), $\mathcal{H}^\beta$ is an interpolation between Reproducing Kernel Hilbert Space and $\mathcal{L}_2$ space. Formally, $\|\mathcal{L}^{\beta/2} f\|_\beta = \|f\|_{L_2}$ where $\mathcal{L} = SS^*$ and $\|f\|_\beta = \|\Sigma^{\frac{1-\beta}{2}} f\|_{\mathcal{H}}$ for $0 \leq \beta \leq 1$. Thus when $\beta = 1$, the $\mathcal{H}^\beta$ is the same as Reproducing Kernel Hilbert Space and when $\beta = 0$ the $\mathcal{H}^\beta$ is the same as $\mathcal{L}_2$ space. The Hilbert scale of function spaces defined through varying $\beta$ quantizes the inductive bias, serving as an regularity condition.

When we select our kernel to be the Matérn covariance kernel (Chen et al., 2021b), our definition of Sobolev space coincide with the Sobolev space (Adams & Fournier, 2003) on the torus $\mathbb{T}^d = [0,1]^d_{\text{per}}$. The $\beta$-power norm definition of Sobolev space served as Fourier charaterization of Sobolev space (Adams & Fournier, 2003; Wendland, 2004) which is the most natural function space for PDE analysis.

**Assumption 2.2** (Assumptions on Kernel and Target Function). We assume the standard capacity condition on kernel covariance operator with a source condition about the regularity of the target function following Caponnetto & De Vito (2007) and assumption of the inverse problem following Lu et al. (2022). These conditions are stated explicitly below:

- **(a) Assumptions on boundedness.** The kernel feature are bounded almost surely, *i.e.* $|k(x,y)| \leq R$ and the observation $y$ is also bounded by $M$ almost surely.
- **(b) Capacity condition** (Steinwart & Scovel, 2012; Steinwart & Christmann, 2008). Consider the spectral representation of the kernel covariance operator $\Sigma = \sum_i \lambda_i \psi_i \otimes \psi_i$, we assume polynomial decay of eigenvalues of the covariance matrix $\lambda_i \propto i^{-\lambda}$ for some $\lambda > 1$. This assumption satisfies for many useful kernels in the literature such as Minh et al. (2006), neural tangent kernels (Bietti & Bach, 2020; Chen & Xu, 2020).
- **(c) Source condition** (Steinwart & Scovel, 2012; Steinwart & Christmann, 2008; Fischer & Steinwart, 2020). We also impose an assumption on the smoothness of the true function, which characterizes the regularity of the test function. There exists $r \in (0,1]$ such that $f^* = \mathcal{L}^{r/2} \phi$ for some $\phi \in L^2$. If $f^*(x) = \langle \theta_*, K_x \rangle_{\mathcal{H}}$, the source condition can also be written as $\|\Sigma^{\frac{1-r}{2}} \theta_*\|_{\mathcal{H}} < \infty$. The source condition can be understood as the target function lies in the $r$-power Sobolev space.
- **(d) Capacity conditions on** $\mathcal{A}$ (Knapik et al., 2011; Cabannes et al., 2021; de Hoop et al., 2021; Lu et al., 2022). For theoretical simplicity, we assume that the self-adjoint operators $\mathcal{A}$ are diagonalizable in the same orthonormal basis $\phi_i$. Thus we can assume $\mathcal{A} = \sum_{i=1}^\infty p_i \psi_i \otimes \psi_i$, for positive constants $p_i > 0$. We further assume $p_i \propto i^{-p}$. We further assume $p < 0$, for the inverse problem we consider inverse problem arising from PDEs where $\mathcal{A}$ is a differential operator.

*Remark* 2. Although the diagonalizable assumptions is strong, the assumption is usually made for theoretical analysis of kernel-based inverse problem solver Knapik et al. (2011); Cabannes et al. (2021); de Hoop et al. (2021); Lu et al. (2022). The parameter $p$ here is used to characterise the order of PDE. For example, operator $\Delta^k$'s spectrum decays at a different polynomial speed as $k$ varies. The co-diagonalization assumption holds since both the Laplacian operator $\Delta$ and the shift-invariant Kernel covariance operator/inner product kernel with uniform data have the Fourier modes as eigenfunction which is guaranteed by Bochner's theorem.

**Example 2.3** (Schrödinger equation on a Hypercube). Consider solving Schrödinger equation on a hypercube $-\Delta u + u = f$ on $\mathbb{T}^d = [0,1]^d_{\text{per}}$, where $\Delta$ is the Laplacian operator. To solve the Schrödinger equation, one observe collocation points $x_i$ uniformly sampled from $\mathbb{T}^d$ with associated function values $y_i = f(x_i) + \varepsilon_i$ $(1 \leq i \leq n)$ where $\varepsilon_i$ is a mean-zero i.i.d observational noise.

**Decomposition of Signals** Following Bartlett et al. (2020b); Tsigler & Bartlett (2023); Cheng et al. (2024), we decompose the risk estimation to the "low dimension" part which concentrates well and "higher dimension" part which performs as regularization. We define the decomposition operations in this paragraph. We first additionally define $\phi_{\leq k} : f \mapsto (\langle f, \phi_i \rangle_{\mathcal{H}})_{i=1}^k$ which maps $\mathcal{H}$ to it's "low dimensional" features in $\mathbb{R}^k$, it intuitively means casting $f \in \mathcal{H}$ to its top $k$ features, similarly we can define $\phi_{>k} : f \mapsto (\langle f, \phi_i \rangle_{\mathcal{H}})_{i=k+1}^\infty$. We also define $\phi^*_{\leq k}$ takes $\theta \in \mathbb{R}^k$ to $\sum_{i=1}^k \theta_i \phi_i$,

similarly we can define $\phi^*_{>k}$ takes $\theta \in \ell_2^\infty$ to $\sum_{i=k+1}^\infty \theta_{i-k}\phi_i$. For function $f \in \mathcal{H}$, we also define $f_{\leq k} := \phi^*_{\leq k}\phi_{\leq k}f = \sum_{i=1}^k \langle f, \phi_i \rangle_\mathcal{H} \phi_i$ which intuitively means only preserving the top $k$ features, for operator $\mathcal{A} : \mathcal{H} \to \mathcal{H}$, we also define $\mathcal{A}_{\leq k} : f \mapsto (\mathcal{A}f)_{\leq k}$. Similarly we could define $f_{>k}$ and $\mathcal{A}_{>k}$. We could show the decomposition $f = f_{\leq k} + f_{>k}$ and $\mathcal{A} = \mathcal{A}_{\leq k} + \mathcal{A}_{>k}$ holds for both signal and operators which is formally proved in Lemma A.1 in the appendix.

We use $\|\cdot\|$ to denote standard $l^2$ norm for vectors, and operator norm for operators. We also use standard big-O notation $O(\cdot), o(\cdot), \Omega(\cdot), \tilde{O}(\cdot)$ (ignore logarithimic terms).

# 3 MAIN THEOREM: EXCESS RISK OF KERNEL ESTIMATOR FOR INVERSE PROBLEM

Using the notations in Section 2, we can reformulate the data generating process as $y = \hat{S}_n \mathcal{A}f^* + \varepsilon$, where $y \in \mathbb{R}^n$ is the label we observed on the $n$ data points $\{x_i\}_{i=1}^n$, $f^*$ is the ground truth function and $\varepsilon \in \mathcal{N}(0, \sigma_\varepsilon^2 I_{n \times n})$ is the Gaussian noise. We first provide closed form solutions to ridge regression via the recently developed generalized representer theorem for inverse problem (Unser, 2021).

---

**Lemma 3.1.** *The least square problem regularized by Kernel Sobolev Norm*

$$\hat{f}_\gamma := \underset{f \in \mathcal{H}^\beta}{\arg\min} \frac{1}{n}\|\hat{S}_n \mathcal{A}f - y\|^2 + \gamma_n \|f\|_{\mathcal{H}^\beta}^2. \tag{3}$$

*has the finite-dimensional representable closed form solution $\hat{f} = \mathcal{A}\Sigma^{\beta-1}\hat{S}_n^* \hat{\theta}_n$ where*

$$\hat{\theta}_n := \underbrace{(\hat{S}_n \mathcal{A}^2 \Sigma^{\beta-1} \hat{S}_n^* + n\gamma_n I)^{-1}}_{\tilde{K}^\gamma} y \in \mathbb{R}^n .$$

---

For the simplicity of presentation, We denote the empirical spectrally transformed kernel $\hat{S}_n \mathcal{A}^2 \Sigma^{\beta-1} \hat{S}_n^*$ as $\tilde{K}$, and the regularized version $\hat{S}_n \mathcal{A}^2 \Sigma^{\beta-1} \hat{S}_n^* + n\gamma_n I$ as $\tilde{K}^\gamma$, and we denote the spectrally transformed covariance operator $\tilde{\Sigma}$ as $\mathcal{A}^2 \Sigma^\beta$.

## 3.1 CONCENTRATION COEFFICIENTS

We expect that $\tilde{K}_{>k} \approx \tilde{\gamma} I$ which serves as a self-regularization term, inspired by Barzilai & Shamir (2023) we quantify this by introducing the concentration coefficient for spectrally transformed kernel $\tilde{K}$ to measure the self-regularization effect of $\tilde{K}_{>k}$.

**Definition 3.2** (Concentration Coefficient $\rho_{n,k}$). We quantify this by what we call the concentration coefficient

$$\rho_{k,n} := \frac{\|\tilde{\Sigma}_{>k}\| + \mu_1(\frac{1}{n}\tilde{K}_{>k}) + \gamma_n}{\mu_n(\frac{1}{n}\tilde{K}_{>k}) + \gamma_n}, \qquad \text{where} \quad \tilde{\Sigma} = \mathcal{A}^2 \Sigma^\beta.$$

Assumptions on feature map is essential to obtain various concentration inequalities, typically sub-Gaussian assumptions on feature map is needed to obtain concentration results. However, this does not hold for many common kernels. Following recent work Barzilai & Shamir (2023), we only require mild condition on features i.e. $\alpha_k, \beta_k = \Theta(1)$ which is applicable in many common kernels (weakest assumption in the literature as far as the authors know), without imposing sub-Gaussian assumptions, but our bound in the interpolation case can be tighter with the sub-Gaussian assumption in Theorem 4.2, where in that case $\rho_{k,n} = \Theta(1)$.

**Assumption 3.3** (Well-behaved features). Given $k \in \mathbb{N}$, we define $\alpha_k, \beta_k$ as follows.

$$\alpha_k := \inf_x \min \left\{ \frac{\sum_{i>k} p_i^a \lambda_i^b \psi_i(x)^2}{\sum_{i>k} p_i^a \lambda_i^b} : \text{finite choices of } a, b \right\},$$

$$\beta_k := \sup_x \max \left\{ \frac{\sum_{i=1}^k \psi_i(x)^2}{k}, \frac{\sum_{i>k} p_i^a \lambda_i^b \psi_i(x)^2}{\sum_{i>k} p_i^a \lambda_i^b} : \text{finite choices of } a, b \right\},$$

$(a, b)$ is picked in our proof of Lemma B.3 in the Appendix. Since $\inf \leq \mathbb{E} \leq \sup$, one always has $0 \leq \alpha_k \leq 1 \leq \beta_k$. We assume that $\alpha_k, \beta_k = \Theta(1)$.

*Remark* 3. For each term in these definitions, the denominator is the expected value of the numerator, so $\alpha_k$ and $\beta_k$ quantify how much the features behave as they are "supposed to". Note that $\alpha_k$ and $\beta_k$ are $\Theta(1)$ in many common kernels. We here give several examples (Barzilai & Shamir (2023))that satisfies the assumptions, includes

- *Kernels With Bounded Eigenfunctions* If $\psi_i^2(x) < M$ uniformly holds for $\forall i, x$ then Assumption 3.4 trivially holds that $\beta_k \leq M$ for any $k \in \mathbb{N}$. Analogously, if $\psi_i^2 \geq M'$ then $\alpha_k \geq M'$. This may be weakened to the the training set such that only a high probability lower bound is needed. Kernels satisfies this assumption includes RBF and shift-invariant kernels (Steinwart et al., 2006, Theorem 3.7) and Kernels on the Hypercube $\{0, 1\}^d$ of form $h\left(\frac{\langle x, x' \rangle}{\|x\| \|x'\|}, \frac{\|x\|^2}{d}, \frac{\|x'\|^2}{d}\right)$ Yang & Salman (2019).

- *Dot-Product Kernels on $\mathcal{S}^d$* Follows the computation in (Barzilai & Shamir, 2023, Appendix G), one can know dot-product Kernels on $\mathcal{S}^d$ satisfies Assumption 3.4. This examples includes Neural Tangent kernel (Jacot et al., 2018) on sphere.

Similar to Barzilai & Shamir (2023), we require regularity condition on $\beta_k$ to overcome technical difficulty in extending to infinite dimension in Lemma C.5:

**Assumption 3.4** (Regularity assumption on $\beta_k$)**.** There exists some sequence of natural numbers $(k_i)_{i=1}^{\infty} \subset \mathbb{N}$ with $k_i \xrightarrow[i \to \infty]{} \infty$ s.t. $\beta_{k_i} \operatorname{tr}(\tilde{\Sigma}_{>k_i}) \xrightarrow[i \to \infty]{} 0$.

We can know $\tilde{\Sigma}_{>k_i}$ is still transformed trace class, so one always has $\operatorname{tr}(\tilde{\Sigma}_{>k_i}) \xrightarrow[i \to \infty]{} 0$. As such, Assumption 3.4 simply states that for infinitely many choices of $k \in \mathbb{N}$, $\beta_k$ does not increase too quickly. This is of course satisfied by the previous examples of kernels with $\beta_k = \Theta(1)$.

### 3.2 Excess Risk and Eigenspectrum of spectrally transformed kernel $\tilde{K}$

We evaluate excess risk in a certain Sobolev space $\mathcal{H}^{\beta'}$ with $\beta' \in [0, \beta]$. The selection of $\beta'$ is independent of certain learning algorithms on source and capacity conditions, but depends on the downstream applications of learned inverse problem solution. We denote $\hat{f} := \mathcal{A}\Sigma^{\beta-1}\hat{S}_n^*(\hat{S}_n \mathcal{A}^2 \Sigma^{\beta-1}\hat{S}_n^* + n\gamma I)^{-1}y$ as $\hat{f}(y)$ to highlight its dependence on $y \in \mathbb{R}^n$. Recall the data generation process, $y = \hat{S}_n \mathcal{A}f^* + \varepsilon$, we consider $\hat{S}_n \mathcal{A}f^*$ and $\varepsilon$ in bias and variance separately. The excess risk $R(\hat{f}(y)) := \|\hat{f} - f^*\|_{\mathcal{H}^{\beta'}}^2$ has the following bias-variance decomposition.

$$\|\hat{f} - P_{\mathcal{H}^{\beta'}}f^*\|_{H^{\beta'}}^2 = \underbrace{\|\hat{f}(\hat{S}_n \mathcal{A}f^*) - f^*\|_{\mathcal{H}^{\beta'}}^2}_{\text{bias: } B} + \underbrace{\mathbb{E}_{\varepsilon}[\|\hat{f}(\varepsilon)\|_{\mathcal{H}^{\beta'}}^2]}_{\text{variance:} V}. \tag{4}$$

Following benign overfitting literature (Barzilai & Shamir, 2023; Bartlett et al., 2020b; Cheng et al., 2024), we perform the analysis on "low dimensional" ($\leq k$) and "high dimensional" ($> k$) components respectively. Therefore, we define $\tilde{K}_{\leq k}$ as $\hat{S}_n \mathcal{A}_{\leq k}^2 \Sigma_{\leq k}^{\beta-1}\hat{S}_n^*$, and $\tilde{K}_{\leq k}^{\gamma}$ as $\tilde{K}_{\leq k} + n\gamma_n I$, similarly we can define $\tilde{K}_{>k}$ and $\tilde{K}_{>k}^{\gamma}$ respectively. We can also have $\tilde{K} = \tilde{K}_{\leq k} + \tilde{K}_{>k}$ (proved in Appendix A.1). To bound the excess risk of minimum norm interpolation kernel estimator, we need to show the "high dimensional" part of the Kernel matrix $\tilde{K}_{>k}$ can behave as a self-regularization. To show this, we present here the concentration bounds of eigenvalues with proof given in Appendix C.1.

**Theorem 3.5** (Eigenspectrum of spectrally transformed kernel $\tilde{K}$)**.** *Suppose Assumption 3.4 holds, and eigenvalues of $\tilde{\Sigma}$ are given in non-increasing order (i.e. $2p + \beta\lambda > 0$). There exists absolute constant $c, C, c_1, c_2 > 0$ s.t. for any $k \leq k' \in [n]$ and $\delta > 0$, it holds w.p. at least $1 - \delta - 4\frac{r_k}{k^4}\exp(-\frac{c}{\beta_k}\frac{n}{r_k}) - 2\exp(-\frac{c}{\beta_k}\max\left(\frac{n}{k}, \log(k)\right))$ that*

$$\mu_k\left(\frac{1}{n}\tilde{K}\right) \leq c_1 \beta_k \left(\left(1 + \frac{k\log(k)}{n}\right)\lambda_k^{\beta}p_k^2 + \log(k+1)\frac{\operatorname{tr}\left(\tilde{\Sigma}_{>k}\right)}{n}\right), \mu_k\left(\frac{1}{n}\tilde{K}\right) \geq c_2 \mathbb{I}_{k,n}\lambda_k^{\beta}p_k^2 + \alpha_k\left(1 - \frac{1}{\delta}\sqrt{\frac{n^2}{\operatorname{tr}(\tilde{\Sigma}_{>k'})^2 / \operatorname{tr}(\tilde{\Sigma}_{>k'}^2)}}\right)\frac{\operatorname{tr}\left(\tilde{\Sigma}_{>k'}\right)}{n},$$

*where $\mu_k$ is the k-th largest eigenvalue of $\frac{1}{n}\tilde{K}$, $\tilde{\Sigma} := \mathcal{A}^2\Sigma^{\beta}$, $r_k := \operatorname{tr}(\tilde{\Sigma}_{>k})/(p_{k+1}^2\lambda_{k+1}^{\beta})$, and*
$$\mathbb{I}_{k,n} = \begin{cases} 1, & \text{if } C\beta_k k \log(k) \leq n \\ 0, & \text{otherwise} \end{cases}.$$

### 3.3 MAIN RESULTS

In this section, we state our main results on the bias and variance of the kernel estimator. The following theorem is the main result for upper bounds of the bias and variance with the proof details given in Appendix D.2 for bounding the variance and Appendix E.3 for bounding the bias.

**Theorem 3.6** (Bound on Variance). *Let $k \in \mathbb{N}$, $\sigma_\varepsilon^2$ is the noise variance and $\rho_{k,n}$ is defined follows Definition 3.2, then w.h.p. the variance can be bounded by*

$$V \leq \sigma_\varepsilon^2 \rho_{k,n}^2 \cdot \Big( \frac{\mathrm{tr}(\hat{S}_n \psi_{\leq k}^* \Lambda_{\mathcal{A}^{-2}\Sigma^{-\beta'}}^{\leq k} \psi_{\leq k} \hat{S}_n^*)}{\mu_k(\psi_{\leq k}\hat{S}_n^* \hat{S}_n \psi_{\leq k}^*)^2} + \overbrace{\frac{\overbrace{\mathrm{tr}(\hat{S}_n \psi_{>k}^* \Lambda_{\mathcal{A}^2\Sigma^{-\beta'+2\beta}}^{>k} \psi_{>k} \hat{S}_n^*)}}{n^2 \|\tilde{\Sigma}_{>k}\|^2} }^{\textit{effective rank}} \Big). \tag{5}$$

*Remark* 4. The variance bound is decomposed into two parts, the $\leq k$ part which characterize the variance of learning the "low dimension" components and $\geq k$ part characterizing the variance of learning "high dimension" components. We implement similar analysis for the bias as follows.

**Theorem 3.7** (Bound on Bias). *Let $k \in \mathbb{N}$, and $\rho_{k,n}$ is defined follows Definition 3.2, then there exists $C_2, c, c' > 0$ s.t. for any $k$ with $c\beta_k k \log(k) \leq n$, every $\delta > 0$, then w.p. at least $1 - \delta - 8\exp(-\frac{c'}{\beta_k^2}\frac{n}{k})$ the bias can be bounded by*

$$B \lesssim \rho_{k,n}^3 \frac{1}{\delta} \Big[ \underbrace{\frac{\|\phi_{>k}\mathcal{A}_{>k}f_{>k}\|_{\Lambda_\Sigma^{>k}}^2}{p_k^2 \lambda_k^{\beta'}} + \|\phi_{>k}f_{>k}^*\|_{\Lambda_{\Sigma^{1-\beta'}}^{>k}}^2}_{\textit{bias on high frequency components, i.e. } >k \textit{ parts}} + \underbrace{\Big( \gamma_n + \frac{\beta_k \mathrm{tr}(\tilde{\Sigma}_{>k})}{n} \Big)^2 \frac{\|\phi_{\leq k}f_{\leq k}^*\|_{\Lambda_{\mathcal{A}^{-2}\Sigma^{1-2\beta}}^{\leq k}}^2}{p_k^2 \lambda_k^{\beta'}}}_{\textit{bias on low frequency components, i.e. } \leq k \textit{ parts}} \Big].$$

(6)

## 4 APPLICATIONS

Our main results can provide bounds for both the regularized (Yang et al., 2021; Lu et al., 2022) and unregularized cases (Chen et al., 2021a) with the same tools. In this section, we present the implication of our results for both regularized regression and minimum norm interpolation kernel estimators.

### 4.1 REGULARIZED REGRESSION

In this section, we demonstrate the implication of our derive bounds for the classical setup where the regularization $\gamma_n$ is relatively large. We consider regularized least square estimator with regularization strength $\gamma_n = \Theta(n^{-\gamma})$. By selecting $k$ as $\lceil n^{\frac{\gamma}{2p+\beta\lambda}} \rceil$ in Theorem 3.6 and Theorem 3.7, we obtain $\rho_{k,n} = \Theta(1)$ and get a bound that matches Lu et al. (2022), which indicates the corectness and tightness of our results.

> **Theorem 4.1** (Bias and Variance for Regularized Regression). *Let the kernel and target function satisfies Assumption 2.2, 3.3 and 3.4, $\gamma_n = \Theta(n^{-\gamma})$, and suppose $2p + \lambda\beta > \gamma > 0, 2p + \lambda r > 0$, and $r > \beta'$, then for any $\delta > 0$, it holds w.p. at least $1 - \delta - O(\frac{1}{n})$ that*
>
> $$V \leq \sigma_\varepsilon^2 O(n^{\max\{\frac{\gamma(1+2p+\lambda\beta')}{2p+\lambda\beta}, 0\}-1}), B \leq \frac{1}{\delta} \cdot O(n^{\frac{\gamma}{2p+\beta\lambda}(\max\{\lambda(\beta'-r), -2p+\lambda(\beta'-2\beta)\})}).$$

*Remark* 5. Once proper regularization norm is selected, *i.e.* $\lambda\beta \geq \frac{\lambda r}{2} - p$, with optimally selected $\gamma = \frac{2p+\lambda\beta}{(2p+\lambda+2r)}$ which balance the variance $n^{\frac{\gamma(1+2p+\lambda\beta')}{2p+\lambda\beta}-1}$ and the bias $n^{\frac{\gamma(\lambda(\beta'-r))}{2p+\beta\lambda}}$, our bound can achieve final bound: $n^{\frac{\lambda(\beta'-r)}{2p+\lambda r+1}}$ matches with the convergence rate build in the literature (Knapik et al., 2011; Lu et al., 2022)

### 4.2 MIN-NORM INTERPOLATION FROM BENIGN OVERFITTING TO TEMPERED OVERFITTING

We now shift our attention to the overparameterized interpolating kernel estimators. Recently, Mallinar et al. (2022) distinguished between three regimes: one where the risk explodes to infinity (called catastrophic overfitting), another where the risk remains bounded (called tempered overfitting),

and a third regime involving consistent estimators whose risk goes to zero (called benign overfitting). These regimes are significantly different. In the tempered overfitting regime, when the noise is small, estimator can still achieve a low risk despite overfitting. This means that the bias goes to zero, and the variance cannot diverge too quickly. Recent work (Rakhlin & Zhai, 2019b; Cui et al., 2021; Barzilai & Shamir, 2023; Cheng et al., 2024) showed that minimum (kernel) norm interpolators are nearly tempered over-fitting. However, as shown in Theorem 4.2, *the PDE operator in the inverse problem can stabilize the variance term and make the min-norm interpolation (kernel) estimators benign over-fitting even in fixed-dimension setting.*

> **Theorem 4.2** (Bias and Variance for Interpolators). *Let the kernel and target function satisfies Assumption 2.2, 3.3 and 3.4, and suppose $2p + \lambda \min\{r, \beta\} > 0$ and $r > \beta'$, then for any $\delta > 0$ it holds w.p. at least $1 - \delta - O(\frac{1}{\log(n)})$ that*
>
> $$V \leq \sigma_\varepsilon^2 \rho_{k,n}^2 \tilde{O}(n^{\max\{2p+\lambda\beta', -1\}}), B \leq \frac{\rho_{k,n}^3}{\delta} \tilde{O}(n^{\max\{\lambda(\beta'-r), -2p+\lambda(\beta'-2\beta)\}}).$$

*Remark* 6. For well-behaved sub-Gaussian features, the concentration coefficients $\rho_{k,n} = \Theta(1)$ Barzilai & Shamir (2023) and in the worst case $\rho_{k,n}$ can become $\tilde{O}(n^{2p+\beta\lambda-1})$ which is shown in the Appendix F.2. Our bound can recover the results in Barzilai & Shamir (2023) by setting $p = 0, \beta = 1, \beta' = 0$ and recover the results in Cui et al. (2021) when $\sigma_\epsilon = 0, \beta' = 0$ and $\rho_{k,n} = 1$.

*Remark* 7. *Since the $p$ considered for PDE inverse problems is a negative number (See Assumption 2.2), our bound showed that the structure of PDE inverse problem made benign over-fitting possible even in the fixed dimesional setting. This result differs the behavior of regression with inverse problem when large over-parameterized model is applied.* The more negative $p$ leads to smaller bound over the variance which indicates Sobolev training is more stable to noise, matches with empirical evidence (Son et al., 2021; Yu et al., 2021; Lu et al., 2022).

### 4.3 IMPLICATION OF OUR RESULTS

**Selection of Inductive Bias:** As demonstrated in Theorem 4.1 and Theorem 4.2, variance is independent of the inductive bias (i.e., $\beta$) and the only dependency is appeared in bounding the bias. At the same time, the upper bound for the bias is a maximum of the orange part and the blue part. The orange part is independent of the inductive bias and only depend on the inverse problem (i.e., $r$ and $\lambda$) and evaluation metric (i.e., $\beta'$), while the blue part is the only part depending on the inductive bias used in the regularization. With properly selected inductive bias $\beta$, one can achieve the best possible convergence rate which only depends on the orange part. When the inductive bias does not focus much on the low frequency eigenfunctions (i.e., $\lambda\beta \geq \frac{\lambda r}{2} - p$), that means, regularized with kernel which is not smooth enough, the rate is dominated by the blue part and is potential sub-optimal. Under the function estimation setting, the selection matches the empirical understanding in semi-supervised learning (Zhou & Burges, 2008; Zhou & Belkin, 2011; Smola & Kondor, 2003; Chapelle et al., 2002; Dong et al., 2020; Zhai et al., 2024b;a) and *theoretically surprisingly matches the smoothness requirement determined in the Bayesian inverse problem literature* (Knapik et al., 2011; Szabó et al., 2013).

**Takeaway to Practitioners:** Our theory demonstrated that to attain optimal performance in physics-informed machine learning, incorporating sufficiently smooth inductive biases is necessary. For PINNs applied to higher-order PDEs, one needs smoother activation functions. This is because the value of $p$ for higher-order PDEs is a negative number with a larger absolute value, thus making the term $\frac{\lambda r}{2} - p$ larger. A larger value of $\frac{\lambda r}{2} - p$ necessitates the use of smoother activation functions Bietti & Bach (2020); Chen & Xu (2020) to ensure the solution satisfies the required smoothness conditions imposed by the higher-order PDE. Another implication of the theory is **the variance stabilization effects** as mentioned before brought about by the PDE operator in the inverse problem. Higher-order PDEs would benefit from more substantial stabilization effects. This motivates the idea that Sobolev training (Son et al., 2021; Yu et al., 2021) may not only aid optimization (Lu et al., 2022) but also contribute to improved generalization error for overparameterized models. However, as previously demonstrated, utilizing a neural network with smoother activations is necessary to leverage these benefits.

## 5 EXPERIMENTS

We conducted additional experiments on neural network to validate our theory as well as theoretical findings beyond kernel methods. To be specific, we consider the Poisson equation

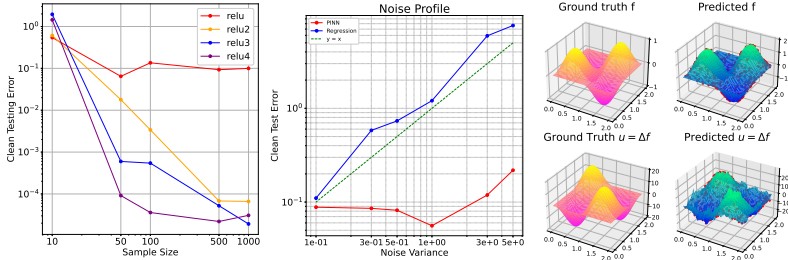

Figure 1: We verified our finding beyond kernel estimators. For all the plotted figure, we learn two dimensional Poisson equation. **(Left)** We examine the impact of smooth inductive bias on convergence. Our findings demonstrate that when the activation function is sufficiently smooth, the inductive bias has a limited effect on improving convergence, which aligns with our theoretical predictions. **(Middle)** Noise profile of Physics-informed interpolator and regression Interpolator. The physics-informed interpolator exhibits benign overfitting, unlike the regression interpolator. **(Right)** Visualization of the ground truth and the learned solutions for $f$ and $u = \Delta f$. The learned solution for $f$ effectively smooths out the high-frequency components in the error of $\Delta f$.

$u = \Delta f$ on $\Omega = [0, 2]^2$ with Dirichlet boundary condition on $\partial \Omega$, where the ground truth $f(x_1, x_2) = \sin(\pi x_1) \sin(\pi x_2)$, where the data points $\{(x_i, y_i)\}_{i=1}^n$ are sampled uniformly from $\Omega$, and $y_i = \Delta f(x_i) + \varepsilon$ with $\varepsilon \sim \mathcal{N}(0, \sigma^2)$. Our experiments are able to illustrate our theory from the following three aspects, and more experimental details can be found on Appendix G.

**Effect of Smoothness of the Inductive Bias** To validate our finding on the necessity of using smoother activation function, we use activation function ReLU, ReLU$^2$, ReLU$^3$, respectively, fix noise level variance $\sigma^2 = 0.1$, and vary number of samples as 50, 100, 500, 1000 and plot the test error against number of samples. The result in Figure 1(Left) verifies our finding that when the inductive bias is not smooth enough, the convergence will benefit from smoother activation function. However, by comparing convergence rate of ReLU$^3$ and ReLU$^4$ in Figure 1(Left), when the activation function is smooth enough, the convergence behavior would not be affected too much. This result verifies our theoretical findings beyond kernel methods.

**Benign Over-fitting of Physics-Informed Interpolator** Following Benning & Burger (2018), we verify the benign overfitting behavior by plotting the noise profiles of the Physics-Informed interpolator. A noise profile characterizes the sensitivity of a learning procedure to noise in the training set, specifically how the asymptotic risk varies with the variance of additive Gaussian noise. We plot the noise profiles of both the regression interpolator and the Physics-Informed interpolator in Figure 1(Middle). We can see that, the standard regression interpolator performs worse under stronger noise level. Instead, the test risk of the Physics-Informed interpolator does not change too much at various noise levels. This supports our theory that Physics-Informed interpolator can still generalize well over noisy data, i.e., benign overfitting.

**The Noise Stabilization Effect** We also plotted the final output of the neural network in Figure 1. The intuition behind our theory of benign overfitting in inverse problems differs from that of standard regression because we predict $\Delta^{-1} u$ rather than $u$ in the regression setting. The operator $\Delta^{-1}$ functions as a kernel smoothing mechanism, where the Green's function serves as the kernel. This smoothing process attenuates high-frequency components, which are the dominant contributors to the prediction error, and thus effectively alleviates their impact. For general PDEs governing physical laws, most behave like differential operators, where the forward problem amplifies high-frequency components. Consequently, solving the inverse problem tends to attenuate these high-frequency components, resulting in a similar noise stabilization effect.

## 6 CONCLUSIONS

In conclusion, we study the behavior of kernel ridge and ridgeless regression methods for linear inverse problems governed by elliptic partial differential equations (PDEs). Our asymptotic analysis reveals that the PDE operator can stabilize the variance and even lead to benign overfitting in fixed-dimensional problems, exhibiting distinct behavior compared to regression problems. Another key focus of our investigation was the impact of different inductive biases introduced by minimizing various Sobolev norms as a form of (implicit) regularization. Interestingly, we found that the final convergence rate is independent of the choice of smooth enough inductive bias for both ridge and ridgeless regression methods. For the regularized least-squares estimator, our results demonstrate that all considered inductive biases can achieve the minimax optimal convergence rate, provided the regularization parameter is appropriately chosen. Notably, our analysis recovered the smoothness condition found by Empirical Bayes in the function regression setting and extended it to the minimum norm interpolation and inverse problem settings.

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

## A   ADDITIONAL NOTATIONS AND SOME USEFUL LEMMAS

For brevity, we denote simplified notation for $\leq k$ and $> k$, for function $f \in \mathcal{H}$, we define $f_{\leq k} := \phi^*_{\leq k} \phi_{\leq k} f$, for operator $\mathcal{A} : \mathcal{H} \to \mathcal{H}$, we also define $\mathcal{A}_{\leq k} : f_{\leq k} \mapsto \phi^*_{\leq k} \phi_{\leq k} \mathcal{A} f_{\leq k}$.

We denote $\mu_n(M)$ as the $n$-th largest eigenvalue of some matrix $M$. We also define $id_{\leq k}$ and $id_{>k}$. We denote $[n]$ as integers between 1 and $n$.

$\phi_{\leq k} \hat{S}^*_n$ is the map from $\mathbb{R}^n \to \mathbb{R}^k$, therefore, we can consider it as $k \times n$ matrix, where each column is the top $k$ features of the data points. $\hat{S}^*_n \phi_{\leq k}$ is the map from $\mathbb{R}^k \to \mathbb{R}^n$, therefore, we can consider it $n \times k$ matrix, and $(\phi_{\leq k} \hat{S}^*_n)^T = \hat{S}^*_n \phi_{\leq k}$. Similar reasoning holds for $> k$ case.

Note that for simplicity, we always convert to using $\psi$ for convenient computation, by using the following: $\phi_{\leq k} = \Lambda^{\leq k}_{\Sigma^{1/2}} \psi_{\leq k}$ and $\phi^*_{\leq k} = \psi^*_{\leq k} \Lambda^{\leq k}_{\Sigma^{1/2}}$, also similar for $> k$. This is because $\mathbb{E}([\hat{S}_n \psi^*_{>k}]^2_{ji}) = 1$ by Lemma A.5.

Next we deliver several useful lemmas.

The following lemma justifies our $< k$ and $\geq k$ decomposition.

**Lemma A.1** (Decomposition lemma)**.** *The following holds:*

  *1. For any function $f \in \mathcal{H}$, $f = f_{\leq k} + f_{>k}$;*

  *2. For any operator $\mathcal{A} : \mathcal{H} \to \mathcal{H}$, $\mathcal{A} = \mathcal{A}_{\leq k} + \mathcal{A}_{>k}$;*

  *3. For the spectrally transformed kernel matrix $\tilde{K}$, $\tilde{K} = \tilde{K}_{\leq k} + \tilde{K}_{>k}$.*

*Proof.* We first prove (1),

$$
f_{\leq k} + f_{>k} = \phi^*_{\leq k} \begin{pmatrix} \langle f, \phi_1 \rangle_\mathcal{H} \\ \langle f, \phi_2 \rangle_\mathcal{H} \\ \cdots \\ \langle f, \phi_k \rangle_\mathcal{H} \end{pmatrix} + \phi^*_{>k} \begin{pmatrix} \langle f, \phi_{k+1} \rangle_\mathcal{H} \\ \langle f, \phi_{k+2} \rangle_\mathcal{H} \\ \cdots \end{pmatrix} = \sum_{i=1}^{k} \langle f, \phi_i \rangle_\mathcal{H} \phi_i + \sum_{i=k+1}^{\infty} \langle f, \phi_i \rangle_\mathcal{H} \phi_i
$$

$$
= \sum_{i=1}^{\infty} \langle f, \phi_i \rangle_\mathcal{H} \phi_i = f.
$$

Then we move on to (2), for any $f \in \mathcal{H}$, we have

$$
(\mathcal{A}_{\leq k} + \mathcal{A}_{>k}) f = (\mathcal{A}f)_{\leq k} + (\mathcal{A}f)_{>k} = \mathcal{A}f. \quad \text{(By (1))}
$$

Finally we prove the statement (3) , this is because

$$
\tilde{K} = \hat{S}_n \mathcal{A}^2 \Sigma^{\beta-1} \hat{S}^*_n = \hat{S}_n (\mathcal{A}^2_{\leq k} \Sigma^{\beta-1}_{\leq k} + \mathcal{A}^2_{>k} \Sigma^{\beta-1}_{>k}) \hat{S}^*_n = \hat{S}_n \mathcal{A}^2_{\leq k} \Sigma^{\beta-1}_{\leq k} \hat{S}^*_n + \hat{S}_n \mathcal{A}^2_{>k} \Sigma^{\beta-1}_{>k} \hat{S}^*_n = \tilde{K}_{\leq k} + \tilde{K}_{>k}.
$$

$\square$

In the following lemma modified from Barzilai & Shamir (2023), we give a lemma which is useful for bounding $\hat{f}(y)_{\leq k}$'s norm in bounding bias and variance in D.3, E.1.

**Lemma A.2.** *Denote $\hat{f}(y) := \mathcal{A}\Sigma^{\beta-1} \hat{S}^*_n (\tilde{K}^\gamma)^{-1} y$ (highlight its dependence on y), we have*

$$
\underbrace{\phi_{\leq k} \hat{f}(y)_{\leq k}}_{k \times 1} + \underbrace{\phi_{\leq k} \mathcal{A}_{\leq k} \Sigma^{\beta-1}_{\leq k} \hat{S}^*_n}_{k \times n} \underbrace{(\tilde{K}^\gamma_{>k})^{-1}}_{n \times n} \underbrace{\hat{S}_n \mathcal{A}_{\leq k} \hat{f}(y)_{\leq k}}_{n \times 1} = \underbrace{\phi_{\leq k} \mathcal{A}_{\leq k} \Sigma^{\beta-1}_{\leq k} \hat{S}^*_n}_{k \times n} \underbrace{(\tilde{K}^\gamma_{>k})^{-1}}_{n \times n} \underbrace{y}_{n \times 1},
$$

*where $\tilde{K}^\gamma_{>k}$ is the regularized version of spectrally transformed matrix, defined as $\hat{S}_n \mathcal{A}^2_{>k} \Sigma^{\beta-1}_{>k} \hat{S}^*_n + n\gamma_n I$.*

*Proof.* First we discuss the ridgeless case i.e. $\gamma_n = 0$, where $\hat{f}$ is the minimum norm solution, then $\hat{f}_{>k}$ is also the minimum norm solution to $\hat{S}_n \mathcal{A}_{>k} \hat{f}_{>k} = y - \hat{S}_n \mathcal{A}_{\leq k} \hat{f}_{\leq k}$, then similar to 3 we can write

$$
\hat{f}_{>k} = \mathcal{A}\Sigma^{\beta-1} \hat{S}^*_n (\hat{S}_n \mathcal{A}^2_{>k} \Sigma^{\beta-1}_{>k} \hat{S}^*_n)^{-1} (y - \hat{S}_n \mathcal{A}_{\leq k} \hat{f}_{\leq k}).
$$

Therefore,

$$
\phi_{>k} \hat{f}_{>k} = \Lambda^{>k}_{\mathcal{A}\Sigma^{\beta-1}} \phi_{>k} \hat{S}^*_n (\hat{S}_n \mathcal{A}^2_{>k} \Sigma^{\beta-1}_{>k} \hat{S}^*_n)^{-1} (y - \hat{S}_n \phi^*_{\leq k} \Lambda^{\leq k}_{\mathcal{A}} \phi_{\leq k} \hat{f}_{\leq k}).
$$

As such, we obtain min norm interpolator is the the minimizer of following

$$
\phi \hat{f}(y) = \underset{\hat{f}_{\leq k}}{\arg\min} \, v(\phi_{\leq k} \hat{f}_{\leq k})
$$

$$
:= [(\phi_{\leq k} \hat{f}_{\leq k})^T, (y - \hat{S}_n \phi^*_{\leq k} \Lambda^{\leq k}_{\mathcal{A}} \phi_{\leq k} \hat{f}_{\leq k})^T (\hat{S}_n \mathcal{A}^2_{>k} \Sigma^{\beta-1}_{>k} \hat{S}^*_n)^{-1} (\phi_{>k} \hat{S}^*_n)^T \Lambda^{>k}_{\mathcal{A}\Sigma^{\beta-1}}].
$$

The vector $\phi\hat{f}(y)$ gives minimum norm if for any additional vector $\eta_{\leq k} \in \mathbb{R}^k$ we have $v(\phi_{\leq k}\hat{f}_{\leq k}(y)) \perp v(\phi_{\leq k}\hat{f}_{\leq k}(y) + \eta_{\leq k}) - v(\phi_{\leq k}\hat{f}_{\leq k}(y))$ in $\mathcal{H}^\beta$ norm.

We first write out the second vector

$$v(\phi_{\leq k}\hat{f}_{\leq k}(y)+\eta_{\leq k})-v(\phi_{\leq k}\hat{f}_{\leq k}(y)) = [\eta_{\leq k}^T, -\eta_{\leq k}^T\Lambda_{\mathcal{A}}^{\leq k}(\hat{S}_n\phi_{\leq k}^*)^T(\hat{S}_n\mathcal{A}_{>k}^2\Sigma_{>k}^{\beta-1}\hat{S}_n^*)^{-1}(\phi_{>k}\hat{S}_n^*)^T\Lambda_{\mathcal{A}\Sigma^{\beta-1}}^{>k}].$$

Then we compute the inner product w.r.t. $\mathcal{H}^\beta$ norm, by A.3 we have:

$$\eta_{\leq k}^T\Lambda_{\Sigma^{1-\beta}}^{\leq k}(\phi_{\leq k}\hat{f}_{\leq k})$$
$$-\eta_{\leq k}^T\Lambda_{\mathcal{A}}^{\leq k}(\hat{S}_n\phi_{\leq k}^*)^T\underbrace{(\hat{S}_n\mathcal{A}_{>k}^2\Sigma_{>k}^{\beta-1}\hat{S}_n^*)^{-1}}_{(1)}\underbrace{(\phi_{>k}\hat{S}_n^*)^T\Lambda_{\mathcal{A}\Sigma^{\beta-1}}^{>k}\Lambda_{\Sigma^{1-\beta}}^{>k}\Lambda_{\mathcal{A}\Sigma^{\beta-1}}^{>k}(\phi_{>k}\hat{S}_n^*)}_{(2)}$$
$$(\hat{S}_n\mathcal{A}_{>k}^2\Sigma_{>k}^{\beta-1}\hat{S}_n^*)^{-1}(y - \hat{S}_n\phi_{\leq k}^*\Lambda_{\mathcal{A}}^{\leq k}\phi_{\leq k}\hat{f}_{\leq k}) = 0.$$

Note that (1) and (2) cancel out, and since the equality above holds for any $\eta_{\leq k}$, we have:

$$\Lambda_{\Sigma^{1-\beta}}^{\leq k}(\phi_{\leq k}\hat{f}_{\leq k}) - \Lambda_{\mathcal{A}}^{\leq k}(\hat{S}_n\phi_{\leq k}^*)^T(\hat{S}_n\mathcal{A}_{>k}^2\Sigma_{>k}^{\beta-1}\hat{S}_n^*)^{-1}(y - \hat{S}_n\phi_{\leq k}^*\Lambda_{\mathcal{A}}^{\leq k}\phi_{\leq k}\hat{f}_{\leq k}) = 0.$$

Therefore,

$$\phi_{\leq k}\hat{f}_{\leq k} - \Lambda_{\mathcal{A}\Sigma^{\beta-1}}^{\leq k}\phi_{\leq k}\hat{S}_n^*(\tilde{K}_{>k}^\gamma)^{-1}(y - \hat{S}_n\mathcal{A}\hat{f}_{\leq k}) = 0.$$

With some simple algebraic manipulation we can obtain the required identity

$$\phi_{\leq k}\hat{f}_{\leq k} + \phi_{\leq k}\mathcal{A}_{\leq k}\Sigma_{\leq k}^{\beta-1}\hat{S}_n^*(\tilde{K}_{>k}^\gamma)^{-1}\hat{S}_n\mathcal{A}\hat{f}_{\leq k} = \phi_{\leq k}\mathcal{A}_{\leq k}\Sigma_{\leq k}^{\beta-1}\hat{S}_n^*(\tilde{K}_{>k}^\gamma)^{-1}y.$$

This finishes our discussion on ridgeless case.

For the regularized case i.e. $\gamma_n > 0$, first we prove

$$\hat{f}(y)_{\leq k} + \mathcal{A}_{\leq k}\Sigma_{\leq k}^{\beta-1}\hat{S}_n^*(\tilde{K}_{>k}^\gamma)^{-1}\hat{S}_n\mathcal{A}_{\leq k}\hat{f}(y)_{\leq k} = \mathcal{A}_{\leq k}\Sigma_{\leq k}^{\beta-1}\hat{S}_n^*(\tilde{K}_{>k}^\gamma)^{-1}y.$$

We know by A.1 $\tilde{K}^\gamma = \tilde{K} + n\gamma I = (\tilde{K}_{>k} + n\gamma I) + \tilde{K}_{\leq k} = \tilde{K}_{>k}^\gamma + \tilde{K}_{\leq k}$, we split $\tilde{K}^\gamma$ into two parts: $\tilde{K}_{>k}^\gamma$ and $\tilde{K}_{\leq k}$. Accordingly, $\hat{f}(y)_{\leq k}$ can be represented as

$$\hat{f}(y)_{\leq k} = \phi_{\leq k}^*\phi_{\leq k}\hat{f}(y) = \phi_{\leq k}^*\phi_{\leq k}\mathcal{A}\Sigma^{\beta-1}\hat{S}_n^*(\tilde{K}^\gamma)^{-1}y$$
$$= \mathcal{A}_{\leq k}\Sigma_{\leq k}^{\beta-1}\hat{S}_n^*(\tilde{K}_{>k}^\gamma + \tilde{K}_{\leq k})^{-1}y.$$

Therefore, taking it back to LHS, we have

$$\hat{f}(y)_{\leq k} + \mathcal{A}_{\leq k}\Sigma_{\leq k}^{\beta-1}\hat{S}_n^*(\tilde{K}_{>k}^\gamma)^{-1}\hat{S}_n\mathcal{A}_{\leq k}\hat{f}(y)_{\leq k} \text{ (LHS)}$$
$$= \mathcal{A}_{\leq k}\Sigma_{\leq k}^{\beta-1}\hat{S}_n^*(\tilde{K}_{>k}^\gamma + \tilde{K}_{\leq k})^{-1}y$$
$$+ \mathcal{A}_{\leq k}\Sigma_{\leq k}^{\beta-1}\hat{S}_n^*(\tilde{K}_{>k}^\gamma)^{-1}\underbrace{\hat{S}_n\mathcal{A}_{\leq k}\mathcal{A}_{\leq k}\Sigma_{\leq k}^{\beta-1}\hat{S}_n^*}_{\text{equals to } \tilde{K}_{\leq k}}(\tilde{K}_{>k}^\gamma + \tilde{K}_{\leq k})^{-1}y \qquad \text{(Expand } \hat{f}(y)_{\leq k})$$
$$= \mathcal{A}_{\leq k}\Sigma_{\leq k}^{\beta-1}\hat{S}_n^*(\tilde{K}_{>k}^\gamma)^{-1}(\tilde{K}_{>k}^\gamma + \tilde{K}_{\leq k})(\tilde{K}_{>k}^\gamma + \tilde{K}_{\leq k})^{-1}y$$
$$= \mathcal{A}_{\leq k}\Sigma_{\leq k}^{\beta-1}\hat{S}_n^*(\tilde{K}_{>k}^\gamma)^{-1}y \text{ (RHS)}.$$

We project LHS and RHS back to $\mathbb{R}^k$ for convenient usage in D.3, E.1, we project the functions in $\mathcal{H}$ back to $\mathbb{R}^k$ so we use $\phi_k$ in both two sides and we obtain

$$\phi_{\leq k}\hat{f}(y)_{\leq k} + \phi_{\leq k}\mathcal{A}_{\leq k}\Sigma_{\leq k}^{\beta-1}\hat{S}_n^*(\tilde{K}_{>k}^\gamma)^{-1}\hat{S}_n\mathcal{A}_{\leq k}\hat{f}(y)_{\leq k} = \phi_{\leq k}\mathcal{A}_{\leq k}\Sigma_{\leq k}^{\beta-1}\hat{S}_n^*(\tilde{K}_{>k}^\gamma)^{-1}y,$$

which concludes the proof. $\qquad\square$

This lemma justifies we can switch between using Sobolev norm and matrix norm by using $\phi$.

**Lemma A.3** (Equivalence between Sobolev norm and Matrix norm). *For any function $f \in \mathcal{H}^{\beta'}$, we have*

$$\|f\|_{\mathcal{H}^{\beta'}}^2 = \|\phi f\|_{\Lambda_{\Sigma^{1-\beta'}}}^2.$$

*And additionally,* $\|f_{\leq k}\|_{\mathcal{H}^{\beta'}}^2 = \|\phi_{\leq k}f_{\leq k}\|_{\Lambda_{\Sigma^{1-\beta'}}^{\leq k}}^2$, $\|f_{>k}\|_{\mathcal{H}^{\beta'}}^2 = \|\phi_{>k}f_{>k}\|_{\Lambda_{\Sigma^{1-\beta'}}^{>k}}^2$.

*Proof.* According to the definition of Sobolev norm, we have

$$\begin{aligned}
\text{LHS} &= \|\Sigma^{\frac{1-\beta'}{2}}f\|_{\mathcal{H}}^2 \\
&= \|\phi\Sigma^{(1-\beta')/2}f\|^2 \qquad \text{(by isometry i.e. } \|f\|_{\mathcal{H}} = \|\phi f\|^2) \\
&= \|\Lambda_{\Sigma^{(1-\beta')/2}}\phi f\|^2 \qquad \text{(by } \phi\phi^* = id : \ell_2^\infty \to \ell_2^\infty) \\
&= \|\phi f\|_{\Lambda_{\Sigma^{1-\beta'}}}^2 = \text{RHS}.
\end{aligned}$$

Then for the $\leq k$ case, we have

$$\|f_{\leq k}\|_{\mathcal{H}^{\beta'}} = \|\phi f_{\leq k}\|_{\Lambda_{\Sigma^{1-\beta'}}}^2$$

Since $(\phi f_{\leq k})_{\leq k} = \phi_{\leq k}f_{\leq k}$, all its $> k$ entries are zero, then

$$\|\phi f_{\leq k}\|_{\Lambda_{\Sigma^{1-\beta'}}}^2 = (\phi f_{\leq k})^T\Lambda_{\Sigma^{1-\beta'}}(\phi f_{\leq k}) = (\phi f_{\leq k})^T\Lambda_{\Sigma^{1-\beta'}}^{\leq k}(\phi f_{\leq k}) = \|\phi_{\leq k}f_{\leq k}\|_{\Lambda_{\Sigma^{1-\beta'}}^{\leq k}}^2.$$

The proof above works similarly for the $> k$ case. $\qquad\square$

**Lemma A.4** (Separation of $< k$ and $> k$ case). *For any function $f \in \mathcal{H}^{\beta'}$, then*

$$\|f\|_{\mathcal{H}^{\beta'}}^2 = \|f_{\leq k}\|_{\mathcal{H}^{\beta'}}^2 + \|f_{>k}\|_{\mathcal{H}^{\beta'}}^2.$$

*Proof.*

$$\begin{aligned}
\|f\|_{\mathcal{H}^{\beta'}}^2 &= \|\phi\Sigma^{(1-\beta')/2}f\|^2 \\
&= \sum_{i=1}^\infty [\phi\Sigma^{(1-\beta')/2}f]_i^2 = \sum_{i=1}^k [\phi\Sigma^{(1-\beta')/2}f]_i^2 + \sum_{i=k+1}^\infty [\phi\Sigma^{(1-\beta')/2}f]_i^2 \\
&= \|\phi_{\leq k}\Sigma_{\leq k}^{(1-\beta')/2}f_{\leq k}\|^2 + \|\phi_{>k}\Sigma_{>k}^{(1-\beta')/2}f_{>k}\|^2 \\
&= \|f_{\leq k}\|_{\mathcal{H}^{\beta'}}^2 + \|f_{>k}\|_{\mathcal{H}^{\beta'}}^2.
\end{aligned}$$

$\qquad\square$

**Lemma A.5.** $\mathbb{E}([\hat{S}_n\psi_{>k}^*]_{ji}^2) = 1$ *holds for any* $i > k, j \in [n]$.

*Proof.*

$$\mathbb{E}([\hat{S}_n\psi_{>k}^*]_{ji}^2) = \mathbb{E}([\langle\psi_i, K_{x_j}\rangle_{\mathcal{H}}^2]) = \mathbb{E}(\psi_i(x_j)^2) = 1.$$

$\qquad\square$

Last we present a lemma which is useful in $> k$ case in deriving bias's bound.

**Lemma A.6.**
$$(A + UCV)^{-1}U = A^{-1}U(I + CVA^{-1}U)^{-1}.$$

*Proof.* By Sherman-Morrison-Woodbury formula we have

$$(A + UCV)^{-1} = A^{-1} - A^{-1}U(C^{-1} + VA^{-1}U)^{-1}VA^{-1}$$

Therefore,

$$\begin{aligned}
(A + UCV)^{-1}U &= A^{-1}U - A^{-1}U(C^{-1} + VA^{-1}U)^{-1}VA^{-1}U \\
&= A^{-1}U(I - (C^{-1} + VA^{-1}U)^{-1}VA^{-1}U) \\
&= A^{-1}U(I - (C^{-1} + VA^{-1}U)^{-1}(C^{-1} + VA^{-1}U) + (C^{-1} + VA^{-1}U)^{-1}C^{-1}) \\
&= A^{-1}U(I - I + (C(C^{-1} + VA^{-1}U))^{-1}) \\
&= A^{-1}U(I + CVA^{-1}U)^{-1}.
\end{aligned}$$

$\qquad\square$

## A.1 PROOF OF LEMMA 3

*Proof.* As mentioned in Definition 2.1, we have $\|f\|_{\mathcal{H}^\beta} = \|\Sigma^{\frac{1-\beta}{2}} f\|_{\mathcal{H}}$ thus we can rewrite the objective function (3) as

$$\hat{f}_\gamma = \arg\min \frac{1}{n}\|\hat{S}_n\mathcal{A}f-y\|^2+\gamma_n\|\Sigma^{\frac{1-\beta}{2}}f\|_{\mathcal{H}} \Leftrightarrow \Sigma^{\frac{1-\beta}{2}}\hat{f}_\gamma = \arg\min \frac{1}{n}\|\hat{S}_n\mathcal{A}\Sigma^{\frac{\beta-1}{2}}g-y\|^2+\gamma_n\|g\|_{\mathcal{H}}.$$

By representer theorem for inverse problem (Unser, 2021), the solution of the optimization problem $g_\gamma = \arg\min \frac{1}{n}\|\hat{S}_n\mathcal{A}\Sigma^{\frac{\beta-1}{2}}g - y\|^2 + \gamma_n\|g\|_{\mathcal{H}}$ have the finite dimensional representation that $g_\gamma = \mathcal{A}\Sigma^{\frac{\beta-1}{2}}\hat{S}_n^*\hat{\theta}_n$ for some $\hat{\theta}_n \in \mathbb{R}^n$. Then we know the $\hat{f}_\gamma = \Sigma^{\frac{\beta-1}{2}}g_\gamma = \mathcal{A}\Sigma^{\beta-1}\hat{S}_n^*\hat{\theta}_n$, for some $\hat{\theta}_n \in \mathbb{R}^n$. Plug the finite dimensional representation of $\hat{f}_\gamma$ to objective function (3) thus we have

$$\hat{\theta}_n = \arg\min_{\theta_n \in \mathbb{R}^n} \frac{1}{n}\|\hat{S}_n\mathcal{A}^2\Sigma^{\beta-1}\hat{S}_n^*\hat{\theta}_n - y\|^2 + \gamma_n\|\Sigma^{\frac{1-\beta}{2}}\mathcal{A}\Sigma^{\beta-1}\hat{S}_n^*\theta_n\|_{\mathcal{H}}^2.$$

Thus we have $\hat{\theta}_n = (\hat{S}_n\mathcal{A}^2\Sigma^{\beta-1}\hat{S}_n^*\hat{S}_n\mathcal{A}^2\Sigma^{\beta-1}\hat{S}_n^* + \gamma_n\hat{S}_n\mathcal{A}^2\Sigma^{\beta-1}\hat{S}_n^*)^{-1}(\hat{S}_n\mathcal{A}^2\Sigma^{\beta-1}\hat{S}_n^*)y = (\hat{S}_n\mathcal{A}^2\Sigma^{\beta-1}\hat{S}_n^* + n\gamma_nI)^{-1}y$. (For $\mathcal{A}$ is self-adjoint and co-diagonalizable with $\Sigma$.) $\square$

## B CONCENTRATION LEMMAS

Here we present several lemmas for bounding several quantities in D, E.

**Lemma B.1.** *Let $k \in [n]$, $a$ be the power of $\mathcal{A}$, and $b$ be the power of $\Sigma$, we bound the trace of this $n \times n$ matrix, w.p. at least $1 - 2\exp(-\frac{1}{2\beta_k^2}n)$ we have*

$$\frac{1}{2}n\sum_{i>k}p_i^a\lambda_i^b \leq \mathrm{tr}(\hat{S}_n\psi_{>k}^*\Lambda_{\mathcal{A}^a\Sigma^b}^{>k}\psi_{>k}\hat{S}_n^*) \leq \frac{3}{2}n\sum_{i>k}p_i^a\lambda_i^b.$$

*Proof.* Note that $\Lambda_{\mathcal{A}^a\Sigma^b}^{>k}$ is a diagonal matrix with entry $p_i^a\lambda_i^b$ ($i > k$).

$$\mathrm{tr}(\hat{S}_n\psi_{>k}^*\Lambda_{\mathcal{A}^a\Sigma^b}^{>k}\psi_{>k}\hat{S}_n^*) = \sum_{j=1}^n [(\hat{S}_n\psi_{>k}^*)(\Lambda_{\mathcal{A}^a\Sigma^b}^{>k})(\psi_{>k}\hat{S}_n^*)]_{jj} = \sum_{j=1}^n \underbrace{\sum_{i=k+1}^\infty p_i^a\lambda_i^b[\hat{S}_n\psi_{>k}^*]_{ji}^2}_{v_j}.$$

Here we denote the term inside $j$ summation as $v_j$, then by A.5, the expectation of the trace is

$$n\sum_{i>k}p_i^a\lambda_i^b.$$

We also know that $v_j$ is lower bounded by 0 and by def. of $\beta_k$ 3.3, it can be upper bounded by

$$v_j = \sum_{i=k+1}^\infty p_i^a\lambda_i^b\psi_i(x_j)^2 \leq \beta_k\underbrace{\sum_{i=k+1}^\infty p_i^a\lambda_i^b}_{\text{denoted as } M}.$$

Then we have $0 \leq v_j \leq M$ for all $j$ and $v_j$ is independent, we can apply the Hoeffding's inequality to bound $\sum_{j=1}^n v_j$:

$$\mathbb{P}(|\sum_{j=1}^n v_j - n\sum_{i>k}p_i^a\lambda_i^b| \geq t) \leq 2\exp\left(\frac{-2t^2}{nM^2}\right).$$

We then pick $t := \frac{n}{2}\sum_{i>k}p_i^a\lambda_i^b$, and we get $\frac{-2t^2}{nM^2} = -\frac{1}{2\beta_k^2}n$, and we know the trace value exactly corresponds to $\sum_{j=1}^n v_j$.

Therefore, w.p. at least $1 - 2\exp(-\frac{1}{2\beta_k^2}n)$,

$$\frac{1}{2}n\sum_{i>k}p_i^a\lambda_i^b \leq \mathrm{tr}(\hat{S}_n\psi_{>k}^*\Lambda_{\mathcal{A}^a\Sigma^b}^{>k}\psi_{>k}\hat{S}_n^*) \leq \frac{3}{2}n\sum_{i>k}p_i^a\lambda_i^b.$$

$\square$

Here we present the modified version of Lemma 2 in Barzilai & Shamir (2023), we rewrite it to fit into our framework for completeness.

**Lemma B.2.** *For any $k \in [n]$ there exists some absolute constant $c', c_2 > 0$ s.t. the following hold simultaneously w.p. at least $1 - 2\exp(-\frac{c'}{\beta_k}\max\{\frac{n}{k}, \log(k)\})$*

1. $\mu_k(\underbrace{\psi_{\leq k}\hat{S}_n^*\hat{S}_n\psi_{\leq k}^*}_{k \times k}) \geq \max\{\sqrt{n} - \sqrt{\frac{1}{2}\max\{n, \beta_k(1 + \frac{1}{c'}k\log(k))\}}, 0\}^2$;

2. $\mu_1(\underbrace{\psi_{\leq k}\hat{S}_n^*\hat{S}_n\psi_{\leq k}^*}_{k \times k}) \leq c_2\max\{n, \beta_k k\log(k)\}$.

*Moreover, there exists some $c > 0$ s.t. if $c\beta_k k\log(k) \leq n$ then w.p. at least $1 - 2\exp(-\frac{c'}{\beta_k}\frac{n}{k})$ and some absolute constant $c_1 > 0$ it holds that*

$$c_1 n \leq \mu_k(\psi_{\leq k}\hat{S}_n^*\hat{S}_n\psi_{\leq k}^*) \leq \mu_1(\psi_{\leq k}\hat{S}_n^*\hat{S}_n\psi_{\leq k}^*) \leq c_2 n.$$

*Proof.* We will bound the singular values $\sigma_i(\underbrace{\hat{S}_n\psi_{\leq k}^*}_{n \times k})$ since $\sigma_i(A)^2 = \mu_i(A^T A)$ for any matrix $A$.

We know rows of this matrix are independent isotropic random vectors in $\mathbb{R}^k$, where randomness is over the choice of $x$, where by the definition of $\beta_k$ 3.3 the rows are heavy-tailed having norm bounded by

$$\|\text{each row of } \hat{S}_n\psi_{\leq k}^*\| \leq \sqrt{k\beta_k}.$$

Here we can use Vershynin (2011)[Theorem 5.41] which is applicable for heavy-tailed rows, there is some absolute constant $c' > 0$ s.t. for every $t \geq 0$, one has that w.p. at least $1 - 2k\exp(-2c't^2)$

$$\sqrt{n} - t\sqrt{k\beta_k} \leq \sigma_k(\hat{S}_n\psi_{\leq k}^*) \leq \sigma_1(\hat{S}_n\psi_{\leq k}^*) \leq \sqrt{n} + t\sqrt{k\beta_k}.$$

We pick $t = \sqrt{\frac{1}{2\beta_k}\max\{\frac{n}{k}, \log(k)\} + \frac{\log(k)}{2c'}}$, then w.p. at least $1 - 2\exp(\frac{-c'}{\beta_k}\max\{\frac{n}{k}, \log(k)\})$ it holds that

$$\sigma_1\left(\hat{S}_n\psi_{\leq k}^*\right)^2 \leq \left(\sqrt{n} + \sqrt{\frac{1}{2}\max(n, k\log(k)) + k\log(k)\frac{\beta_k}{2c'}}\right)^2$$

$$\leq \left(\sqrt{n} + \frac{1}{\sqrt{2}}\sqrt{n + \left(1 + \frac{\beta_k}{c'}\right)k\log(k)}\right)^2$$

$$\leq 3n + \left(1 + \frac{\beta_k}{c'}\right)k\log(k),$$

where the last inequality followed from the fact that $(a + b)^2 \leq 2(a^2 + b^2)$ for any $a, b \in \mathbb{R}$. Since $\beta_k \geq 1$ 3.3, we obtain $\sigma_1\left(\hat{S}_n\psi_{\leq k}^*\right)^2 \leq c_2\max\{n, \beta_k k\log(k)\}$ for a suitable $c_2 > 0$, proving (2).

For the lower bound, we simultaneously have

$$\sigma_k\left(\hat{S}_n\psi_{\leq k}^*\right) \geq \sqrt{n} - \frac{1}{\sqrt{2}}\sqrt{\frac{1}{2}\max(n, k\log(k)) + k\log(k)\frac{\beta_k}{2c'}}$$

$$\geq \sqrt{n} - \sqrt{\frac{1}{2}\max\left(n, \beta_k\left(1 + \frac{1}{c'}\right)k\log(k)\right)}.$$

Since the singular values are non-negative, the above implies

$$\sigma_k\left(\hat{S}_n\psi_{\leq k}^*\right) \geq \max\{\sqrt{n} - \sqrt{\frac{1}{2}\max\left(n, \beta_k\left(1 + \frac{1}{c'}\right)k\log(k)\right)}, 0\}^2$$

which proves (1).

Next we move on to prove the moreover part, taking $c = (1 + \frac{1}{c'})$ we now have by assumption that $\frac{n}{k} \geq c\beta_k \log(k) \geq \log(k)$ (where we used the fact that $c \geq 1$ and $\beta_k \geq 1$), the probability that (1) and (2) hold is $1 - 2\exp(-\frac{c'}{\beta_k}\frac{n}{k})$. Furthermore, plugging $c\beta_k k \log(k) \leq n$ into the lower bound (1) obtains the following

$$\mu_k\left(\psi_{\leq k}\hat{S}_n^*\hat{S}_n\psi_{\leq k}^*\right) \geq \max\left(\sqrt{n} - \sqrt{\frac{1}{2}\max\left(n, c\beta_k k \log(k)\right)}, 0\right)^2$$

$$\geq \left(\sqrt{n} - \sqrt{\frac{n}{2}}\right)^2 = \left(1 - \frac{1}{\sqrt{2}}\right)^2 n.$$

Similarly since $\beta_k k \log(k) \leq n$, the upper bound (2) becomes

$$\mu_1\left(\psi_{\leq k}\hat{S}_n^*\hat{S}_n\psi_{\leq k}^*\right) \leq c_2 n.$$

$\square$

**Lemma B.3.** *There exists some constant $c, c', c_1, c_2 > 0$ s.t. for any $k \in \mathbb{N}$ with $c\beta_k k \log(k) \leq n$, it holds w.p. at least $1 - 8\exp(-\frac{c'}{\beta_k^2}\frac{n}{k})$, the following hold simultaneously*

1. $c_1 n \sum_{i>k} p_i^{-2}\lambda_i^{-\beta'} \leq \operatorname{tr}(\hat{S}_n\psi_{\leq k}^*\Lambda_{\mathcal{A}^{-2}\Sigma^{-\beta'}}^{\leq k}\psi_{\leq k}\hat{S}_n^*) \leq c_2 n \sum_{i>k} p_i^{-2}\lambda_i^{-\beta'}$;

2. $c_1 n \sum_{i>k} p_i^2\lambda_i^{-\beta'+2\beta}\operatorname{tr}(\hat{S}_n\psi_{\leq k}^*\Lambda_{\mathcal{A}^2\Sigma^{-\beta'+2\beta}}^{\leq k}\psi_{\leq k}\hat{S}_n^*) \leq c_2 n \sum_{i>k} p_i^2\lambda_i^{-\beta'+2\beta}$;

3. $\mu_k(\psi_{\leq k}\hat{S}_n^*\hat{S}_n\psi_{\leq k}^*) \geq c_1 n$;

4. $\mu_1(\psi_{\leq k}\hat{S}_n^*\hat{S}_n\psi_{\leq k}^*) \leq c_2 n$.

*Proof.* By Lemma B.1, (1) and (2) each hold w.p. at least $1 - 2\exp(-\frac{1}{2\beta_k^2}n)$, so the probability of they both hold is at least $(1 - 2\exp(-\frac{1}{2\beta_k^2}n))^2$. And by Lemma B.2, (3), (4) simultaneously holds with probability at least $1 - 2\exp(-\frac{c'}{\beta_k}\frac{n}{k})$. Therefore, the probability of all four statements hold is at least

$$(1 - 2\exp(-\frac{1}{2\beta_k^2}n))^2(1 - 2\exp(-\frac{c'}{\beta_k}\frac{n}{k}))$$

$$\geq 1 - 8\exp(-\min\{\frac{1}{2\beta_k^2}n, \frac{c'}{\beta_k}\frac{n}{k}\})$$

$$\geq 1 - 8\exp\{-\min(\frac{1}{2\beta_k^2}, \frac{c'}{\beta_k})\frac{n}{k}).$$

Since we know $\beta_k \geq 1$ 3.3, then we replace $c'$ with $\min\{\frac{1}{2}, c'\}$ results in the desired bound holding w.p. at least $1 - 8\exp(-\frac{c'}{\beta_k^2}\frac{n}{k})$.

$\square$

**Lemma B.4** (Concentration bounds on $\|\hat{S}_n\mathcal{A}_{>k}f_{>k}^*\|^2$ in E.1)**.** *For any $k \in [n]$ and $\delta > 0$, it holds w.p. at least $1 - \delta$ that*

$$\|\hat{S}_n\mathcal{A}_{>k}f_{>k}^*\|^2 \leq \frac{1}{\delta}n\|\phi_{>k}\mathcal{A}_{>k}f_{>k}^*\|_{\Sigma_{>k}}^2.$$

*Proof.* Let $v_j := \langle \mathcal{A}_{>k} f^*_{>k}, K_{x_j} \rangle^2_{\mathcal{H}}$, then LHS is equal to $\sum_{j=1}^n v_j$. Since $x_j$ is independent, it holds that $v_j$ are independent random variables with mean

$$
\mathbb{E}[v_j] = \mathbb{E}[\langle \phi^*_{>k}\phi_{>k}\mathcal{A}_{>k}f^*_{>k}, \sum_{i=1}^\infty \phi_i(x_j)\phi_i \rangle^2_{\mathcal{H}}]
$$

$$
= \mathbb{E}[\langle \sum_{i=k+1}^\infty [\phi_{>k}\mathcal{A}_{>k}f^*_{>k}]_i \phi_i, \sum_{i=1}^\infty \phi_i(x_j)\phi_i \rangle^2_{\mathcal{H}}]
$$

$$
= \mathbb{E}[(\sum_{i=k+1}^\infty [\phi_{>k}\mathcal{A}_{>k}f^*_{>k}]_i \phi_i(x_j))^2]
$$

$$
= \sum_{i>k}\sum_{l>k} \sqrt{\lambda_i}\sqrt{\lambda_l}[\phi_{>k}\mathcal{A}_{>k}f^*_{>k}]_i[\phi_{>k}\mathcal{A}_{>k}f^*_{>k}]_l \underbrace{\mathbb{E}_{x_j}\psi_i(x_j)\psi_l(x_j)}_{=1 \text{ if } i=l; 0 \text{ otherwise}}
$$

$$
= \sum_{i>k} \lambda_i[\phi_{>k}\mathcal{A}_{>k}f^*_{>k}]_i^2 = \|\phi_{>k}\mathcal{A}_{>k}f^*_{>k}\|^2_{\Lambda^{\geq k}_\Sigma}.
$$

Then we can apply Markov's inequality:

$$
\mathbb{P}(\sum_{j=1}^n v_j \geq \frac{1}{\delta}n\|\phi_{>k}\mathcal{A}_{>k}f^*_{>k}\|^2_{\Sigma_{>k}}) \leq \delta.
$$

$\square$

## C  BOUNDS ON EIGENVALUES

**Theorem C.1.** *Suppose Assumption 3.4 holds, and eigenvalues of $\tilde{\Sigma}$ are given in non-increasing order (i.e. $2p + \beta\lambda > 0$). There exists absolute constant $c, C, c_1, c_2 > 0$ s.t. for any $k \leq k' \in [n]$ and $\delta > 0$, it holds w.p. at least $1 - \delta - 4\frac{r_k}{k^4}\exp(-\frac{c}{\beta_k}\frac{n}{r_k}) - 2\exp(-\frac{c}{\beta_k}\max(\frac{n}{k}, \log(k)))$ that*

$$
\mu_k\left(\frac{1}{n}\tilde{K}\right) \leq c_1\beta_k\left(\left(1 + \frac{k\log(k)}{n}\right)\lambda_k^\beta p_k^2 + \log(k+1)\frac{\mathrm{tr}\left(\tilde{\Sigma}_{>k}\right)}{n}\right)
$$

$$
\mu_k\left(\frac{1}{n}\tilde{K}\right) \geq c_2\mathbb{I}_{k,n}\lambda_k^\beta p_k^2 + \alpha_k\left(1 - \frac{1}{\delta}\sqrt{\frac{n^2}{\mathrm{tr}(\tilde{\Sigma}_{>k'})^2/\mathrm{tr}(\tilde{\Sigma}^2_{>k'})}}\right)\frac{\mathrm{tr}\left(\tilde{\Sigma}_{>k'}\right)}{n},
$$

*where $\mu_k$ is the $k$-th largest eigenvalue of $\tilde{K}$, $\tilde{\Sigma} := \mathcal{A}^2\Sigma^\beta$, $r_k := \mathrm{tr}(\tilde{\Sigma}_{>k})/(p_{k+1}^2\lambda_{k+1}^\beta)$, and*

$$
\mathbb{I}_{k,n} = \begin{cases} 1, & \text{if } C\beta_k k\log(k) \leq n \\ 0, & \text{otherwise} \end{cases}.
$$

*Proof.* We hereby give the proof of Theorem 3.5. From Lemma C.3, we have that

$$
\lambda_{i+k-\min(n,k)}^\beta p_{i+k-\min(n,k)}^2 \mu_{\min(n,k)}(D_k) + \mu_n(\frac{1}{n}\tilde{K}_{>k}) \leq \mu_i(\frac{1}{n}\tilde{K}) \leq \lambda_i^\beta p_i^2 \mu_1(D_k) + \mu_1(\frac{1}{n}\tilde{K}_{>k}),
$$

where $D_k$ is as defined in the lemma.

We bound the two terms at the RHS seperately. From Lemma C.6, it holds w.p. at least $1 - 4\frac{r_k}{k^4}\exp(-\frac{c'}{\beta_k}\frac{n}{r_k})$ that for some absolute constants $c', c'_1 > 0$,

$$
\mu_1(\frac{1}{n}\tilde{K}_{>k}) \leq c'_1\left(p_{k+1}^2\lambda_{k+1}^\beta + \beta_k\log(k+1)\frac{\mathrm{tr}(\tilde{\Sigma}_{>k})}{n}\right).
$$

For the other term, because $\mu_i(D_k) = \mu_i(\frac{1}{n}(\hat{S}_n\Sigma_{\leq k}^{-1/2})(\hat{S}_n\Sigma_{\leq k}^{-1/2})^T) = \mu_i(\frac{1}{n}\psi_{\leq k}\hat{S}_n^*\hat{S}_n\psi_{\leq k}^*)$, by B.2 ther exists some absolute constants $c'', c''_1 > 0$, s.t. w.p. at least $1 - 2\exp(-\frac{c''}{\beta_k}\max\{\frac{n}{k}, \log(k)\})$

$$
\lambda_i^\beta p_i^2 \mu_1(D_k) \leq c''_1\frac{1}{n}\max\{n, \beta_k k\log(k)\}\lambda_i^\beta p_i^2 \leq c''_1\beta_k\left(1 + \frac{k\log(k)}{n}\right)\lambda_i^\beta p_i^2,
$$

where the last inequality uses the fact that $\beta_k \geq 1$.

Therefore, by taking $c = \max(c', c'')$, both events hold w.p. at least $1 - \delta - 4\frac{r_k}{k^4}\exp(-\frac{c}{\beta_k}\frac{n}{r_k}) - 2\exp(-\frac{c}{\beta_k}\max(\frac{n}{k}, \log(k)))$ and the upper bound of $\mu_i(\frac{1}{n}\tilde{K})$ now becomes

$$\mu_k\left(\frac{1}{n}\tilde{K}\right) \leq c_1\beta_k\left(\left(1 + \frac{k\log(k)}{n}\right)\lambda_k^\beta p_k^2 + \log(k+1)\frac{\operatorname{tr}\left(\tilde{\Sigma}_{>k}\right)}{n}\right)$$

for some suitable absolute constant $c_1 = \max(c_1', c_1'') > 0$.

The other equation of this theorem is proved similarly as the "moreover" part in Lemma B.2, which states that $\mu_k(D_k) \geq c_2$ if $C\beta_k k \log(k) \leq n$, and from the lower bound of Lemma C.6, it holds w.p. at least $1 - \delta$. $\qquad\square$

**Lemma C.2** (Extension of Ostrowski's theorem). *We present the abstract matrix version here and we can obtain the bounds by substituting inside, let $i, k \in \mathbb{N}$ satisfy $1 \leq i \leq \min(k,n)$ and a matrix $X_k \in \mathbb{R}^{n \times k}$. Let $D_k := \frac{1}{n}X_k X_k^T \in \mathbb{R}^{n \times n}$. Suppose that the eigenvalues of $\Sigma$ are given in non-increasing order $\lambda_1 \geq \lambda_2 \geq \ldots$ then*

$$\lambda_{i+k-\min(n,k)}\mu_{\min(n,k)}(D_k) \leq \mu_i\left(\frac{1}{n}X_k\Sigma_{\leq k}X_k^\top\right) \leq \lambda_i\mu_1(D_k).$$

*Proof.* We extends Ostrowski's theorem to the non-square case, where the proof is similar to Lemma 5 in Barzilai & Shamir (2023). Let $\pi_1$ denote the number of positive eigenvalues of $\frac{1}{n}X_k\Sigma_{\leq k}X_k^T$, it follows from Dancis (1986)[Theorem 1.5, Ostrowski's theorem] that for $1 \leq i \leq \pi_1$,

$$\lambda_{i+k-\min(n,k)}\mu_{\min(n,k)}(D_k) \leq \mu_i(\frac{1}{n}X_k\Sigma_{\leq k}X_k^T) \leq \lambda_i\mu_1(D_k).$$

Now we'll only have to consider the case where $\pi_i < i$. By definition of $\pi_1$ there are some orthonormal eigenvectors of $X_k\Sigma_{\leq k}X_k^T$, $v_{\pi_1+1}, \ldots, v_n$ with eigenvalues 0. Since $\Sigma \succeq 0$, for each such 0 eigenvector $v$,

$$0 = (X_k^T v)^T \Sigma_{\leq k}(X_k^T v) \Rightarrow X_k^T v = 0.$$

In particular, $D_k$ has $v_{\pi_1+1}, \ldots, v_n$ as 0 eigenvectors and since $D_k \succeq 0$, we have that $\mu_{\pi_1+1}(D_k), \ldots, \mu_n(D_k) = 0$. So for $i > \pi_1$ we have

$$\lambda_{i+k-\min(n,k)}\mu_{\min(n,k)}(D_k) \leq \mu_i(\frac{1}{n}X_k\Sigma_{\leq k}X_k^T) \leq \lambda_i\mu_1(D_k).$$

$\qquad\square$

**Lemma C.3** (Symmetric Bound on eigenvalues of $\frac{1}{n}\tilde{K}$). *Let $i, k \in \mathbb{N}$ satisfy $1 \leq i \leq n$ and $i \leq k$, let $D_k = \frac{1}{n}\hat{S}_n\Sigma_{\leq k}^{-1}\hat{S}_n^* = \frac{1}{n}(\hat{S}_n\Sigma_{\leq k}^{-1/2})(\hat{S}_n\Sigma_{\leq k}^{-1/2})^T$, and eigenvalues of $\tilde{\Sigma}$ is non-increasing i.e. $2p + \lambda\beta > 0$, then*

$$\lambda_{i+k-\min(n,k)}^\beta p_{i+k-\min(n,k)}^2 \mu_{\min(n,k)}(D_k) + \mu_n(\frac{1}{n}\tilde{K}_{>k}) \leq \mu_i(\frac{1}{n}\tilde{K}) \leq \lambda_i^\beta p_i^2 \mu_1(D_k) + \mu_1(\frac{1}{n}\tilde{K}_{>k}).$$

*In particular*

$$\lambda_{i+k-\min(n,k)}^\beta p_{i+k-\min(n,k)}^2 \mu_{\min(n,k)}(D_k) \leq \mu_i(\frac{1}{n}\tilde{K}) \leq \lambda_i^\beta p_i^2 \mu_1(D_k) + \mu_1(\frac{1}{n}\tilde{K}_{>k}).$$

*Proof.* We can decompose $\tilde{K}$ into the sum of two hermitian matrices $\tilde{K}_{\leq k}$ and $\tilde{K}_{>k}$. Then we can use Weyl's theorem Horn & Johnson (1985)[Corollary 4.3.15] to bound the eigenvalues of $\tilde{K}$ as

$$\mu_i(\tilde{K}_{\leq k}) + \mu_n(\tilde{K}_{>k}) \leq \mu_i(\tilde{K}) \leq \mu_i(\tilde{K}_{\leq k}) + \mu_1(\tilde{K}_{>k}).$$

Then since $\tilde{K}_{\leq k} = (\hat{S}_n\Sigma_{\leq k}^{-1/2})\mathcal{A}^2\Sigma^\beta(\hat{S}_n\Sigma_{\leq k}^{-1/2})^T$, we use the extension of Ostrowski's theorem derived at Lemma C.2 to obtain the bound:

$$\lambda_{i+k-\min(n,k)}^\beta p_{i+k-\min(n,k)}^2 \mu_{\min(n,k)}(D_k) \leq \mu_i(\frac{1}{n}\tilde{K}_{\leq k}) \leq \lambda_i\mu_1(D_k).$$

Therefore, by combining the two results, it yields:

$$\lambda_{i+k-\min(n,k)}^\beta p_{i+k-\min(n,k)}^2 \mu_{\min(n,k)}(D_k) + \mu_n(\frac{1}{n}\tilde{K}_{>k}) \leq \mu_i(\frac{1}{n}\tilde{K}) \leq \lambda_i^\beta p_i^2 \mu_1(D_k) + \mu_1(\frac{1}{n}\tilde{K}_{>k}).$$

The "in particular" part follows from $\mu_n(\frac{1}{n}\tilde{K}_{>k}) \geq 0$. $\qquad\square$

**Lemma C.4** (Symmetric Bound on eigenvalues of $\frac{1}{n}\tilde{K}_{>k}$). *For any $\delta > 0$, it holds w.p. at least $1 - \delta$ that for all $i \in [n]$,*

$$\alpha_k \frac{1}{n} \operatorname{tr}(\tilde{\Sigma}_{>k}) \left(1 - \frac{1}{\delta}\sqrt{\frac{n^2}{\operatorname{tr}(\tilde{\Sigma}_{>k})^2 / \operatorname{tr}(\tilde{\Sigma}_{>k}^2)}}\right) \le \mu_i(\frac{1}{n}\tilde{K}_{>k}) \le \beta_k \frac{1}{n} \operatorname{tr}(\tilde{\Sigma}_{>k}) \left(1 + \frac{1}{\delta}\sqrt{\frac{n^2}{\operatorname{tr}(\tilde{\Sigma}_{>k})^2 / \operatorname{tr}(\tilde{\Sigma}_{>k}^2)}}\right)$$

*where $\tilde{\Sigma} := \mathcal{A}^2 \Sigma^\beta$.*

*Proof.* We decompose the matrix into the diagonal component and non-diagonal component and bound them respectively, we denote diagonal component as $\operatorname{diag}(\frac{1}{n}\tilde{K}_{>k})$ and $\Delta_{>k} := \frac{1}{n}\tilde{K}_{>k} - \operatorname{diag}(\frac{1}{n}\tilde{K}_{>k}^\gamma)$.

Recall that $\tilde{K}_{>k} := \hat{S}_n \mathcal{A}_{>k}^2 \Sigma_{>k}^{\beta-1} \hat{S}_n^*$, and for any $i \in [n]$,

$$[\frac{1}{n}\tilde{K}_{>k}]_{ii} = \frac{1}{n} \langle K_{x_i}, \mathcal{A}_{>k}^2 \Sigma_{>k}^{\beta-1} K_{x_i} \rangle_{\mathcal{H}}$$

$$= \frac{1}{n} \langle \sum_{l=1}^{\infty} \phi_l(x_i)\phi_l, \sum_{l=k+1}^{\infty} p_l^2 \lambda_l^{\beta-1} \phi_l(x_i)\phi_l \rangle_{\mathcal{H}}$$

$$= \frac{1}{n} \sum_{l=k+1}^{\infty} p_l^2 \lambda_l^\beta \psi_l(x_i)^2.$$

Therefore, by definition of $\alpha_k$ and $\beta_k$, we have

$$\alpha_k \frac{1}{n} \operatorname{tr}(\mathcal{A}_{>k}^2 \Sigma_{>k}^\beta) \le [\frac{1}{n}\tilde{K}_{>k}]_{ii} \le \beta_k \frac{1}{n} \operatorname{tr}(\mathcal{A}_{>k}^2 \Sigma_{>k}^\beta).$$

Therefore,

$$\alpha_k \frac{1}{n} \operatorname{tr}(\mathcal{A}_{>k}^2 \Sigma_{>k}^\beta) I \preceq \operatorname{diag}(\frac{1}{n}\tilde{K}_{>k}) \preceq \beta_k \frac{1}{n} \operatorname{tr}(\mathcal{A}_{>k}^2 \Sigma_{>k}^\beta) I.$$

Then by Weyl's theorem Horn & Johnson (1985)[Corollary 4.3.15], we can bound the eigenvalues of $\frac{1}{n}\tilde{K}_{>k}$ as

$$\alpha_k \frac{1}{n} \operatorname{tr}(\mathcal{A}_{>k}^2 \Sigma_{>k}^\beta) + \mu_n(\Delta_{>k}) \le \mu_i(\frac{1}{n}\tilde{K}_{>k}) \le \beta_k \frac{1}{n} \operatorname{tr}(\mathcal{A}_{>k}^2 \Sigma_{>k}^\beta) + \mu_1(\Delta_{>k}).$$

It remains to bound the eigenvalues of $\Delta_{>k}$, we first bound the expectation of the matrix norm using

$$\mathbb{E}[\|\Delta_{>k}\|] \le \mathbb{E}[\|\Delta_{>k}\|_F^2]^{1/2} = \sqrt{\sum_{i,j=1, i \ne j}^{n} \mathbb{E}[(\frac{1}{n}\sum_{l>k} p_l^2 \lambda_l^\beta \psi_l(x_i)\psi_l(x_j))^2]}$$

$$= \sqrt{\frac{n(n-1)}{n^2} \operatorname{tr}(\mathcal{A}_{>k}^4 \Sigma_{>k}^{2\beta})} \le \sqrt{\operatorname{tr}(\mathcal{A}_{>k}^4 \Sigma_{>k}^{2\beta})} = \frac{1}{n} \operatorname{tr}(\tilde{\Sigma}_{>k}) \sqrt{\frac{n^2}{\operatorname{tr}(\tilde{\Sigma}_{>k})^2 / \operatorname{tr}(\tilde{\Sigma}_{>k}^2)}}.$$

By Markov's inequality,

$$\mathbb{P}(\|\Delta_{>k}\| \ge \frac{1}{\delta}\mathbb{E}[\|\Delta_{>k}\|]) \le \delta.$$

So w.p. at least $1 - \delta$ it holds that

$$\|\Delta_{>k}\| \le \frac{1}{\delta}\mathbb{E}[\|\Delta_{>k}\|] \le \frac{1}{n\delta} \operatorname{tr}(\tilde{\Sigma}_{>k}) \sqrt{\frac{n^2}{\operatorname{tr}(\tilde{\Sigma}_{>k})^2 / \operatorname{tr}(\tilde{\Sigma}_{>k}^2)}}.$$

$\square$

**Lemma C.5** (Upper bound of largest eigenvalue). *Suppose Assumption 3.4 holds, and eigenvalues of $\tilde{\Sigma}$ are given in non-increasing order (i.e. $2p + \beta\lambda > 0$). There exists absolute constant $c$, $c' > 0$ s.t. it holds w.p. at least $1 - 4\frac{r_k}{k^4} \exp(-\frac{c'}{\beta_k}\frac{n}{r_k})$ that*

$$\mu_1 \left(\frac{1}{n}\hat{S}_n \mathcal{A}^2 \Sigma^{\beta-1} \hat{S}_n^*\right) \le c \left(p_{k+1}^2 \lambda_{k+1}^\beta + \beta_k \log(k+1)\frac{\operatorname{tr}(\tilde{\Sigma}_{>k})}{n}\right).$$

*where $\tilde{\Sigma} := \mathcal{A}^2 \Sigma^\beta$, $r_k := \frac{\operatorname{tr}(\tilde{\Sigma}_{>k})}{p_{k+1}^2 \lambda_{k+1}^\beta}$.*

*Proof.* Let $m_k = \mu_1(\frac{1}{n}\tilde{K}_{>k})$, $\tilde{K}_{k+1:p} = \hat{S}_n\mathcal{A}_{k+1:p}^2\Sigma_{k+1:p}^{\beta-1}\hat{S}_n^*$, the meaning of the footnote $k+1:p$ follows similar rule as the footnote $>k$, and let $\tilde{\Sigma} = \mathcal{A}^2\Sigma^\beta$, $\hat{\tilde{\Sigma}}_{>k} = \frac{1}{n}\mathcal{A}_{>k}\Sigma_{>k}^{\frac{\beta-1}{2}}\hat{S}_n^*\hat{S}_n\Sigma_{>k}^{\frac{\beta-1}{2}}\mathcal{A}_{>k} = \mathcal{A}_{>k}\Sigma_{>k}^{\frac{\beta-1}{2}}\hat{\Sigma}\Sigma_{>k}^{\frac{\beta-1}{2}}\mathcal{A}_{>k}$. Observe that $m_k = ||\hat{\tilde{\Sigma}}_{>k}||$, we would like to bound $||\hat{\tilde{\Sigma}}_{>k}||$ using the matrix Chernoff inequality with intrinsic dimension. Tropp (2015)[Theorem 7.2.1]. However, this inequality was proved for finite matrices, so we'll approximate the infinite matrix with finite ones. $m_k$ can be bounded as:

$$m_k = ||\frac{1}{n}\tilde{K}_{k+1:p'} + \frac{1}{n}\tilde{K}_{>p'}|| \le ||\frac{1}{n}\tilde{K}_{k+1:p'}|| + ||\frac{1}{n}\tilde{K}_{>p'}|| = ||\hat{\tilde{\Sigma}}_{k+1:p'}|| + m_{p'}.$$

Furthermore, $m_p'$ can be bounded as

$$m_{p'} \le \frac{1}{n}\operatorname{tr}(\tilde{K}_{>p'}) = \frac{1}{n}\sum_{j=1}^n\sum_{i>p'}p_i^2\lambda_i^\beta\psi_i(x_j)^2 \le \beta_{p'}\sum_{i>p'}p_i^2\lambda_i^\beta \le \beta_{p'}\operatorname{tr}(\tilde{\Sigma}_{>p'}).$$

If $p$ is finite, we can take $p = p'$ and $m_p' = 0$. Otherwise, $p$ is infinite, and $m_{p'} \le \beta_{p'}\operatorname{tr}(\Sigma_{>p'})$. By assumption 3.4:
$$\forall\epsilon > 0, \exists p' \in \mathbb{N} \text{ s.t. } m_{p'} < \epsilon.$$

We define $S_{k+1:p'}^j := \frac{1}{n}\mathcal{A}_{k+1:p'}\Sigma_{k+1:p'}^{\frac{\beta-1}{2}}\hat{S}^{j*}\hat{S}^j\Sigma_{k+1:p'}^{\frac{\beta-1}{2}}\mathcal{A}_{k+1:p'}$, where $\hat{S}^j f = \langle f, K_{x_j}\rangle_{\mathcal{H}}$ and $\hat{S}^{j*}\theta = \theta_j K_{x_j}$. Then we will have $\hat{\tilde{\Sigma}}_{k+1:p'} = \sum_{j=1}^n S_{k+1:p'}^j$. We need a bound on both $\mu_1(S_{k+1:p'}^j)$ and $\mu_1(\mathbb{E}\hat{\tilde{\Sigma}}_{k+1:p'})$. For the first,

$$\mu_1(S_{k+1:p'}^j) = \frac{1}{n}\sum_{i=k+1}^{p'}p_i^2\lambda_i^\beta\psi_i(x_j)^2 \le \frac{1}{n}\sum_{i=k+1}^\infty p_i^2\lambda_i^\beta\psi_i(x_j)^2 \le \frac{\beta_k}{n}\operatorname{tr}(\tilde{\Sigma}_{>k}).$$

Let $L := \frac{\beta_k}{n}\operatorname{tr}(\tilde{\Sigma}_{>k})$ denoting the RHS. For the second item, $\mathbb{E}\hat{\tilde{\Sigma}}_{k+1:p'} = \tilde{\Sigma}_{k+1:p'} = \operatorname{diag}(p_{k+1}^2\lambda_{k+1}^\beta, \ldots, p_{p'}^2\lambda_{p'}^\beta)$. Thus, $\mathbb{E}\hat{\tilde{\Sigma}}_{k+1:p'} = p_{k+1}^2\lambda_{k+1}^\beta$.

Now the conditions of Tropp (2015)[Theorem 7.2.1] are satisfied. So, for $r_{k:p'} := \frac{\operatorname{tr}(\tilde{\Sigma}_{k+1:p'})}{p_{k+1}^2\lambda_{k+1}^\beta}$ and any $t \ge 1 + \frac{L}{p_{k+1}^2\lambda_{k+1}^\beta} = 1 + \frac{\beta_k r_k}{n}$,

$$\mathbb{P}(||\hat{\tilde{\Sigma}}_{k+1:p'}|| \ge tp_{k+1}^2\lambda_{k+1}^\beta) \le 2r_{k:p'}\left(\frac{e^{t-1}}{t^t}\right)^{p_{k+1}^2\lambda_{k+1}^\beta/L}.$$

Using the fact that $p_{k+1}^2\lambda_{k+1}^\beta/L = n/\beta_k r_k$ and $e^{t-1} \le e^t$, $r_{k:p'} \le r_k$,

$$\mathbb{P}(m_k - m_{p'} \ge tp_{k+1}^2\lambda_{k+1}^\beta) \le \mathbb{P}(||\hat{\tilde{\Sigma}}_{k+1:p'}|| \ge tp_{k+1}^2\lambda_{k+1}^\beta) \le 2r_k\left(\frac{e}{t}\right)^{nt/\beta_k r_k}.$$

Now pick $t = e^3 + 2\frac{\beta_k r_k}{n}\ln(k+1)$, then

$$\mathbb{P}(m_k - m_{p'} \ge tp_{k+1}^2\lambda_{k+1}^\beta) \le 2\frac{r_k}{(k+1)^4}\exp\left(-2\frac{e^3}{\beta_k}\frac{n}{r_k}\right).$$

As a result, we obtain that for $c' = 2e^3$, $c = e^3$, the inequality holds w.p. at least $1 - 4\frac{r_k}{k^4}\exp(-\frac{c'}{\beta_k}\frac{n}{r_k})$ that

$$m_k \le c\left(p_{k+1}^2\lambda_{k+1}^\beta + \beta_k\log(k+1)\frac{\operatorname{tr}(\tilde{\Sigma}_{>k})}{n}\right) + m_{p'}.$$

As $p'$ tends to $\infty$ in some sequence determined by Assumption 1, $m_p'$ tends to 0. Therefore, we obtain the desired result. $\square$

In the following we present an important lemma for bounding largest and smallest eigenvalues of unregularized spectrally transformed matrix. This lemma would be useful to bound concentration coefficient $\rho_{k,n}$ in the interpolation case.

**Lemma C.6** (Bounds on $\mu_1(\frac{1}{n}\tilde{K}_{>k})$ and $\mu_n(\frac{1}{n}\tilde{K}_{>k'})$). *Suppose Assumption 3.4 holds, then there exists absolute constant $c, c' > 0$ s.t. it holds w.p. at least $1 - 4\frac{r_k}{k^4}\exp(-\frac{c'}{\beta_k}\frac{n}{r_k})$ that*

$$\mu_1(\frac{1}{n}\tilde{K}_{>k}) \leq c\left(p_{k+1}^2\lambda_{k+1}^\beta + \beta_k\log(k+1)\frac{\mathrm{tr}(\tilde{\Sigma}_{>k})}{n}\right).$$

*And for any $k' \in \mathbb{N}$ with $k' > k$, and any $\delta > 0$ it holds w.p. at least $1 - \delta - 4\frac{r_k}{k^4}\exp(-\frac{c'}{\beta_k}\frac{n}{r_k})$ that*

$$\alpha_{k'}\left(1 - \frac{1}{\delta}\sqrt{\frac{n^2}{\mathrm{tr}(\tilde{\Sigma}_{>k'})^2/\mathrm{tr}(\tilde{\Sigma}_{>k'}^2)}}\right) \leq \mu_n(\frac{1}{n}\tilde{K}_{>k'}),$$

*where $\tilde{\Sigma} := \mathcal{A}^2\Sigma^\beta$, $r_k := \frac{\mathrm{tr}(\tilde{\Sigma}_{>k})}{p_{k+1}^2\lambda_{k+1}^\beta}$.*

*Proof.* By Weyl's theorem Horn & Johnson (1985)[Corollary 4.3.15], for any $k' \geq k$ we have $\mu_n(\tilde{K}_{\geq k}) \geq \mu_n(\tilde{K}_{\geq k'}) + \mu_n(\tilde{K}_{k:k'}) \geq \mu_n(\tilde{K}_{\geq k'})$. So the lower bound comes from C.4(with $k'$) and the upper bound directly comes from C.5. □

# D  UPPER BOUND FOR THE VARIANCE

**Lemma D.1** (Upper bound for variance). *We define the variance of the noise be $\sigma_\varepsilon^2$ and evaluate variance in $\mathcal{H}^{\beta'}$ norm, If for some $k \in \mathbb{N}$, $\tilde{K}_{>k}^\gamma$ is positive-definite then*

$$V \leq \sigma_\varepsilon^2 \cdot \left[\frac{(\mu_1(\tilde{K}_{>k}^\gamma)^{-1})^2}{(\mu_n(\tilde{K}_{>k}^\gamma)^{-1})^2}\frac{\mathrm{tr}(\hat{S}_n\psi_{\leq k}^*\Lambda_{\mathcal{A}^{-2}\Sigma^{-\beta'}}^{\leq k}\psi_{\leq k}\hat{S}_n^*)}{\mu_k(\psi_{\leq k}\hat{S}_n^*\hat{S}_n\psi_{\leq k}^*)^2}\right.$$
$$\left. + (\mu_1(\tilde{K}_{>k}^\gamma)^{-1})^2\,\mathrm{tr}(\hat{S}_n\psi_{>k}^*\Lambda_{\mathcal{A}^2\Sigma^{-\beta'+2\beta}}^{>k}\psi_{>k}\hat{S}_n^*)\right].$$

*Proof.* Recall $V = \mathbb{E}_\varepsilon[\|\hat{f}(\varepsilon)\|_{\mathcal{H}^{\beta'}}^2]$, we can split the variance into $\|\hat{f}(\varepsilon)_{\leq k}\|_{\mathcal{H}^{\beta'}}^2$ and $\|\hat{f}(\varepsilon)_{>k}\|_{\mathcal{H}^{\beta'}}^2$ according to Lemma A.4. To bound these, by Lemma A.3 we could bound $\|\phi_{\leq k}\hat{f}(\varepsilon)_{\leq k}\|_{\Lambda_{\Sigma_{\leq k}^{1-\beta'}}}^2$,

$\|\phi_{>k}\hat{f}(\varepsilon)_{>k}\|_{\Lambda_{\Sigma_{>k}^{1-\beta'}}}^2$ respectively using matrix inequalities.

First we handle $\|\phi_{\leq k}\hat{f}(\varepsilon)_{\leq k}\|_{\Lambda_{\Sigma_{\leq k}^{1-\beta'}}}^2$, using Lemma A.2 while substituting $y$ with $\varepsilon$, we have

$$\phi_{\leq k}\hat{f}(\varepsilon)_{\leq k} + \phi_{\leq k}\mathcal{A}_{\leq k}\Sigma_{\leq k}^{\beta-1}\hat{S}_n^*(\tilde{K}_{>k}^\gamma)^{-1}\hat{S}_n\mathcal{A}_{\leq k}\hat{f}(\varepsilon)_{\leq k} = \phi_{\leq k}\mathcal{A}_{\leq k}\Sigma_{\leq k}^{\beta-1}\hat{S}_n^*(\tilde{K}_{>k}^\gamma)^{-1}\varepsilon.$$

We multiply by $(\phi_{\leq k}\hat{f}(\varepsilon)_{\leq k})^T\Lambda_{\mathcal{A}^{-2}\Sigma^{-\beta+(1-\beta')}}^{\leq k} \in \mathbb{R}^{1\times k}$, on two sides respectively (note that the motivation of multiplying an additional diagonal matrix term here is to make the $\mu_k$ term only have $\mu_k(\psi_{\leq k}\hat{S}_n^*\hat{S}_n\psi_{\leq k}^*)$), and this would not affect the polynomial bound.

Then since $\|\phi_{\leq k}\hat{f}(\varepsilon)_{\leq k}\|_{\Lambda_{\mathcal{A}^{-2}\Sigma^{-\beta+(1-\beta')}}^{\leq k}}^2 \geq 0$, we have

$$\underbrace{(\phi_{\leq k}\hat{f}(\varepsilon)_{\leq k})^T\Lambda_{\mathcal{A}^{-2}\Sigma^{-\beta+(1-\beta')}}^{\leq k}\phi_{\leq k}\mathcal{A}_{\leq k}\Sigma_{\leq k}^{\beta-1}\hat{S}_n^*(\tilde{K}_{>k}^\gamma)^{-1}\hat{S}_n\mathcal{A}_{\leq k}\hat{f}(\varepsilon)_{\leq k}}_{\text{Quadratic term w.r.t. } \phi_{\leq k}\hat{f}(\varepsilon)_{\leq k}}$$
$$\leq \underbrace{(\phi_{\leq k}\hat{f}(\varepsilon)_{\leq k})^T\Lambda_{\mathcal{A}^{-2}\Sigma^{-\beta+(1-\beta')}}^{\leq k}\phi_{\leq k}\mathcal{A}_{\leq k}\Sigma_{\leq k}^{\beta-1}\hat{S}_n^*(\tilde{K}_{>k}^\gamma)^{-1}\varepsilon}_{\text{Linear term w.r.t. } \phi_{\leq k}\hat{f}(\varepsilon)_{\leq k}}.$$

Then we lower bound the quadratic term and upper bound the linear term respectively, first we lower bound the quadratic term:

$$(\phi_{\leq k}\hat{f}(\varepsilon)_{\leq k})^T \Lambda^{\leq k}_{\mathcal{A}^{-2}\Sigma^{-\beta+(1-\beta')}} \phi_{\leq k}\mathcal{A}_{\leq k}\Sigma^{\beta-1}_{\leq k}\hat{S}^*_n(\tilde{K}^\gamma_{>k})^{-1}\hat{S}_n\mathcal{A}_{\leq k}\hat{f}(\varepsilon)_{\leq k}$$

Diagonalize the operators,

$$= (\phi_{\leq k}\hat{f}(\varepsilon)_{\leq k})^T \Lambda^{\leq k}_{\mathcal{A}^{-2}\Sigma^{-\beta+(1-\beta')}} \phi_{\leq k}(\phi^*_{\leq k}\Lambda^{\leq k}_{\mathcal{A}\Sigma^{\beta-1}}\phi_{\leq k})\hat{S}^*_n(\tilde{K}^\gamma_{>k})^{-1}\hat{S}_n(\phi^*_{\leq k}\Lambda^{\leq k}_{\mathcal{A}}\phi_{\leq k})\hat{f}(\varepsilon)_{\leq k}$$

$$= (\phi_{\leq k}\hat{f}(\varepsilon)_{\leq k})^T \Lambda^{\leq k}_{\mathcal{A}^{-1}\Sigma^{-\beta'}} \phi_{\leq k}\hat{S}^*_n(\tilde{K}^\gamma_{>k})^{-1}\hat{S}_n\phi^*_{\leq k}\Lambda^{\leq k}_{\mathcal{A}}(\phi_{\leq k}\hat{f}(\varepsilon)_{\leq k})\ (\phi_{\leq k}\phi^*_{\leq k} = id_{\leq k})$$

By $\phi_{\leq k} = \Lambda^{\leq k}_{\Sigma^{1/2}}\psi_{\leq k}$ and $\phi^*_{\leq k} = \psi^*_{\leq k}\Lambda^{\leq k}_{\Sigma^{1/2}}$,

$$= \underbrace{(\phi_{\leq k}\hat{f}(\varepsilon)_{\leq k})^T}_{1\times k} \underbrace{\Lambda^{\leq k}_{\mathcal{A}^{-1}\Sigma^{1/2-\beta'}}}_{k\times k} \underbrace{\psi_{\leq k}\hat{S}^*_n}_{k\times n} \underbrace{(\tilde{K}^\gamma_{>k})^{-1}}_{n\times n} \underbrace{\hat{S}_n\psi^*_{\leq k}}_{n\times k} \underbrace{\Lambda^{\leq k}_{\mathcal{A}\Sigma^{1/2}}(\phi_{\leq k}\hat{f}(\varepsilon)_{\leq k})}_{k\times 1}$$

$$\geq \mu_n((\tilde{K}^\gamma_{>k})^{-1})\ \mu_k(\psi_{\leq k}\hat{S}^*_n\hat{S}_n\psi^*_{\leq k})\ (\phi_{\leq k}\hat{f}(\varepsilon)_{\leq k})^T\Lambda^{\leq k}_{\Sigma^{1-\beta'}}(\phi_{\leq k}\hat{f}(\varepsilon)_{\leq k}).$$

The last inequality is because $\mu_k(AB) = \mu_k(BA)$ for $k \times k$ matrix $A, B$ by (Horn & Johnson, 1985, Theorem 1.3.20).

We continue to derive the bound

$$\mu_n((\tilde{K}^\gamma_{>k})^{-1})\ \mu_k(\psi_{\leq k}\hat{S}^*_n\hat{S}_n\psi^*_{\leq k})\ (\phi_{\leq k}\hat{f}(\varepsilon)_{\leq k})^T\Lambda^{\leq k}_{\Sigma^{1-\beta'}}(\phi_{\leq k}\hat{f}(\varepsilon)_{\leq k})$$

$$= \|\phi_{\leq k}\hat{f}(\varepsilon)_{\leq k}\|^2_{\Lambda^{\leq k}_{\Sigma^{1-\beta'}}}\ \mu_n((\tilde{K}^\gamma_{>k})^{-1})\ \mu_k(\psi_{\leq k}\hat{S}^*_n\hat{S}_n\psi^*_{\leq k})$$

$$= \|\hat{f}(\varepsilon)_{\leq k}\|^2_{\mathcal{H}^{\beta'}}\ \mu_n((\tilde{K}^\gamma_{>k})^{-1})\ \mu_k(\psi_{\leq k}\hat{S}^*_n\hat{S}_n\psi^*_{\leq k}).$$

This finishes lower bound of the quadratic term, we continue to upper bound the linear term

$$(\phi_{\leq k}\hat{f}(\varepsilon)_{\leq k})^T \Lambda^{\leq k}_{\mathcal{A}^{-2}\Sigma^{-\beta+(1-\beta')}} \phi_{\leq k}\mathcal{A}_{\leq k}\Sigma^{\beta-1}_{\leq k}\hat{S}^*_n(\tilde{K}^\gamma_{>k})^{-1}\varepsilon$$

$$= (\phi_{\leq k}\hat{f}(\varepsilon)_{\leq k})^T \Lambda^{\leq k}_{\mathcal{A}^{-2}\Sigma^{-\beta+(1-\beta')}} \phi_{\leq k}\phi^*_{\leq k}\Lambda^{\leq k}_{\mathcal{A}\Sigma^{\beta-1}}\phi_{\leq k}\hat{S}^*_n(\tilde{K}^\gamma_{>k})^{-1}\varepsilon$$

$$= \underbrace{(\phi_{\leq k}\hat{f}(\varepsilon)_{\leq k})^T \Lambda^{\leq k}_{\mathcal{A}^{-1}\Sigma^{1/2-\beta'}}}_{1\times k} \underbrace{\psi_{\leq k}\hat{S}^*_n(\tilde{K}^\gamma_{>k})^{-1}\varepsilon}_{k\times 1}\ (\text{By } \phi_{\leq k}\phi^*_{\leq k} = id_{\leq k} \text{ and } \phi_{\leq k} = \Lambda^{\leq k}_{\Sigma^{1/2}}\psi_{\leq k})$$

$$= \underbrace{(\phi_{\leq k}\hat{f}(\varepsilon)_{\leq k})^T \Lambda^{\leq k}_{\Sigma^{(1-\beta')/2}}}_{1\times k} \underbrace{\Lambda^{\leq k}_{\mathcal{A}^{-1}\Sigma^{-\beta'/2}}\psi_{\leq k}\hat{S}^*_n(\tilde{K}^\gamma_{>k})^{-1}\varepsilon}_{k\times 1}$$

$$\leq \|\phi_{\leq k}\hat{f}(\varepsilon)_{\leq k}\|_{\Lambda^{\leq k}_{\Sigma^{1-\beta'}}}\|\Lambda^{\leq k}_{\mathcal{A}^{-1}\Sigma^{-\beta'/2}}\psi_{\leq k}\hat{S}^*_n(\tilde{K}^\gamma_{>k})^{-1}\varepsilon\|$$

$$= \|\hat{f}(\varepsilon)_{\leq k}\|_{\mathcal{H}^{\beta'}}\|\Lambda^{\leq k}_{\mathcal{A}^{-1}\Sigma^{-\beta'/2}}\psi_{\leq k}\hat{S}^*_n(\tilde{K}^\gamma_{>k})^{-1}\varepsilon\|.$$

Therefore, we obtain

$$\|\hat{f}(\varepsilon)_{\leq k}\|^2_{\mathcal{H}^{\beta'}}\ \mu_n((\tilde{K}^\gamma_{>k})^{-1})\ \mu_k(\psi_{\leq k}\hat{S}^*_n\hat{S}_n\psi^*_{\leq k}) \leq \|\hat{f}(\varepsilon)_{\leq k}\|_{\mathcal{H}^{\beta'}}\|\Lambda^{\leq k}_{\mathcal{A}^{-1}\Sigma^{-\beta'/2}}\psi_{\leq k}\hat{S}^*_n(\tilde{K}^\gamma_{>k})^{-1}\varepsilon\|.$$

Therefore,

$$\|\hat{f}(\varepsilon)_{\leq k}\|^2_{\mathcal{H}^{\beta'}} \leq \frac{\varepsilon^T(\tilde{K}^\gamma_{>k})^{-1}\hat{S}_n\psi^*_{\leq k}\Lambda^{\leq k}_{\mathcal{A}^{-2}\Sigma^{-\beta'}}\psi_{\leq k}\hat{S}^*_n(\tilde{K}^\gamma_{>k})^{-1}\varepsilon}{\mu_n((\tilde{K}^\gamma_{>k})^{-1})^2\ \mu_k(\psi_{\leq k}\hat{S}^*_n\hat{S}_n\psi^*_{\leq k})^2}.$$

Then we take expectation w.r.t $\varepsilon$ we have

$$\mathbb{E}_\varepsilon\|\hat{f}(\varepsilon)_{\leq k}\|^2_{\mathcal{H}^{\beta'}} \leq \sigma^2_\varepsilon \cdot \frac{\text{tr}(\overbrace{(\tilde{K}^\gamma_{>k})^{-1}}^{n\times n}\hat{S}_n\psi^*_{\leq k}\overbrace{\Lambda^{\leq k}_{\mathcal{A}^{-2}\Sigma^{-\beta'}}}^{n\times n}\psi_{\leq k}\hat{S}^*_n\overbrace{(\tilde{K}^\gamma_{>k})^{-1}}^{n\times n})}{\mu_n((\tilde{K}^\gamma_{>k})^{-1})^2\ \mu_k(\psi_{\leq k}\hat{S}^*_n\hat{S}_n\psi^*_{\leq k})^2}$$

$$\leq \sigma^2_\varepsilon \cdot \frac{\mu_1((\tilde{K}^\gamma_{>k})^{-1})^2}{\mu_n((\tilde{K}^\gamma_{>k})^{-1})^2} \frac{\text{tr}(\overbrace{\hat{S}_n\psi^*_{\leq k}\Lambda^{\leq k}_{\mathcal{A}^{-2}\Sigma^{-\beta'}}\psi_{\leq k}\hat{S}^*_n}^{n\times n})}{\mu_k(\underbrace{\psi_{\leq k}\hat{S}^*_n\hat{S}_n\psi^*_{\leq k}}_{k\times k})^2},$$

where the last inequality is by using the fact that $\text{tr}(MM'M) \leq \mu_1(M)^2\,\text{tr}(M')$ for positive-definite matrix $M, M'$.

Now we move on to bound the $> k$ components $\|\phi_{>k}\hat{f}(\varepsilon)_{>k}\|^2_{\Lambda^{>k}_{\Sigma^{1-\beta'}}}$

$$\|\phi_{>k}\hat{f}(\varepsilon)_{>k}\|^2_{\Lambda^{>k}_{\Sigma^{1-\beta'}}}$$
$$= \|\phi_{>k}\mathcal{A}_{>k}\Sigma^{\beta-1}_{>k}\hat{S}^*_n(\tilde{K}^\gamma)^{-1}\varepsilon\|^2_{\Lambda^{>k}_{\Sigma^{1-\beta'}}}$$
$$= \varepsilon^T(\tilde{K}^\gamma)^{-1}\hat{S}_n\Sigma^{\beta-1}_{>k}\mathcal{A}_{>k}\phi^*_{>k}\Lambda^{>k}_{\Sigma^{1-\beta'}}\phi_{>k}\mathcal{A}_{>k}\Sigma^{\beta-1}_{>k}\hat{S}^*_n(\tilde{K}^\gamma)^{-1}\varepsilon$$
$$= \varepsilon^T(\tilde{K}^\gamma)^{-1}\hat{S}_n\phi^*_{>k}\Lambda^{>k}_{\mathcal{A}^2\Sigma^{(-\beta'+2\beta-1)}}\phi_{>k}\hat{S}^*_n(\tilde{K}^\gamma)^{-1}\varepsilon \text{ (By } 2(\beta-1)+(1-\beta')=-\beta'+2\beta-1).$$

We take expectation over $\varepsilon$

$$\mathbb{E}_\varepsilon\|\phi_{>k}\hat{f}(\varepsilon)_{>k}\|^2_{\Lambda_{\mathcal{A}^2\Sigma^\beta}} \leq \sigma^2_\varepsilon\mu_1((\tilde{K}^\gamma)^{-1})^2\,\mathrm{tr}(\hat{S}_n\phi^*_{>k}\Lambda^{>k}_{\mathcal{A}^2\Sigma^{(-\beta'+2\beta-1)}}\phi_{>k}\hat{S}^*_n)$$
$$\leq \sigma^2_\varepsilon\mu_1((\tilde{K}^\gamma_{>k})^{-1})^2\,\mathrm{tr}(\underbrace{\hat{S}_n\phi^*_{>k}\Lambda^{>k}_{\mathcal{A}^2\Sigma^{(-\beta'+2\beta-1)}}\phi_{>k}\hat{S}^*_n}_{n\times n})$$
$$= \sigma^2_\varepsilon\mu_1((\tilde{K}^\gamma_{>k})^{-1})^2\,\mathrm{tr}(\underbrace{\hat{S}_n\psi^*_{>k}\Lambda^{>k}_{\mathcal{A}^2\Sigma^{(-\beta'+2\beta)}}\psi_{>k}\hat{S}^*_n}_{n\times n}),$$

where the second last inequality is still using the fact that $\mathrm{tr}(MM'M) \leq \mu_1(M)^2\,\mathrm{tr}(M')$ for positive-definite matrix $M, M'$, and the last inequality is using $\tilde{K}^\gamma \succeq \tilde{K}^\gamma_{>k}$ to infer $\mu_1((\tilde{K}^\gamma)^{-1}) \leq \mu_1((\tilde{K}^\gamma_{>k})^{-1})$. $\qquad\square$

**Theorem D.2** (Bound on Variance with concentration coefficient). *Following previous Theorem D.1's assumptions, we can express the bound of variance using concentration coefficient* $\rho_{n,k}$

$$V \leq \sigma^2_\varepsilon\rho^2_{k,n}\cdot\Big(\frac{\mathrm{tr}(\hat{S}_n\psi^*_{\leq k}\Lambda^{\leq k}_{\mathcal{A}^{-2}\Sigma^{-\beta'}}\psi_{\leq k}\hat{S}^*_n)}{\mu_k(\psi_{\leq k}\hat{S}^*_n\hat{S}_n\psi^*_{\leq k})^2} + \frac{\overbrace{\mathrm{tr}(\hat{S}_n\psi^*_{>k}\Lambda^{>k}_{\mathcal{A}^2\Sigma^{-\beta'+2\beta}}\psi_{>k}\hat{S}^*_n)}^{\text{effective rank}}}{n^2\|\tilde{\Sigma}_{>k}\|^2}\Big).$$

*Proof.* By D.1 we have

$$V \leq \sigma^2_\varepsilon\cdot\Big(\frac{(\mu_1(\tilde{K}^\gamma_{>k})^{-1})^2}{(\mu_n(\tilde{K}^\gamma_{>k})^{-1})^2}\frac{\mathrm{tr}(\hat{S}_n\psi^*_{\leq k}\Lambda^{\leq k}_{\mathcal{A}^{-2}\Sigma^{-\beta'}}\psi_{\leq k}\hat{S}^*_n)}{\mu_k(\psi_{\leq k}\hat{S}^*_n\hat{S}_n\psi^*_{\leq k})^2}$$
$$+ (\mu_1(\tilde{K}^\gamma_{>k})^{-1})^2\,\mathrm{tr}(\hat{S}_n\psi^*_{>k}\Lambda^{>k}_{\mathcal{A}^2\Sigma^{-\beta'+2\beta}}\psi_{>k}\hat{S}^*_n)\Big).$$

Then by $\mu_1(\tilde{K}^\gamma_{>k})^{-1} = \dfrac{1}{n\mu_n(\frac{1}{n}\tilde{K}^\gamma_{>k})}, \mu_n(\tilde{K}^\gamma_{>k})^{-1} = \dfrac{1}{n\mu_1(\frac{1}{n}\tilde{K}^\gamma_{>k})}$, we have

$$\frac{(\mu_1(\tilde{K}^\gamma_{>k})^{-1})^2}{(\mu_n(\tilde{K}^\gamma_{>k})^{-1})^2} = \frac{\mu_1(\tilde{K}^\gamma_{>k})^2}{\mu_n(\tilde{K}^\gamma_{>k})^2} \leq \frac{(\mu_1(\tilde{K}_{>k})+\gamma)^2}{(\mu_n(\tilde{K}_{>k})+\gamma)^2} \leq \rho^2_{k,n}.$$

And

$$(\mu_1(\tilde{K}^\gamma_{>k})^{-1})^2$$
$$\leq \frac{1}{n^2}\frac{1}{\mu_n(\frac{1}{n}\tilde{K}^\gamma_{>k})^2}$$
$$= \frac{1}{n^2}\frac{\|\tilde{\Sigma}_{>k}\|^2}{\mu_n(\frac{1}{n}\tilde{K}^\gamma_{>k})^2}\frac{1}{\|\tilde{\Sigma}_{>k}\|^2}$$
$$\leq \frac{\rho^2_{k,n}}{n^2}\frac{1}{\|\tilde{\Sigma}_{>k}\|^2}.$$

Therefore,

$$
\begin{aligned}
V &\le \sigma_\varepsilon^2 \cdot \Big( \frac{(\mu_1(\tilde{K}_{>k}^\gamma)^{-1})^2}{(\mu_n(\tilde{K}_{>k}^\gamma)^{-1})^2} \frac{\mathrm{tr}(\hat{S}_n \psi_{\le k}^* \Lambda_{\mathcal{A}^{-2}\Sigma^{-\beta'}}^{\le k} \psi_{\le k} \hat{S}_n^*)}{\mu_k(\psi_{\le k} \hat{S}_n^* \hat{S}_n \psi_{\le k}^*)^2} \\
&\quad + (\mu_1(\tilde{K}_{>k}^\gamma)^{-1})^2 \, \mathrm{tr}(\hat{S}_n \psi_{>k}^* \Lambda_{\mathcal{A}^2\Sigma^{-\beta'+2\beta}}^{>k} \psi_{>k} \hat{S}_n^*) \Big) \\
&\le \sigma_\varepsilon^2 \cdot \Big( \rho_{k,n}^2 \frac{\mathrm{tr}(\hat{S}_n \psi_{\le k}^* \Lambda_{\mathcal{A}^{-2}\Sigma^{-\beta'}}^{\le k} \psi_{\le k} \hat{S}_n^*)}{\mu_k(\psi_{\le k} \hat{S}_n^* \hat{S}_n \psi_{\le k}^*)^2} \\
&\quad + \frac{\rho_{k,n}^2}{n^2} \frac{1}{\|\tilde{\Sigma}_{>k}\|^2} \, \mathrm{tr}(\hat{S}_n \psi_{>k}^* \Lambda_{\mathcal{A}^2\Sigma^{-\beta'+2\beta}}^{>k} \psi_{>k} \hat{S}_n^*) \Big) \\
&\le \sigma_\varepsilon^2 \rho_{k,n}^2 \cdot \Big( \frac{\mathrm{tr}(\hat{S}_n \psi_{\le k}^* \Lambda_{\mathcal{A}^{-2}\Sigma^{-\beta'}}^{\le k} \psi_{\le k} \hat{S}_n^*)}{\mu_k(\psi_{\le k} \hat{S}_n^* \hat{S}_n \psi_{\le k}^*)^2} + \frac{\overbrace{\mathrm{tr}(\hat{S}_n \psi_{>k}^* \Lambda_{\mathcal{A}^2\Sigma^{-\beta'+2\beta}}^{>k} \psi_{>k} \hat{S}_n^*)}^{\text{effective rank}}}{n^2 \|\tilde{\Sigma}_{>k}\|^2} \Big).
\end{aligned}
$$

$\square$

**Lemma D.3** (Simplified Upper bound for variance using concentration). *There exists some absolute constant $c, c', C_1 > 0$ s.t. for any $k \in \mathbb{N}$ with $c\beta_k k \log(k) \le n$, it holds w.p. at least $1 - 8\exp(\frac{-c'}{\beta_k^2} \frac{n}{k})$, the variance can be upper bounded as:*

$$
V \le C_1 \sigma_\varepsilon^2 \rho_{k,n}^2 \Big( \frac{\sum_{i \le k} p_i^{-2} \lambda_i^{-\beta'}}{n} + \frac{\sum_{i>k} p_i^2 \lambda_i^{-\beta'+2\beta}}{n\|\tilde{\Sigma}_{>k}\|^2} \Big).
$$

*Proof.* By Theorem D.2, we have

$$
V \le \sigma_\varepsilon^2 \rho_{k,n}^2 \cdot \Big( \frac{\mathrm{tr}(\hat{S}_n \psi_{\le k}^* \Lambda_{\mathcal{A}^{-2}\Sigma^{-\beta'}}^{\le k} \psi_{\le k} \hat{S}_n^*)}{\mu_k(\psi_{\le k} \hat{S}_n^* \hat{S}_n \psi_{\le k}^*)^2} + \frac{\overbrace{\mathrm{tr}(\hat{S}_n \psi_{>k}^* \Lambda_{\mathcal{A}^2\Sigma^{-\beta'+2\beta}}^{>k} \psi_{>k} \hat{S}_n^*)}^{\text{effective rank}}}{n^2 \|\tilde{\Sigma}_{>k}\|^2} \Big).
$$

Then we can apply concentration inequalities, by Lemma B.3, it holds w.p. at least $1 - 8\exp(\frac{-c'}{\beta_k^2} \frac{n}{k})$ that

$$
\begin{aligned}
V &\le \sigma_\varepsilon^2 \rho_{k,n}^2 \cdot \Big( \frac{c_2 n \sum_{i \le k} p_i^{-2} \lambda_i^{-\beta'}}{c_1^2 n^2} + \frac{c_2 n \sum_{i>k} p_i^2 \lambda_i^{-\beta'+2\beta}}{n^2 \|\tilde{\Sigma}_{>k}\|^2} \Big) \\
&\le \sigma_\varepsilon^2 \rho_{k,n}^2 \max\{\frac{c_2}{c_1^2}, c_2\} \Big( \frac{\sum_{i \le k} p_i^{-2} \lambda_i^{-\beta'}}{n} + \frac{\sum_{i>k} p_i^2 \lambda_i^{-\beta'+2\beta}}{n\|\tilde{\Sigma}_{>k}\|^2} \Big).
\end{aligned}
$$

Then we take $C_1$ to be $\max\{\frac{c_2}{c_1^2}, c_2\}$ to obtain the desired bound. $\square$

# E  UPPER BOUND FOR THE BIAS

**Lemma E.1** (Upper bound for bias). *Suppose that for some $k < n$, the matrix $\tilde{K}_{>k}^\gamma$ is positive-definite, then*

$$
\begin{aligned}
B &\le 3\Big( \frac{\mu_1((\tilde{K}_{>k}^\gamma)^{-1})^2}{\mu_n((\tilde{K}_{>k}^\gamma)^{-1})^2} \frac{\mu_1(\psi_{\le k} \hat{S}_n^* \hat{S}_n \psi_{\le k}^*)}{\mu_k(\psi_{\le k} \hat{S}_n^* \hat{S}_n \psi_{\le k}^*)^2 \mu_k(\Lambda_{\mathcal{A}^2\Sigma^{\beta'}}^{\le k})} \|\hat{S}_n \mathcal{A}_{>k} f_{>k}^*\|^2 \\
&\quad + \frac{\|\phi_{\le k} f_{\le k}^*\|_{\Lambda_{\mathcal{A}^{-2}\Sigma^{1-2\beta}}^{\le k}}^2}{\mu_n((\tilde{K}_{>k}^\gamma)^{-1})^2 \mu_k(\psi_{\le k} \hat{S}_n^* \hat{S}_n \psi_{\le k}^*)^2 \mu_k(\Lambda_{\mathcal{A}^2\Sigma^{\beta'}}^{\le k})} \\
&\quad + \|\phi_{>k} f_{>k}^*\|_{\Lambda_{\Sigma^{1-\beta'}}^{>k}}^2 \\
&\quad + \|\Lambda_{\Sigma^{1-\beta'}}^{>k}\| \, \mu_1[(\tilde{K}_{>k}^\gamma)^{-1}]^2 \|\hat{S}_n \mathcal{A}_{>k} f_{>k}^*\|^2 \mu_1(\underbrace{\hat{S}_n \psi_{>k}^* \Lambda_{\mathcal{A}^2\Sigma^{2\beta-1}}^{>k} \psi_{>k} \hat{S}_n^*}_{n \times n}) \\
&\quad + \|\Lambda_{\Sigma^{-\beta'+\beta}}^{>k}\| \frac{\mu_1((\tilde{K}_{>k}^\gamma)^{-1})}{\mu_n((\tilde{K}_{>k}^\gamma)^{-1})^2} \frac{\mu_1(\psi_{\le k} \hat{S}_n^* \hat{S}_n \psi_{\le k}^*)}{\mu_k(\psi_{\le k} \hat{S}_n^* \hat{S}_n \psi_{\le k}^*)^2} \|\phi_{\le k} f_{\le k}^*\|_{\Lambda_{\mathcal{A}^{-2}\Sigma^{1-2\beta}}^{\le k}} \Big).
\end{aligned}
$$

*Proof.* Similar as variance, by lemma A.4 we can bound $\leq k$ and $> k$ separately, for brevity we define the error vector $\xi := \phi(\hat{f}(\hat{S}_n \mathcal{A} f^*) - f^*) \in \ell_2^\infty$, by lemma A.3 we can bound $\|\xi_{\leq k}\|_{\Sigma^{1-\beta'}}$ and $\|\xi_{>k}\|_{\Sigma^{1-\beta'}}$ separately.

We first discuss $\|\xi_{\leq k}\|_{\Sigma^{1-\beta'}}$, by lemma A.2, we have

$$\phi_{\leq k}\hat{f}(\hat{S}_n \mathcal{A} f^*) + \phi_{\leq k}\mathcal{A}_{\leq k}\Sigma_{\leq k}^{\beta-1}\hat{S}_n^*(\tilde{K}_{>k}^\gamma)^{-1}\hat{S}_n\mathcal{A}\hat{f}(\hat{S}_n \mathcal{A} f^*)_{\leq k} = \phi_{\leq k}\mathcal{A}_{\leq k}\Sigma_{\leq k}^{\beta-1}\hat{S}_n^*(\tilde{K}_{>k}^\gamma)^{-1}\hat{S}_n\mathcal{A}f^*. \tag{7}$$

By definition of $\xi$, we have $\xi_{\leq k} = \phi_{\leq k}(\hat{f} - f^*) = \phi_{\leq k}\hat{f}_{\leq k} - \phi_{\leq k}f^*_{\leq k}$, so we have $\phi_{\leq k}\hat{f} = \xi_{\leq k} + \phi_{\leq k}f^*_{\leq k}$.

$$\begin{aligned}
\text{LHS of (7)} &= \xi_{\leq k} + \phi_{\leq k}f^*_{\leq k} + \phi_{\leq k}\mathcal{A}_{\leq k}\Sigma_{\leq k}^{\beta-1}\hat{S}_n^*(\tilde{K}_{>k}^\gamma)^{-1}\hat{S}_n\phi_{\leq k}^*\Lambda_{\mathcal{A}}^{\leq k}(\xi_{\leq k} + \phi_{\leq k}f^*_{\leq k}) \\
&= \xi_{\leq k} + \phi_{\leq k}f^*_{\leq k} + \phi_{\leq k}\mathcal{A}_{\leq k}\Sigma_{\leq k}^{\beta-1}\hat{S}_n^*(\tilde{K}_{>k}^\gamma)^{-1}\hat{S}_n\phi_{\leq k}^*\Lambda_{\mathcal{A}}^{\leq k}\xi_{\leq k} \\
&\quad + \underbrace{\phi_{\leq k}\mathcal{A}_{\leq k}\Sigma_{\leq k}^{\beta-1}\hat{S}_n^*(\tilde{K}_{>k}^\gamma)^{-1}\hat{S}_n\phi_{\leq k}^*\Lambda_{\mathcal{A}}^{\leq k}\phi_{\leq k}f^*_{\leq k}}_{(*)} .
\end{aligned}$$

And

$$\begin{aligned}
\text{RHS of (7)} &= \phi_{\leq k}\mathcal{A}_{\leq k}\Sigma_{\leq k}^{\beta-1}\hat{S}_n^*(\tilde{K}_{>k}^\gamma)^{-1}\hat{S}_n(\phi_{\leq k}^*\Lambda_{\mathcal{A}}^{\leq k}\phi_{\leq k}f^*_{\leq k} + \phi_{>k}^*\Lambda_{\mathcal{A}}^{>k}\phi_{>k}f^*_{>k}) \\
&= \underbrace{\phi_{\leq k}\mathcal{A}_{\leq k}\Sigma_{\leq k}^{\beta-1}\hat{S}_n^*(\tilde{K}_{>k}^\gamma)^{-1}\hat{S}_n\phi_{\leq k}^*\Lambda_{\mathcal{A}}^{\leq k}\phi_{\leq k}f^*_{\leq k}}_{(*)} \\
&\quad + \phi_{\leq k}\mathcal{A}_{\leq k}\Sigma_{\leq k}^{\beta-1}\hat{S}_n^*(\tilde{K}_{>k}^\gamma)^{-1}\hat{S}_n\phi_{>k}^*\Lambda_{\mathcal{A}}^{>k}\phi_{>k}f^*_{>k}.
\end{aligned}$$

The two (*) terms get cancelled out, therefore

$$\begin{aligned}
\xi_{\leq k} &+ \phi_{\leq k}\mathcal{A}_{\leq k}\Sigma_{\leq k}^{\beta-1}\hat{S}_n^*(\tilde{K}_{>k}^\gamma)^{-1}\hat{S}_n\phi_{\leq k}^*\Lambda_{\mathcal{A}}^{\leq k}\xi_{\leq k} \\
&= \phi_{\leq k}\mathcal{A}_{\leq k}\Sigma_{\leq k}^{\beta-1}\hat{S}_n^*(\tilde{K}_{>k}^\gamma)^{-1}\hat{S}_n\phi_{>k}^*\Lambda_{\mathcal{A}}^{>k}\phi_{>k}f^*_{>k} - \phi_{\leq k}f^*_{\leq k}.
\end{aligned}$$

We multiply $\xi_{\leq k}^T\Lambda_{\mathcal{A}^{-1}\Sigma^{1-\beta-\beta'}/2}^{\leq k}$ in both sides and since $\|\xi_{\leq k}\|_{\Lambda_{\mathcal{A}^{-1}\Sigma^{1-\beta-\beta'}/2}^{\leq k}}^2 \geq 0$,

$$\begin{aligned}
&\xi_{\leq k}^T\Lambda_{\mathcal{A}^{-1}\Sigma^{1-\beta-\beta'}/2}^{\leq k}\phi_{\leq k}\mathcal{A}_{\leq k}\Sigma_{\leq k}^{\beta-1}\hat{S}_n^*(\tilde{K}_{>k}^\gamma)^{-1}\hat{S}_n\phi_{\leq k}^*\Lambda_{\mathcal{A}}^{\leq k}\xi_{\leq k} \\
&\leq \xi_{\leq k}^T\Lambda_{\mathcal{A}^{-1}\Sigma^{1-\beta-\beta'}/2}^{\leq k}\phi_{\leq k}\mathcal{A}_{\leq k}\Sigma_{\leq k}^{\beta-1}\hat{S}_n^*(\tilde{K}_{>k}^\gamma)^{-1}\hat{S}_n\phi_{>k}^*\Lambda_{\mathcal{A}}^{>k}\phi_{>k}f^*_{>k} - \xi_{\leq k}^T\Lambda_{\mathcal{A}^{-1}\Sigma^{1-\beta-\beta'}/2}^{\leq k}\phi_{\leq k}f^*_{\leq k}.
\end{aligned}$$

LHS is the quadratic term w.r.t. $\xi_{\leq k}$ and RHS is the linear term w.r.t. $\xi_{\leq k}$, similar to Variance case, we lower bound LHS and upper bound RHS respectively.

$$\begin{aligned}
\text{LHS} &= \overbrace{\xi_{\leq k}^T}^{1\times k}\overbrace{\Lambda_{\Sigma^{-\beta'}/2}^{\leq k}}^{k\times k}\overbrace{\phi_{\leq k}\hat{S}_n^*}^{k\times n}\overbrace{(\tilde{K}_{>k}^\gamma)^{-1}}^{n\times n}\overbrace{\hat{S}_n\phi_{\leq k}^*}^{n\times k}\overbrace{\Lambda_{\mathcal{A}}^{\leq k}}^{k\times k}\overbrace{\xi_{\leq k}}^{k\times 1} \\
&= \xi_{\leq k}^T\Lambda_{\Sigma^{(1-\beta')/2}}^{\leq k}\psi_{\leq k}\hat{S}_n^*(\tilde{K}_{>k}^\gamma)^{-1}\hat{S}_n\psi_{\leq k}^*\Lambda_{\mathcal{A}\Sigma^{1/2}}^{\leq k}\xi_{\leq k}.
\end{aligned}$$

Since $(1-\beta') + \beta'/2 = (1-\beta')/2 + 1/2$, it can be lower bounded by

$$\begin{aligned}
&\mu_n((\tilde{K}_{>k}^\gamma)^{-1})\,(\xi_{\leq k}^T\Lambda_{\Sigma^{1-\beta'}}^{\leq k}\xi_{\leq k})\mu_k\left(\psi_{\leq k}\hat{S}_n^*\hat{S}_n\psi_{\leq k}^*\right)\mu_k\left(\Lambda_{\mathcal{A}\Sigma^{\beta'/2}}^{\leq k}\right) \\
&= \|\xi_{\leq k}\|_{\Lambda_{\Sigma^{1-\beta'}}^{\leq k}}^2\mu_n((\tilde{K}_{>k}^\gamma)^{-1})\mu_k\left(\psi_{\leq k}\hat{S}_n^*\hat{S}_n\psi_{\leq k}^*\right)\mu_k\left(\Lambda_{\mathcal{A}\Sigma^{\beta'/2}}^{\leq k}\right).
\end{aligned}$$

Next we upper bound RHS, first we bound the first term in RHS

$$\begin{aligned}
\text{First term in RHS} &= \xi_{\leq k}^T\Lambda_{\mathcal{A}^{-1}\Sigma^{1-\beta-\beta'}/2}^{\leq k}\phi_{\leq k}\mathcal{A}_{\leq k}\Sigma_{\leq k}^{\beta-1}\hat{S}_n^*(\tilde{K}_{>k}^\gamma)^{-1}\hat{S}_n\phi_{>k}^*\Lambda_{\mathcal{A}}^{>k}\phi_{>k}f^*_{>k} \\
&= \xi_{\leq k}^T\Lambda_{\Sigma^{-\beta'}/2}^{\leq k}\phi_{\leq k}\hat{S}_n^*(\tilde{K}_{>k}^\gamma)^{-1}\hat{S}_n\phi_{>k}^*\Lambda_{\mathcal{A}}^{>k}\phi_{>k}f^*_{>k}.
\end{aligned}$$

Since $(1-\beta')/2 - 1/2 = -\beta'/2$, it equals to

$$\xi_{\leq k}^T \Lambda_{\Sigma(1-\beta')/2}^{\leq k} \Lambda_{\Sigma^{-1}/2}^{\leq k} \phi_{\leq k} \hat{S}_n^*(\tilde{K}_{>k}^\gamma)^{-1} \hat{S}_n \mathcal{A}_{>k} f_{>k}^*$$

$$= \xi_{\leq k}^T \Lambda_{\Sigma(1-\beta')/2}^{\leq k} \psi_{\leq k} \hat{S}_n^*(\tilde{K}_{>k}^\gamma)^{-1} \hat{S}_n \mathcal{A}_{>k} f_{>k}^*$$

$$\leq \|\xi_{\leq k}\|_{\Lambda_{\Sigma(1-\beta')}^{\leq k}} \mu_1((\tilde{K}_{>k}^\gamma)^{-1}) \sqrt{\mu_1(\underbrace{\psi_{\leq k} \hat{S}_n^* \hat{S}_n \psi_{\leq k}^*}_{k \times k})} \|\hat{S}_n \mathcal{A}_{>k} f_{>k}^*\|.$$

Then we bound the second term in RHS.

$$\text{Second term in RHS} = \xi_{\leq k}^T \Lambda_{\mathcal{A}^{-1}\Sigma^{1-\beta-\beta'/2}}^{\leq k} \phi_{\leq k} f_{\leq k}^* = \xi_{\leq k}^T \Lambda_{\Sigma(1-\beta')/2}^{\leq k} \Lambda_{\mathcal{A}^{-1}\Sigma^{1/2-\beta}}^{\leq k} \phi_{\leq k} f_{\leq k}^*$$

$$\leq \|\xi_{\leq k}\|_{\Lambda_{\Sigma^{1-\beta'}}^{\leq k}} \|\phi_{\leq k} f_{\leq k}^*\|_{\Lambda_{\mathcal{A}^{-2}\Sigma^{1-2\beta}}^{\leq k}}.$$

Therefore, gather the terms we have

$$\|\xi_{\leq k}\|_{\Lambda_{\Sigma^{1-\beta'}}^{\leq k}}^2 \mu_n((\tilde{K}_{>k}^\gamma)^{-1}) \mu_k\left(\Lambda_{\mathcal{A}^{1/2}\Sigma^{\beta'/4}}^{\leq k} \psi_{\leq k} \hat{S}_n^*(\tilde{K}_{>k}^\gamma)^{-1} \hat{S}_n \psi_{\leq k}^* \Lambda_{\mathcal{A}^{1/2}\Sigma^{\beta'/4}}^{\leq k}\right)$$

$$\leq \|\xi_{\leq k}\|_{\Lambda_{\Sigma(1-\beta')}^{\leq k}} \mu_1((\tilde{K}_{>k}^\gamma)^{-1}) \sqrt{\mu_1(\underbrace{\psi_{\leq k} \hat{S}_n^* \hat{S}_n \psi_{\leq k}^*}_{k \times k})} \|\hat{S}_n \mathcal{A}_{>k} f_{>k}^*\|$$

$$+ \|\xi_{\leq k}\|_{\Lambda_{\Sigma^{1-\beta'}}^{\leq k}} \|\phi_{\leq k} f_{\leq k}^*\|_{\Lambda_{\mathcal{A}^{-2}\Sigma^{1-2\beta}}^{\leq k}}.$$

So

$$\|\xi_{\leq k}\|_{\Lambda_{\Sigma^{1-\beta'}}^{\leq k}} \leq \frac{\mu_1((\tilde{K}_{>k}^\gamma)^{-1})}{\mu_n((\tilde{K}_{>k}^\gamma)^{-1})} \frac{\sqrt{\mu_1(\psi_{\leq k} \hat{S}_n^* \hat{S}_n \psi_{\leq k}^*)}}{\mu_k\left(\psi_{\leq k} \hat{S}_n^* \hat{S}_n \psi_{\leq k}^*\right) \mu_k\left(\Lambda_{\mathcal{A}\Sigma^{\beta'/2}}^{\leq k}\right)} \|\hat{S}_n \mathcal{A}_{>k} f_{>k}^*\|$$

$$+ \frac{\|\phi_{\leq k} f_{\leq k}^*\|_{\Lambda_{\mathcal{A}^{-2}\Sigma^{1-2\beta}}^{\leq k}}}{\mu_n((\tilde{K}_{>k}^\gamma)^{-1}) \mu_k\left(\psi_{\leq k} \hat{S}_n^* \hat{S}_n \psi_{\leq k}^*\right) \mu_k\left(\Lambda_{\mathcal{A}\Sigma^{\beta'/2}}^{\leq k}\right)}.$$

By $\|a+b\|^2 \leq 2(\|a\|^2 + \|b\|^2)$, we can bound $\|\xi_{\leq k}\|_{\Sigma^{1-\beta'}}^2$ by

$$2\left(\frac{\mu_1((\tilde{K}_{>k}^\gamma)^{-1})^2}{\mu_n((\tilde{K}_{>k}^\gamma)^{-1})^2} \frac{\mu_1(\overbrace{\psi_{\leq k} \hat{S}_n^* \hat{S}_n \psi_{\leq k}^*}^{k \times k})}{\mu_k(\psi_{\leq k} \hat{S}_n^* \hat{S}_n \psi_{\leq k}^*)^2 \mu_k(\Lambda_{\mathcal{A}^2\Sigma^{\beta'}}^{\leq k})} \|\hat{S}_n \mathcal{A}_{>k} f_{>k}^*\|^2\right.$$

$$\left. + \frac{\|\phi_{\leq k} f_{\leq k}^*\|_{\Lambda_{\mathcal{A}^{-2}\Sigma^{1-2\beta}}^{\leq k}}^2}{\mu_k(\psi_{\leq k} \hat{S}_n^* \hat{S}_n \psi_{\leq k}^*)^2 \mu_k(\Lambda_{\mathcal{A}^2\Sigma^{\beta'}}^{\leq k})}\right).$$

Now we discuss the $>k$ case, which is more complicated, we bound it by three quantities by the fact that $(A+B+C)^2 \leq 3(A^2 + B^2 + C^2)$ and bound them respectively as follows

$$\|\phi_{>k} f_{>k}^* - \phi_{>k} \mathcal{A}_{>k} \Sigma_{>k}^{\beta-1} \hat{S}_n^*(\tilde{K}^\gamma)^{-1} \hat{S}_n \mathcal{A} f^*\|_{\Lambda_{\Sigma^{1-\beta'}}^{>k}}^2$$

$$\leq 3(\|\phi_{>k} f_{>k}^*\|_{\Lambda_{\Sigma^{1-\beta'}}^{>k}}^2 + \|\phi_{>k} \mathcal{A}_{>k} \Sigma_{>k}^{\beta-1} \hat{S}_n^*(\tilde{K}^\gamma)^{-1} \hat{S}_n \mathcal{A}_{>k} f_{>k}^*\|_{\Lambda_{\Sigma^{1-\beta'}}^{>k}}^2 + \|\phi_{>k} \mathcal{A}_{>k} \Sigma_{>k}^{\beta-1} \hat{S}_n^*(\tilde{K}^\gamma)^{-1} \hat{S}_n \mathcal{A}_{\leq k} f_{\leq k}^*\|_{\Lambda_{\Sigma^{1-\beta'}}^{>k}}^2).$$

We first bound the second term

$$\|\phi_{>k} \mathcal{A}_{>k} \Sigma_{>k}^{\beta-1} \hat{S}_n^*(\tilde{K}^\gamma)^{-1} \hat{S}_n \mathcal{A}_{>k} f_{>k}^*\|_{\Lambda_{\Sigma^{1-\beta'}}^{>k}}^2$$

$$\leq \|\Lambda_{\Sigma^{1-\beta'}}^{>k}\| \|\phi_{>k} \mathcal{A}_{>k} \Sigma_{>k}^{\beta-1} \hat{S}_n^*(\tilde{K}^\gamma)^{-1} \hat{S}_n \mathcal{A}_{>k} f_{>k}^*\|^2$$

$$= \|\Lambda_{\Sigma^{1-\beta'}}^{>k}\| \|\Lambda_{\mathcal{A}\Sigma^{\beta-1}}^{>k} \phi_{>k} \hat{S}_n^*(\tilde{K}^\gamma)^{-1} \hat{S}_n \phi_{>k}^* \Lambda_{\mathcal{A}}^{>k} \phi_{>k} f_{>k}^*\|^2$$

$$\leq \|\Lambda_{\Sigma^{1-\beta'}}^{>k}\| \mu_1[(\tilde{K}^\gamma)^{-1}]^2 \|\hat{S}_n \mathcal{A}_{>k} f_{>k}^*\|^2 \mu_1(\underbrace{\hat{S}_n \phi_{>k}^* \Lambda_{\mathcal{A}^2\Sigma^{2(\beta-1)}}^{>k} \phi_{>k} \hat{S}_n^*}_{n \times n})$$

$$\leq \|\Lambda_{\Sigma^{1-\beta'}}^{>k}\| \mu_1[(\tilde{K}_{>k}^\gamma)^{-1}]^2 \|\hat{S}_n \mathcal{A}_{>k} f_{>k}^*\|^2 \mu_1(\underbrace{\hat{S}_n \phi_{>k}^* \Lambda_{\mathcal{A}^2\Sigma^{2(\beta-1)}}^{>k} \phi_{>k} \hat{S}_n^*}_{n \times n}).$$

The last inequality is by $\mu_1((\tilde{K}_{>k}^\gamma)^{-1}) \geq \mu_1((\tilde{K}^\gamma)^{-1})$.

Then we move on to bound the third term, that is, we want to bound

$$\|\phi_{>k}\mathcal{A}_{>k}\Sigma_{>k}^{\beta-1}\hat{S}_n^*(\tilde{K}^\gamma)^{-1}\hat{S}_n\mathcal{A}_{\leq k}f_{\leq k}^*\|_{\Lambda_{\Sigma^{1-\beta'}}^{>k}}^2$$

$$= \|\Lambda_{\mathcal{A}\Sigma^{\beta-1}}^{>k}\phi_{>k}\hat{S}_n^*(\tilde{K}^\gamma)^{-1}\hat{S}_n\phi_{\leq k}^*\Lambda_{\overline{\mathcal{A}}}^{\leq k}\phi_{\leq k}f_{\leq k}^*\|_{\Lambda_{\Sigma^{1-\beta'}}^{>k}}^2.$$

First we deal with $(\tilde{K}^\gamma)^{-1}(\hat{S}_n\phi_{\leq k}^*)$ first, we can write it as

$$(\tilde{K}^\gamma)^{-1}(\hat{S}_n\phi_{\leq k}^*) = (\tilde{K}_{>k}^\gamma + (\hat{S}_n\phi_{\leq k}^*)\Lambda_{\mathcal{A}^2\Sigma^{\beta-1}}^{\leq k}(\phi_{\leq k}\hat{S}_n^*))^{-1}(\hat{S}_n\phi_{\leq k}^*),$$

then apply A.6 with $A = \tilde{K}_{>k}^\gamma$, $U = \hat{S}_n\phi_{\leq k}^*$, $C = \Lambda_{\mathcal{A}^2\Sigma^{\beta-1}}^{\leq k}$, $V = \phi_{\leq k}\hat{S}_n^*$, we have it equal to

$$(\tilde{K}_{>k}^\gamma)^{-1}(\hat{S}_n\phi_{\leq k}^*)(I_k + \Lambda_{\mathcal{A}^2\Sigma^{\beta-1}}^{\leq k}(\phi_{\leq k}\hat{S}_n^*)(\tilde{K}_{>k}^\gamma)^{-1}(\hat{S}_n\phi_{\leq k}^*))^{-1}.$$

Then we sub. the identity above to obtain

$$\|\Lambda_{\mathcal{A}\Sigma^{\beta-1}}^{>k}\phi_{>k}\hat{S}_n^*(\tilde{K}^\gamma)^{-1}\hat{S}_n\phi_{\leq k}^*\Lambda_{\overline{\mathcal{A}}}^{\leq k}\phi_{\leq k}f_{\leq k}\|_{\Lambda_{\Sigma^{1-\beta'}}^{>k}}^2$$

$$= \|\Lambda_{\Sigma^{(1-\beta')/2}}^{>k}\Lambda_{\mathcal{A}\Sigma^{\beta-1}}^{>k}\phi_{>k}\hat{S}_n(\tilde{K}_{>k}^\gamma)^{-1}\hat{S}_n\phi_{\leq k}^*(I_k + \Lambda_{\mathcal{A}^2\Sigma^{\beta-1}}^{\leq k}\phi_{\leq k}\hat{S}_n^*(\tilde{K}_{>k}^\gamma)^{-1}\hat{S}_n\phi_{\leq k}^*)^{-1}\Lambda_{\overline{\mathcal{A}}}^{\leq k}\phi_{\leq k}f_{\leq k}^*\|^2$$

$$= \|\Lambda_{\mathcal{A}\Sigma^{(-\beta'+2\beta-1)/2}}^{>k}\phi_{>k}\hat{S}_n^*(\tilde{K}_{>k}^\gamma)^{-1}\hat{S}_n\phi_{\leq k}^*(\Lambda_{\mathcal{A}^2\Sigma^{\beta-1/2}}^{\leq k}(\Lambda_{\mathcal{A}^{-2}\Sigma^{-\beta}}^{\leq k} + \psi_{\leq k}\hat{S}_n^*(\tilde{K}_{>k}^\gamma)^{-1}\hat{S}_n\psi_{\leq k}^*)\Lambda_{\Sigma^{1/2}}^{\leq k})^{-1}\Lambda_{\overline{\mathcal{A}}}^{\leq k}\phi_{\leq k}f_{\leq k}^*\|^2$$

$$= \|\Lambda_{\mathcal{A}\Sigma^{(-\beta'+2\beta-1)/2}}^{>k}\phi_{>k}\hat{S}_n^*(\tilde{K}_{>k}^\gamma)^{-1}\hat{S}_n\phi_{\leq k}^*\Lambda_{\Sigma^{-1/2}}^{\leq k}(\Lambda_{\mathcal{A}^{-2}\Sigma^{-\beta}}^{\leq k} + \psi_{\leq k}\hat{S}_n^*(\tilde{K}_{>k}^\gamma)^{-1}\hat{S}_n\psi_{\leq k}^*)^{-1}\Lambda_{\mathcal{A}^{-2}\Sigma^{1/2-\beta}}^{\leq k}\Lambda_{\overline{\mathcal{A}}}^{\leq k}\phi_{\leq k}f_{\leq k}^*\|^2$$

$$= \|\underbrace{\Lambda_{\mathcal{A}\Sigma^{(-\beta'+2\beta)/2}}^{>k}\psi_{>k}\hat{S}_n^*(\tilde{K}_{>k}^\gamma)^{-1/2}}_{(1)}\underbrace{(\tilde{K}_{>k}^\gamma)^{-1/2}}_{(2)}\underbrace{\hat{S}_n\psi_{\leq k}^*}_{(3)}\underbrace{(\Lambda_{\mathcal{A}^{-2}\Sigma^{-\beta}}^{\leq k} + \psi_{\leq k}\hat{S}_n^*(\tilde{K}_{>k}^\gamma)^{-1}\hat{S}_n\psi_{\leq k}^*)^{-1}}_{(4)}\underbrace{\Lambda_{\mathcal{A}^{-1}\Sigma^{1/2-\beta}}^{\leq k}\phi_{\leq k}f_{\leq k}^*}_{(5)}\|^2.$$

Above can be bounded by

$$\underbrace{\|(\tilde{K}_{>k}^\gamma)^{-1/2}\hat{S}_n\psi_{>k}^*\Lambda_{\mathcal{A}^2\Sigma^{-\beta'+2\beta}}^{>k}\psi_{>k}\hat{S}_n^*(\tilde{K}_{>k}^\gamma)^{-1/2}\|}_{(1)}\underbrace{\mu_1((\tilde{K}_{>k}^\gamma)^{-1})}_{(2)}$$

$$\underbrace{\mu_1(\psi_{\leq k}\hat{S}_n^*\hat{S}_n\psi_{\leq k}^*)}_{(3)}\underbrace{\mu_1((\psi_{\leq k}\hat{S}_n^*(\tilde{K}_{>k}^\gamma)^{-1}\hat{S}_n\psi_{\leq k}^*)^{-1})^2}_{(4)}\underbrace{\|\phi_{\leq k}f_{\leq k}^*\|_{\Lambda_{\mathcal{A}^{-2}\Sigma^{1-2\beta}}^{\leq k}}}_{(5)}.$$

For (1) it can be upper bounded by

$$\|(\tilde{K}_{>k}^\gamma)^{-1/2}\hat{S}_n\psi_{>k}^*\Lambda_{\mathcal{A}^2\Sigma^{-\beta'+2\beta}}^{>k}\psi_{>k}\hat{S}_n^*(\tilde{K}_{>k}^\gamma)^{-1/2}\|$$

$$\leq \|\Lambda_{\Sigma^{-\beta'+\beta}}^{>k}\|\|I_n - n\gamma_n(\tilde{K}_{>k}^\gamma)^{-1}\|$$

$$\leq \|\Lambda_{\Sigma^{-\beta'+\beta}}^{>k}\|,$$

where the last transition is by the fact that $I_n - n\gamma_n(\tilde{K}_{>k}^\gamma)^{-1}$ is PSD matrix with norm bounded by 1 for $\gamma_n \geq 0$.

For (4), it can be upper bounded by

$$\mu_1((\psi_{\leq k}\hat{S}_n^*(\tilde{K}_{>k}^\gamma)^{-1}\hat{S}_n\psi_{\leq k}^*)^{-1})^2$$

$$= \frac{1}{\mu_k((\psi_{\leq k}\hat{S}_n^*(\tilde{K}_{>k}^\gamma)^{-1}\hat{S}_n\psi_{\leq k}^*))^2}$$

$$\leq \frac{1}{\mu_k((\psi_{\leq k}\hat{S}_n^*\hat{S}_n\psi_{\leq k}^*))^2\mu_n((\tilde{K}_{>k}^\gamma)^{-1})^2}.$$

Therefore, the third term overall can be bounded by

$$\|\Lambda_{\Sigma^{-\beta'+\beta}}^{>k}\|\frac{\mu_1((\tilde{K}_{>k}^\gamma)^{-1})}{\mu_n((\tilde{K}_{>k}^\gamma)^{-1})^2}\frac{\mu_1(\psi_{\leq k}\hat{S}_n^*\hat{S}_n\psi_{\leq k}^*)}{\mu_k(\psi_{\leq k}\hat{S}_n^*\hat{S}_n\psi_{\leq k}^*)^2}\|\phi_{\leq k}f_{\leq k}^*\|_{\Lambda_{\mathcal{A}^{-2}\Sigma^{1-2\beta}}^{\leq k}}.$$

We gather all the terms then we get the desired bound. $\qquad\square$

**Lemma E.2** (Simplified Upper bound for bias using concentration). *There exists some absolute constant $c, c', C_2 > 0$ s.t. for any $k \in \mathbb{N}$ with $c\beta_k k \log(k) \leq n$, it holds w.p. at least $1 - \delta - 8\exp(-\frac{c'}{\beta_k^2}\frac{n}{k})$, the bias can be upper bounded as:*

$$
B \leq C_2 \Big( \frac{\mu_1(\frac{1}{n}\tilde{K}_{>k}^\gamma)^2}{\mu_n(\frac{1}{n}\tilde{K}_{>k}^\gamma)^2} \frac{1}{p_k^2 \lambda_k^{\beta'}} \big( \frac{1}{\delta} \|\phi_{>k}\mathcal{A}_{>k}f_{>k}\|_{\Lambda_\Sigma^{\geq k}}^2 \big)
$$

$$
+ \frac{\mu_1(\frac{1}{n}\tilde{K}_{>k}^\gamma)^2 \|\phi_{\leq k}f_{\leq k}^*\|_{\Lambda_{\mathcal{A}^{-2}\Sigma^{1-2\beta}}^{\leq k}}^2}{p_k^2 \lambda_k^{\beta'}}
$$

$$
+ \|\phi_{>k}f_{>k}^*\|_{\Lambda_{\Sigma^{1-\beta'}}^{>k}}^2
$$

$$
+ \|\Lambda_{\Sigma^{1-\beta'}}^{>k}\| \frac{1}{\mu_n(\frac{1}{n}\tilde{K}_{>k}^\gamma)^2} \big( \frac{1}{\delta} \|\phi_{>k}\mathcal{A}_{>k}f_{>k}\|_{\Lambda_\Sigma^{\geq k}}^2 \big) (p_{k+1}^2 \lambda_{k+1}^{2\beta-1})
$$

$$
+ \|\Lambda_{\Sigma^{-\beta'+\beta}}^{>k}\| \frac{\mu_1(\frac{1}{n}\tilde{K}_{>k}^\gamma)^2}{\mu_n(\frac{1}{n}\tilde{K}_{>k}^\gamma)} \|\phi_{\leq k}f_{\leq k}^*\|_{\Lambda_{\mathcal{A}^{-2}\Sigma^{1-2\beta}}^{\leq k}}^2 \Big).
$$

*Proof.* Recall that from E.1 we have

$$
B \leq 3 \Bigg( \frac{\mu_1((\tilde{K}_{>k}^\gamma)^{-1})^2}{\mu_n((\tilde{K}_{>k}^\gamma)^{-1})^2} \frac{\mu_1(\psi_{\leq k}\hat{S}_n^* \hat{S}_n \psi_{\leq k}^*)}{\mu_k(\psi_{\leq k}\hat{S}_n^* \hat{S}_n \psi_{\leq k}^*)^2 \mu_k(\Lambda_{\mathcal{A}^2\Sigma^{\beta'}}^{\leq k})} \|\hat{S}_n \mathcal{A}_{>k}f_{>k}^*\|^2
$$

$$
+ \frac{\|\phi_{\leq k}f_{\leq k}^*\|_{\Lambda_{\mathcal{A}^{-2}\Sigma^{1-2\beta}}^{\leq k}}^2}{\mu_n((\tilde{K}_{>k}^\gamma)^{-1})^2 \mu_k(\psi_{\leq k}\hat{S}_n^* \hat{S}_n \psi_{\leq k}^*)^2 \mu_k(\Lambda_{\mathcal{A}^2\Sigma^{\beta'}}^{\leq k})}
$$

$$
+ \|\phi_{>k}f_{>k}^*\|_{\Lambda_{\Sigma^{1-\beta'}}^{>k}}^2
$$

$$
+ \|\Lambda_{\Sigma^{1-\beta'}}^{>k}\| \mu_1[(\tilde{K}_{>k}^\gamma)^{-1}]^2 \|\hat{S}_n \mathcal{A}_{>k}f_{>k}\|^2 \mu_1\big(\underbrace{\hat{S}_n \psi_{>k}^* \Lambda_{\mathcal{A}^2\Sigma^{2\beta-1}}^{>k} \psi_{>k}\hat{S}_n^*}_{n \times n}\big)
$$

$$
+ \|\Lambda_{\Sigma^{-\beta'+\beta}}^{>k}\| \frac{\mu_1((\tilde{K}_{>k}^\gamma)^{-1})}{\mu_n((\tilde{K}_{>k}^\gamma)^{-1})^2} \frac{\mu_1(\psi_{\leq k}\hat{S}_n^* \hat{S}_n \psi_{\leq k}^*)}{\mu_k(\psi_{\leq k}\hat{S}_n^* \hat{S}_n \psi_{\leq k}^*)^2} \|\phi_{\leq k}f_{\leq k}^*\|_{\Lambda_{\mathcal{A}^{-2}\Sigma^{1-2\beta}}^{\leq k}} \Bigg).
$$

We first apply $\mu_1((\tilde{K}_{>k}^\gamma)^{-1}) = \frac{1}{n\mu_n(\frac{1}{n}\tilde{K}_{>k}^\gamma)}$ and $\mu_n((\tilde{K}_{>k}^\gamma)^{-1}) = \frac{1}{n\mu_1(\frac{1}{n}\tilde{K}_{>k}^\gamma)}$ , also apply concentration inequalities using Lemma B.3, Lemma B.4 and Lemma C.2 , then w.p. at least $1 - \delta - 8\exp(-\frac{c}{\beta_k^2}\frac{n}{k})$, we can obtain bound like this

$$
\Big( \frac{\mu_1(\frac{1}{n}\tilde{K}_{>k}^\gamma)^2}{\mu_n(\frac{1}{n}\tilde{K}_{>k}^\gamma)^2} \frac{c_1 n}{c_2^2 n^2 p_k^2 \lambda_k^{\beta'}} \big( \frac{1}{\delta} n \|\phi_{>k}\mathcal{A}_{>k}f_{>k}\|_{\Lambda_\Sigma^{\geq k}}^2 \big)
$$

$$
+ \frac{\mu_1(\frac{1}{n}\tilde{K}_{>k}^\gamma)^2 \|\phi_{\leq k}f_{\leq k}^*\|_{\Lambda_{\mathcal{A}^{-2}\Sigma^{1-2\beta}}^{\leq k}}^2}{c_1^2 p_k^2 \lambda_k^{\beta'}}
$$

$$
+ \|\phi_{>k}f_{>k}^*\|_{\Lambda_{\Sigma^{1-\beta'}}^{>k}}^2
$$

$$
+ \|\Lambda_{\Sigma^{1-\beta'}}^{>k}\| \frac{1}{n^2 \mu_n(\frac{1}{n}\tilde{K}_{>k}^\gamma)^2} \big( \frac{1}{\delta} n \|\phi_{>k}\mathcal{A}_{>k}f_{>k}\|_{\Lambda_\Sigma^{\geq k}}^2 \big) (n p_{k+1}^2 \lambda_{k+1}^{2\beta-1})
$$

$$
+ \|\Lambda_{-\beta'+\beta}^{>k}\| \frac{n^2 \mu_1(\frac{1}{n}\tilde{K}_{>k}^\gamma)^2}{n\mu_n(\frac{1}{n}\tilde{K}_{>k}^\gamma)} \frac{c_2 n}{c_1^2 n^2} \|\phi_{\leq k}f_{\leq k}^*\|_{\Lambda_{\mathcal{A}^{-2}\Sigma^{1-2\beta}}^{\leq k}}^2 \Big).
$$

This can be upper bounded by

$$C_2 \Big( \frac{\mu_1(\frac{1}{n}\tilde{K}_{>k}^{\gamma})^2}{\mu_n(\frac{1}{n}\tilde{K}_{>k}^{\gamma})^2} \frac{1}{p_k^2 \lambda_k^{\beta'}} \Big( \frac{1}{\delta} \|\phi_{>k}\mathcal{A}_{>k}f_{>k}\|_{\Lambda_{\tilde{\Sigma}}^{\geq k}}^2 \Big)$$

$$+ \frac{\mu_1(\frac{1}{n}\tilde{K}_{>k}^{\gamma})^2 \|\phi_{\leq k}f_{\leq k}^*\|_{\Lambda_{\mathcal{A}^{-2}\Sigma^{1-2\beta}}^{\leq k}}^2}{p_k^2 \lambda_k^{\beta'}}$$

$$+ \|\phi_{>k}f_{>k}^*\|_{\Lambda_{\Sigma^{1-\beta'}}^{>k}}^2$$

$$+ \|\Lambda_{\Sigma^{1-\beta'}}^{>k}\| \frac{1}{\mu_n(\frac{1}{n}\tilde{K}_{>k}^{\gamma})^2} \Big( \frac{1}{\delta} \|\phi_{>k}\mathcal{A}_{>k}f_{>k}\|_{\Lambda_{\tilde{\Sigma}}^{\geq k}}^2 \Big)(p_{k+1}^2 \lambda_{k+1}^{2\beta-1})$$

$$+ \|\Lambda_{-\beta'+\beta}^{>k}\| \frac{\mu_1(\frac{1}{n}\tilde{K}_{>k}^{\gamma})^2}{\mu_n(\frac{1}{n}\tilde{K}_{>k}^{\gamma})} \|\phi_{\leq k}f_{\leq k}^*\|_{\Lambda_{\mathcal{A}^{-2}\Sigma^{1-2\beta}}^{\leq k}}^2 \Big)$$

where $C_2 > 0$ is some constant only depends on $c_1, c_2$. $\qquad \square$

**Theorem E.3** (Bound on bias). *There exists some absolute constant $C_2, c, c' > 0$ s.t. for any $k \in \mathbb{N}$ with $c\beta_k k \log(k) \leq n$, it holds w.p. at least $1 - \delta - 8\exp(-\frac{c'}{\beta_k^2}\frac{n}{k})$, the bias can be further bounded as*

$$B \leq C_2 \frac{\rho_{k,n}^3}{\delta} \Big( \|\phi_{>k}\mathcal{A}_{>k}f_{>k}\|_{\Lambda_{\tilde{\Sigma}}^{\geq k}}^2 \frac{1}{p_k^2 \lambda_k^{\beta'}} + \|\phi_{\leq k}f_{\leq k}^*\|_{\Lambda_{\mathcal{A}^{-2}\Sigma^{1-2\beta}}^{\leq k}}^2 \Big(\gamma_n + \frac{\beta_k \operatorname{tr}(\tilde{\Sigma}_{>k})}{n}\Big)^2 \frac{1}{p_k^2 \lambda_k^{\beta'}}$$

$$+ \|\phi_{>k}f_{>k}^*\|_{\Lambda_{\Sigma^{1-\beta'}}^{>k}}^2 \Big).$$

*Proof.* We refer result from previous lemma E.2.

$$B \leq C_2 \Big( \frac{\mu_1(\frac{1}{n}\tilde{K}_{>k}^{\gamma})^2}{\mu_n(\frac{1}{n}\tilde{K}_{>k}^{\gamma})^2} \frac{1}{p_k^2 \lambda_k^{\beta'}} \Big( \frac{1}{\delta} \|\phi_{>k}\mathcal{A}_{>k}f_{>k}\|_{\Lambda_{\tilde{\Sigma}}^{\geq k}}^2 \Big)$$

$$+ \frac{\mu_1(\frac{1}{n}\tilde{K}_{>k}^{\gamma})^2 \|\phi_{\leq k}f_{\leq k}^*\|_{\Lambda_{\mathcal{A}^{-2}\Sigma^{1-2\beta}}^{\leq k}}^2}{p_k^2 \lambda_k^{\beta'}}$$

$$+ \|\phi_{>k}f_{>k}^*\|_{\Lambda_{\Sigma^{1-\beta'}}^{>k}}^2$$

$$+ \|\Lambda_{\Sigma^{1-\beta'}}^{>k}\| \frac{1}{\mu_n(\frac{1}{n}\tilde{K}_{>k}^{\gamma})^2} \Big( \frac{1}{\delta} \|\phi_{>k}\mathcal{A}_{>k}f_{>k}\|_{\Lambda_{\tilde{\Sigma}}^{\geq k}}^2 \Big)(p_{k+1}^2 \lambda_{k+1}^{2\beta-1})$$

$$+ \|\Lambda_{\Sigma^{-\beta'+\beta}}^{>k}\| \frac{\mu_1(\frac{1}{n}\tilde{K}_{>k}^{\gamma})^2}{\mu_n(\frac{1}{n}\tilde{K}_{>k}^{\gamma})} \|\phi_{\leq k}f_{\leq k}^*\|_{\Lambda_{\mathcal{A}^{-2}\Sigma^{1-2\beta}}^{\leq k}}^2 \Big).$$

Note that by definition of $\rho_{k,n}$ (refer to Definition 3.2), we have a following estimations:

$$\frac{\mu_1(\frac{1}{n}\tilde{K}_{>k}^{\gamma})^2}{\mu_n(\frac{1}{n}\tilde{K}_{>k}^{\gamma})^2} = \frac{(\mu_1(\frac{1}{n}\tilde{K}_{>k}) + \gamma_n)^2}{(\mu_n(\frac{1}{n}\tilde{K}_{>k}) + \gamma_n)^2} \leq \rho_{k,n}^2,$$

$$\mu_1(\frac{1}{n}\tilde{K}_{>k}^{\gamma})^2 = \frac{\mu_1(\frac{1}{n}\tilde{K}_{>k}^{\gamma})^2}{\mu_n(\frac{1}{n}\tilde{K}_{>k}^{\gamma})^2} \mu_n(\frac{1}{n}\tilde{K}_{>k}^{\gamma})^2$$

$$\leq \rho_{k,n}^2 (\frac{1}{n}\operatorname{tr}(\frac{1}{n}\tilde{K}_{>k}^{\gamma}))^2 \leq \rho_{k,n}^2 (\gamma_n + \frac{1}{n}\sum_{j=1}^{n}\sum_{i>k}\lambda_i^\beta p_i^2 \psi_i(x_j)^2)^2$$

$$\leq \rho_{k,n}^2 (\gamma_n + \frac{\beta_k \operatorname{tr}(\tilde{\Sigma}_{>k})}{n})^2,$$

$$\frac{\|\Lambda_{\mathcal{A}^2\Sigma^\beta}^{>k}\|}{\mu_n(\frac{1}{n}\tilde{K}_{>k})} \leq \rho_{k,n}$$

and

$$\|\Lambda^{>k}_{\Sigma^{-\beta'+\beta}}\|\frac{\mu_1(\frac{1}{n}\tilde{K}^{\gamma}_{>k})^2}{\mu_n(\frac{1}{n}\tilde{K}^{\gamma}_{>k})} = \frac{\|\Lambda^{>k}_{\mathcal{A}^2\Sigma^{\beta}}\|}{\mu_n(\frac{1}{n}\tilde{K}_{>k})}\|\Lambda^{>k}_{\mathcal{A}^{-2}\Sigma^{-\beta'}}\|\mu_1(\frac{1}{n}\tilde{K}^{\gamma}_{>k})^2$$

$$\leq \rho^3_{k,n}(\gamma_n + \frac{\beta_k\,\mathrm{tr}(\tilde{\Sigma}_{>k})}{n})^2\|\Lambda^{>k}_{\mathcal{A}^{-2}\Sigma^{-\beta'}}\|.$$

We bound first and forth term first

$$\frac{\mu_1(\frac{1}{n}\tilde{K}^{\gamma}_{>k})^2}{\mu_n(\frac{1}{n}\tilde{K}^{\gamma}_{>k})^2}\frac{1}{p_k^2\lambda_k^{\beta'}}(\frac{1}{\delta}\|\phi_{>k}\mathcal{A}_{>k}f_{>k}\|^2_{\Lambda^{\geq k}_{\Sigma}}) + \|\Lambda^{>k}_{\Sigma^{1-\beta'}}\|\frac{1}{\mu_n(\frac{1}{n}\tilde{K}^{\gamma}_{>k})^2}(\frac{1}{\delta}\|\phi_{>k}\mathcal{A}_{>k}f_{>k}\|^2_{\Lambda^{\geq k}_{\Sigma}})(p^2_{k+1}\lambda^{2\beta-1}_{k+1})$$

$$\leq(\frac{1}{\delta}\|\phi_{>k}\mathcal{A}_{>k}f_{>k}\|^2_{\Lambda^{\geq k}_{\Sigma}})(\rho^2_{k,n}\frac{1}{p_k^2\lambda_k^{\beta'}} + \frac{\|\Lambda^{>k}_{\mathcal{A}^4\Sigma^{2\beta}}\|}{\mu_n(\frac{1}{n}\tilde{K}^{\gamma}_{>k})^2}p^2_{k+1}\lambda^{2\beta-1}_{k+1}\|\Lambda^{>k}_{\mathcal{A}^{-4}\Sigma^{1-\beta'-2\beta}}\|)$$

$$\leq\rho^2_{k,n}(\frac{1}{\delta}\|\phi_{>k}\mathcal{A}_{>k}f_{>k}\|^2_{\Lambda^{\geq k}_{\Sigma}})(\frac{1}{p_k^2\lambda_k^{\beta'}} + p^2_{k+1}\lambda^{2\beta-1}_{k+1}\|\Lambda^{>k}_{\mathcal{A}^{-4}\Sigma^{1-\beta'-2\beta}}\|).$$

Since two terms here have the same order, we can just bound it by

$$c_1\rho^2_{k,n}(\frac{1}{\delta}\|\phi_{>k}\mathcal{A}_{>k}f_{>k}\|^2_{\Lambda^{\geq k}_{\Sigma}})\frac{1}{p_k^2\lambda_k^{\beta'}}$$

where $c_1$ is some constant.

Next we bound the second and fifth term

$$\frac{\mu_1(\frac{1}{n}\tilde{K}^{\gamma}_{>k})^2\|\phi_{\leq k}f^*_{\leq k}\|^2_{\Lambda^{\leq k}_{\mathcal{A}^{-2}\Sigma^{1-2\beta}}}}{p_k^2\lambda_k^{\beta'}} + \|\Lambda^{>k}_{\Sigma^{-\beta'+\beta}}\|\frac{\mu_1(\frac{1}{n}\tilde{K}^{\gamma}_{>k})^2}{\mu_n(\frac{1}{n}\tilde{K}^{\gamma}_{>k})}\|\phi_{\leq k}f^*_{\leq k}\|^2_{\Lambda^{\leq k}_{\mathcal{A}^{-2}\Sigma^{1-2\beta}}}$$

$$\leq\|\phi_{\leq k}f^*_{\leq k}\|^2_{\Lambda^{\leq k}_{\mathcal{A}^{-2}\Sigma^{1-2\beta}}}(\frac{1}{p_k^2\lambda_k^{\beta'}}\rho^2_{k,n}(\gamma_n + \frac{\beta_k\,\mathrm{tr}(\tilde{\Sigma}_{>k})}{n})^2 + \rho^3_{k,n}(\gamma_n + \frac{\beta_k\,\mathrm{tr}(\tilde{\Sigma}_{>k})}{n})^2\|\Lambda^{>k}_{\mathcal{A}^{-2}\Sigma^{-\beta'}}\|).$$

We know $\frac{1}{p_k^2\lambda_k^{\beta'}}$ and $\|\Lambda^{>k}_{\mathcal{A}^{-2}\Sigma^{-\beta'}}\|$ are of the same order, and $\rho_{k,n} \geq 1$ by its definition, therefore, the second term would be dominated by the fifth term. So we can bound it by

$$c_2\rho^3_{k,n}\|\phi_{\leq k}f^*_{\leq k}\|^2_{\Lambda^{\leq k}_{\mathcal{A}^{-2}\Sigma^{1-2\beta}}}(\gamma_n + \frac{\beta_k\,\mathrm{tr}(\tilde{\Sigma}_{>k})}{n})^2\frac{1}{p_k^2\lambda_k^{\beta'}}.$$

Therefore, the final bound becomes

$$C_2(c_1\rho^2_{k,n}(\frac{1}{\delta}\|\phi_{>k}\mathcal{A}_{>k}f_{>k}\|^2_{\Lambda^{\geq k}_{\Sigma}})\frac{1}{p_k^2\lambda_k^{\beta'}} + c_2\rho^3_{k,n}\|\phi_{\leq k}f^*_{\leq k}\|^2_{\Lambda^{\leq k}_{\mathcal{A}^{-2}\Sigma^{1-2\beta}}}(\gamma_n + \frac{\beta_k\,\mathrm{tr}(\tilde{\Sigma}_{>k})}{n})^2\frac{1}{p_k^2\lambda_k^{\beta'}}$$

$$+\|\phi_{>k}f^*_{>k}\|^2_{\Lambda^{>k}_{\Sigma^{1-\beta'}}})$$

$$\leq C'_2\frac{\rho^3_{k,n}}{\delta}(\|\phi_{>k}\mathcal{A}_{>k}f_{>k}\|^2_{\Lambda^{\geq k}_{\Sigma}}\frac{1}{p_k^2\lambda_k^{\beta'}} + \|\phi_{\leq k}f^*_{\leq k}\|^2_{\Lambda^{\leq k}_{\mathcal{A}^{-2}\Sigma^{1-2\beta}}}(\gamma_n + \frac{\beta_k\,\mathrm{tr}(\tilde{\Sigma}_{>k})}{n})^2\frac{1}{p_k^2\lambda_k^{\beta'}}$$

$$+\|\phi_{>k}f^*_{>k}\|^2_{\Lambda^{>k}_{\Sigma^{1-\beta'}}}),$$

$C'_2$ is w.r.t. $C_2, c_1, c_2$, and we finally just take $C_2 = C'_2$ to finish the proof. $\qquad\square$

## F APPLICATIONS

### F.1 REGULARIZED CASE

**Theorem F.1** (Regularized case, Proof of Theorem 4.1). *Let the kernel and target function satisfies Assumption 2.2, $\gamma_n = \Theta(n^{-\gamma})$, and $\gamma < 2p + \beta\lambda$, $2p + \lambda r > 0$ and $r > \beta'$ then for any $\delta > 0$, it holds w.p. $1 - \delta - O(\frac{1}{\log(n)})$ that*

$$V = \sigma_\varepsilon^2 O(n^{\max\{\frac{\gamma(1+2p+\lambda\beta')}{2p+\lambda\beta},0\}-1}), B \leq \frac{1}{\delta} \cdot \tilde{O}_n(n^{\frac{\gamma}{2p+\beta\lambda}(\max\{\lambda(\beta'-r),-2p+\lambda(\beta'-2\beta)\})}).$$

*Proof.* We use the two lemmas D.3, E.3 for upper bounding bias and variance in this proof, there exists some absolute constants $c, c' > 0$, first we need to pick $k$ s.t. $c\beta_k k \log(k) \leq n$, then the two lemmas will simultaneously hold w.p. at least $1 - \delta - 16 \exp(-\frac{c'}{\beta_k^2} \frac{n}{k})$. With regularization, we can pick $k$ large enough s.t. the concentration coefficient $\rho_{k,n} = o(1)$, to achieve so, we want $\mu_1(\frac{1}{n}\tilde{K}_{>k}) = O(\gamma_n)$. By Lemma F.6, we can show w.p. at least $1 - 4\frac{r_k}{k^4} \exp(-\frac{c'}{\beta_k} \frac{n}{r_k})$

$$\mu_1(\frac{1}{n}\tilde{K}_{>k}) = O_n(p_{k+1}^2 \lambda_{k+1}^\beta) = O_n(k^{-2p-\beta\lambda}) = O_n(\gamma_n) = O_n(n^{-\gamma}). \tag{8}$$

This can be achieved by setting $k(n) = \lceil n^{\frac{\gamma}{2p+\beta\lambda}} \rceil$, note that we have $\frac{\gamma}{2p+\beta\lambda} < 1$, therefore, $k(n) = O(\frac{n}{\log(n)})$ and the lemmas can be used for sufficient large $n$.

We combine the probability of both D.3, E.3 and 8 hold:

$$1 - \delta - 16 \exp\left(-\frac{c'}{\beta_k^2} \frac{n}{k}\right) - O(\frac{1}{k^3})\exp(-\Omega(\frac{n}{k})) = 1 - \delta - O(\frac{1}{n})$$

where we use the fact that $\frac{c'}{\beta_k^2} \frac{n}{k} = \Omega(\log(n))$ since $k(n) = O(\frac{n}{\log(n)})$.

Then now we can assume D.3, E.3 and 8 hold, and we provide the bound on variance and bias respectively.

By Theorem D.3 and we sub. $p_i = \Theta(i^{-p})$, $\lambda_i = \Theta(i^{-\lambda})$, $\|\Sigma_{>k}\| = p_{k+1}^2 \lambda_{k+1}^\beta = \Theta((k+1)^{-\beta\lambda-2p}) = \Theta(k^{-\beta\lambda-2p})$,

$$V \leq C_1 \sigma_\varepsilon^2 \rho_{k,n}^2 \left(\frac{\sum_{i\leq k} p_i^{-2}\lambda_i^{-\beta'}}{n} + \frac{\sum_{i>k} p_i^2\lambda_i^{-\beta'+2\beta}}{n\|\tilde{\Sigma}_{>k}\|^2}\right)$$

$$= \sigma_\varepsilon^2 O(1)O\left(\frac{\max\{k^{1+2p+\lambda\beta'}, 1\}}{n}, \frac{k^{1-2p+\lambda(\beta'-2\beta)}}{nk^{-2\beta\lambda-4p}}\right) = \sigma_\varepsilon^2 \tilde{O}\left(\frac{\max\{k^{1+2p+\lambda\beta'}, 1\}}{n}\right).$$

We substitute $k$ with $\lceil n^{\frac{\gamma}{2p+\beta\lambda}} \rceil$ to obtain the final bound

$$V = \sigma_\varepsilon^2 O(n^{\max\{\frac{\gamma(1+2p+\lambda\beta')}{2p+\lambda\beta}, 0\}-1}).$$

For bias, recall that by Theorem E.3, we have

$$B \leq C_2 \frac{\rho_{k,n}^3}{\delta} (\|\phi_{>k}\mathcal{A}_{>k}f_{>k}\|_{\Lambda_\Sigma^{\geq k}}^2 \frac{1}{p_k^2\lambda_k^{\beta'}}$$

$$+ \|\phi_{\leq k}f_{\leq k}^*\|_{\Lambda_{\mathcal{A}^{-2}\Sigma^{1-2\beta}}^{\leq k}}^2 (\gamma_n + \frac{\beta_k \operatorname{tr}(\tilde{\Sigma}_{>k})}{n})^2 \frac{1}{p_k^2\lambda_k^{\beta'}}$$

$$+ \|\phi_{>k}f_{>k}^*\|_{\Lambda_{\Sigma^{1-\beta'}}^{>k}}^2).$$

By $\operatorname{tr}(\tilde{\Sigma}_{>k}) = \sum_{i>k} p_i^2\lambda_i^\beta = O(k\lambda_k^\beta p_k^2) = O(k\gamma_n)$, then

$$(\gamma_n + \frac{\beta_k \operatorname{tr}(\tilde{\Sigma}_{>k})}{n})^2 = O((\gamma_n + \frac{n}{k}\gamma_n)^2) = O(\gamma_n^2) = O(k^{-4p-2\lambda\beta})$$

Recall that

$$\frac{\|\phi_{\leq k}f_{\leq k}^*\|_{\Lambda_{\mathcal{A}^{-2}\Sigma^{1-2\beta}}^{\leq k}}^2}{p_k^2\lambda_k^{\beta'}} = \tilde{O}(k^{\max\{1+4p-\lambda(1-\beta'-2\beta)-2r', 2p+\lambda\beta'\}}).$$

Therefore, the second term's bound is

$$O(k^{\max\{1-2r-\lambda(1-\beta'), -2p+\lambda(\beta'-2\beta)\}}).$$

Since $2p + \lambda r > 0$ and $r > \beta'$, we have $2p + 2r' + \lambda > 1$, and $2r' + (1-\beta')\lambda > 1$, We can quote Lemma F.5 for the remaining terms, so the third term's bound is

$$O(k^{1-2r'-(1-\beta')\lambda}).$$

First term's bound is the same as the second

$$O(k^{\max\{1-2r'-\lambda(1-\beta'),-2p+\lambda(\beta'-2\beta)\}}).$$

So we sub. $k = \lceil n^{\frac{\gamma}{2p+\beta\lambda}} \rceil$ to obtain

$$B \leq \frac{1}{\delta} \cdot \tilde{O}_n(n^{\frac{\gamma}{2p+\beta\lambda}(\max\{1-2r'-\lambda(1-\beta'),-2p+\lambda(\beta'-2\beta)\})}).$$

And we substitute $r' = \frac{1-\lambda(1-r)}{2}$ to obtain the final bound

$$B = O(n^{\frac{\gamma}{2p+\beta\lambda}(\max\{\lambda(\beta'-r),-2p+\lambda(\beta'-2\beta)\})}).$$

$\square$

## F.2 Interpolation Case

**Theorem F.2** (Interpolation case, proof of Theorem 4.2). *Let the kernel and target function satisfies Assumption 2.2, $2p+\beta\lambda > 0$, $2p+\lambda r > 0$ and $r > \beta'$, then for any $\delta > 0$ it holds w.p. at least $1 - \delta - O(\frac{1}{\log(n)})$ that*

$$V \leq \sigma_\varepsilon^2 \rho_{k,n}^2 \tilde{O}(n^{\max\{2p+\lambda\beta',-1\}}), B \leq \frac{\rho_{k,n}^3}{\delta} \tilde{O}(n^{\max\{\lambda(\beta'-r),-2p+\lambda(\beta'-2\beta)\}}),$$

*where $\rho_{k,n} = \tilde{O}(n^{2p+\beta\lambda-1})$, when features are well-behaved i.e. subGaussian it can be improved to $\rho_{k,n} = o(1)$.*

*Proof.* Same as regularized case, we use the two theorems D.3, E.3 for upper bounding bias and variance in this proof, there exists some absolute constants $c, c' > 0$, first we need to pick $k$ s.t. $c\beta_k k \log(k) \leq n$, then the two lemmas will simultaneously hold w.p. at least $1 - \delta - 16\exp(-\frac{c'}{\beta_k^2}\frac{n}{k})$. Since $\beta_k = o(1)$ we know it can be upper bounded by $C_0$ for some $C_0 > 0$. Similar to Barzilai & Shamir (2023), we let $k := k(n) := \frac{n}{\max\{cC0,1\}\log n}$ and we also let $k' := k'(n) = n^2 \log^4(n)$. So the probability of those theorems hold become $1 - \delta - O(\frac{1}{n})$.

In this case, $\rho_{k,n}$ cannot be regularized to $o(1)$ if the features are not well-behaved, we compute its bound first, which requires bounding $\mu_1(\frac{1}{n}\tilde{K}_{>k})$ and $\mu_n(\frac{1}{n}\tilde{K}_{>k})$ respectively.

We apply Lemma C.6 by setting $\delta = \log n$, then w.p. $1 - \frac{1}{\log(n)}$ we have

$$\mu_n(\frac{1}{n}\tilde{K}_{>k}) \geq \alpha_k(1 - \frac{1}{\log n}\sqrt{\frac{n^2}{\operatorname{tr}(\tilde{\Sigma}_{>k'})^2/\operatorname{tr}(\tilde{\Sigma}_{>k'}^2)}})\frac{\operatorname{tr}(\tilde{\Sigma}_{>k'})}{n}$$

$$= \Omega((1 - \log n\sqrt{\frac{1}{\log^4 n}})\frac{\operatorname{tr}(\tilde{\Sigma}_{>k'})}{n})$$

$$= \Omega(\frac{(k')^{1-2p-\beta\lambda}}{n})$$

$$= \Omega(\frac{(n^2 \log^4 n)^{1-2p-\beta\lambda}}{n})$$

$$= \tilde{\Omega}(n^{1-4p-2\beta\lambda}).$$

Note that the first equality is because we have $\operatorname{tr}(\tilde{\Sigma}_{>k'})^2/\operatorname{tr}(\tilde{\Sigma}_{>k'}^2) = \frac{(\sum_{i>k'} p_i^2 \lambda_i^\beta)^2}{\sum_{i>k'} p_i^4 \lambda_i^{2\beta}} = \frac{k'^{2-2p-\lambda\beta}}{k'^{1-2p-\lambda\beta}} = k' = n^2 \log^4(n)$, $\tilde{\Omega}$ means we neglect logarithmic terms.

For $\mu_1(\frac{1}{n}\tilde{K}_{>k})$ term by Lemma F.6, we have w.p. $1 - O(\frac{1}{k^3})\exp(-\Omega(\frac{n}{k}))$

$$\mu_1(\frac{1}{n}\tilde{K}_{>k}) = O(p_{k+1}^2 \lambda_{k+1}^\beta) = O(k^{-2p-\beta\lambda}) = \tilde{O}(n^{-2p-\beta\lambda}). \tag{9}$$

Using the bound of $\mu_1(\frac{1}{n}\tilde{K}_{>k})$ and $\mu_n(\frac{1}{n}\tilde{K}_{>k})$, we have $\rho_{k,n} = \tilde{O}(n^{2p+\beta\lambda-1})$.

At the same time, we have Eq. 9, Lemma C.6, Theorem D.3, E.3 all hold simultaneously hold with probability $1 - \delta - O(\frac{1}{\log(n)})$.

Recall from Lemma D.3 that

$$V \leq C_1 \sigma_\varepsilon^2 \rho_{k,n}^2 \Big( \frac{\sum_{i \leq k} p_i^{-2} \lambda_i^{-\beta'}}{n} + \frac{\sum_{i>k} p_i^2 \lambda_i^{-\beta'+2\beta}}{n \|\tilde{\Sigma}_{>k}\|^2} \Big)$$

$$= \sigma_\varepsilon^2 \rho_{k,n}^2 O\Big( \frac{\max\{k^{1+2p+\lambda\beta'}, 1\}}{n} + \frac{k^{1-2p+\lambda(\beta'-2\beta)}}{nk^{-2\beta\lambda-4p}} \Big)$$

$$= \sigma_\varepsilon^2 \rho_{k,n}^2 \tilde{O}\Big( \frac{\max\{k^{1+2p+\lambda\beta'}, 1\}}{n} \Big).$$

So we sub. $k = \tilde{\Theta}(n)$ and the final bound of variance is

$$V \leq \sigma_\varepsilon^2 \rho_{k,n}^2 \tilde{O}(n^{\max\{2p+\lambda\beta', -1\}}).$$

For bias, similar to the regularized case, the bound is

$$\frac{1}{\delta} \rho_{k,n}^3 O(k^{\max\{1-2r'-\lambda(1-\beta'), -2p+\lambda(\beta'-2\beta)\}}).$$

The main difference is the choice of $k$, since $k = \tilde{\Theta}(n)$, the final bound is

$$\frac{1}{\delta} \rho_{k,n}^3 O(n^{\max\{1-2r'-\lambda(1-\beta'), -2p+\lambda(\beta'-2\beta)\}}).$$

Note that if the features are well-behaved, then $\rho_{k,n}$ can be improved to $o(1)$. $\qquad \square$

### F.3  LEMMAS FOR SUBSTITUTING POLYNOMIAL DECAY

**Lemma F.3.** *Let $a \in \mathbb{R}$, $1 < k \in \mathbb{N}$, then*

$$\sum_{i \leq k} i^{-a} \leq \begin{cases} 1 + \frac{1}{1-a} k^{1-a} & a < 1 \\ 1 + \log(k) & a = 1 \\ 1 + \frac{1}{a-1} & a > 1. \end{cases}$$

*Therefore, $\sum_{i \leq k} i^{-a} = \tilde{O}(\max\{k^{-a+1}, 1\})$*

*Proof.* We know that, for $a < 1$

$$\sum_{i \leq k} i^{-a} \leq 1 + \int_1^k x^{-a}\, dx = 1 + \frac{1}{1-a}(k^{1-a} - 1) \leq 1 + \frac{1}{1-a} k^{1-a}.$$

For $a = 1$

$$\sum_{i \leq k} i^{-a} \leq 1 + \int_1^k x^{-a}\, dx = 1 + \log(k).$$

For $a > 1$

$$\sum_{i \leq k} i^{-a} \leq 1 + \int_1^\infty x^{-a}\, dx = 1 + \frac{1}{a-1}.$$

$\qquad \square$

**Lemma F.4.** *Let $a \in \mathbb{R}$, $1 < k \in \mathbb{N}$, then*

$$\sum_{i>k} i^{-a} \in \begin{cases} \infty & a \leq 1 \\ [\frac{1}{a-1}(k+1)^{-a+1}, (k+1)^{-a} + \frac{1}{a-1}(k+1)^{-a+1}] & a > 1. \end{cases}$$

*Therefore, $\sum_{i>k} i^{-a}$ is $O(k^{-a+1})$ if $a > 1$, otherwise it diverges to infinity*

*Proof.* We know that,

$$\int_{k+1}^\infty x^{-a}\, dx \leq \sum_{i>k} i^{-a} \leq (k+1)^{-a} + \int_{k+1}^\infty x^{-a}\, dx.$$

If $a < 1$ then $\int_{k+1}^\infty x^{-a} = \infty$ which implies the series diverge, otherwise, $\int_{k+1}^\infty x^{-a} = \frac{1}{a+1}(k+1)^{-a+1}$

$\qquad \square$

**Lemma F.5.** *Assume $[\phi f^*]_i = \Theta(i^{-r'})$, $\Sigma$'s polynomial decaying eigenvalues satisfy $\lambda_i = \Theta(i^{-\lambda})$ ($\lambda > 0$), and $\mathcal{A}$'s eigenvalue is $\Theta(i^{-p})$ ($p < 0$), then*

$$\|\phi_{>k}\mathcal{A}_{>k}f^*_{>k}\|^2_{\Lambda^{\geq k}_{\Sigma}} = \Theta\left(\frac{1}{k^{2p+2r'+\lambda-1}}\right) \ if \ 2p + 2r' + \lambda > 1;$$

$$\|\phi_{\leq k}f^*_{\leq k}\|^2_{\Lambda^{\leq k}_{\mathcal{A}^{-2}\Sigma^{1-2\beta}}} = \tilde{O}(\max\{k^{1+2p-\lambda(1-2\beta)-2r'}, 1\});$$

$$\|\phi_{>k}f^*_{>k}\|^2_{\Lambda^{>k}_{\Sigma^{1-\beta'}}} = \Theta\left(\frac{1}{k^{2r'+(1-\beta')\lambda-1}}\right) \ if \ 2r' + (1-\beta')\lambda > 1,$$

*where $r' = \frac{1-\lambda(1-r)}{2}$.*

*Proof.* We know from F.4 that,

$$\|\phi_{>k}\mathcal{A}_{>k}f^*_{>k}\|^2_{\Lambda^{\geq k}_{\Sigma}} = \sum_{i>k}[\phi f^*]^2_i \cdot p^2_i\lambda_i = \sum_{i>k}\Theta\left(\frac{1}{i^{2p+2r'+\lambda}}\right) = \Theta\left(\frac{1}{k^{2p+2r'+\lambda-1}}\right) \ if \ 2p+2r'+\lambda > 1.$$

Similarly, using F.3

$$\|\phi_{\leq k}f^*_{\leq k}\|^2_{\Lambda^{\leq k}_{\mathcal{A}^{-2}\Sigma^{1-2\beta}}} = \sum_{i\leq k}[\phi f^*]^2_i \cdot p^2_i\lambda^{1-2\beta}_i = \sum_{i\leq k}\Theta\left(\frac{1}{i^{2r'-2p+\lambda(2\beta-1)}}\right) = \tilde{O}(\max\{k^{1+2p-\lambda(1-2\beta)-2r'}, 1\}).$$

Using F.4 again, we'll have

$$\|\phi_{>k}f^*_{>k}\|^2_{\Lambda^{>k}_{\Sigma^{1-\beta'}}} = \sum_{i>k}[\phi f^*]^2_i \cdot \lambda^{\beta'-1}_i = \sum_{i>k}\Theta\left(\frac{1}{i^{2r'+(1-\beta')\lambda}}\right) = \Theta\left(\frac{1}{k^{2r'+(1-\beta')\lambda-1}}\right) \ if \ 2r'+(1-\beta')\lambda > 1.$$

$\square$

**Lemma F.6.** *Assume $\Sigma$'s polynomial decaying eigenvalues satisfy $\lambda_i = \Theta(i^{-\lambda})$ ($\lambda > 0$), and $\mathcal{A}$'s eigenvalue is $\Theta(i^{-p})$. And we suppose $\frac{\beta_k k \log(k)}{n} = o(1), \beta_k = o(1)$.*

*Then it holds w.p. at least $1 - O(\frac{1}{k^3})\exp(-\Omega(\frac{n}{k}))$ that*

$$\mu_1(\frac{1}{n}\tilde{K}_{>k}) = O(\lambda^{\beta}_{k+1}p^2_{k+1}) = O(k^{-2p-\beta\lambda}).$$

*Proof.* We use C.6 then there exists absolute constant $c, c' > 0$ s.t. it holds w.p. at least $1 - 4\frac{r_k}{k^4}\exp(-\frac{c'}{\beta_k}\frac{n}{r_k})$ that

$$\mu_1(\frac{1}{n}\tilde{K}_{>k}) \leq c(\lambda^{\beta}_{k+1}p^2_{k+1} + \beta_k \log(k+1)\frac{\text{tr}(\tilde{\Sigma}_{>k})}{n})$$

$$= O(\lambda^{\beta}_{k+1}p^2_{k+1}(1 + \beta_k \log(k+1)\frac{k}{n}))$$

$$= O(\lambda^{\beta}_{k+1}p^2_{k+1}),$$

where $\tilde{\Sigma} := \mathcal{A}^2\Sigma^{\beta}$, $r_k := \frac{\text{tr}(\tilde{\Sigma}_{>k})}{p^2_{k+1}\lambda^{\beta}_{k+1}}$.
The last inequality is because $\frac{k\log(k+1)}{n} = o(1)$.

Now we bound the probability of this holds, we can derive $r_k = \frac{k^{1-2p-\lambda\beta}}{(k+1)^{-2p-\lambda\beta}} = \Theta(k)$, $1 - 4\frac{r_k}{k^4}\exp(\frac{-c'}{\beta_k}\frac{n}{r_k}) = 1 - O(\frac{1}{k^3})\exp(-\Omega(\frac{n}{k}))$. $\square$

# G IMPLEMENTATION DETAILS OF EXPERIMENTS

we consider the Poisson equation $u = \Delta f$ on $\Omega = [0,2]^2$ with Dirichlet boundary condition on $\partial\Omega$, where the ground truth $f(x_1, x_2) = \sin(\pi x_1)\sin(\pi x_2)$, where the data points $\{(x_i, y_i)\}^n_{i=1}$ are sampled uniformly from $\Omega$, and $y_i = \Delta f(x_i) + \varepsilon$ with $\varepsilon \sim \mathcal{N}(0, \sigma^2)$. The training loss function is $\min_\theta \hat{L}(\theta) := \frac{1}{n}\sum^n_{i=1}(\Delta \hat{f}(x_i; \theta) - y_i)^2$. To satisfy the boundary condition, we enforce $\hat{f}(x) = x_1(x_1 - 2)x_2(x_2 - 2)f_{\text{NN}}(x)$, where $f_{\text{NN}}$ is the neural network (Liang et al., 2021). For clean

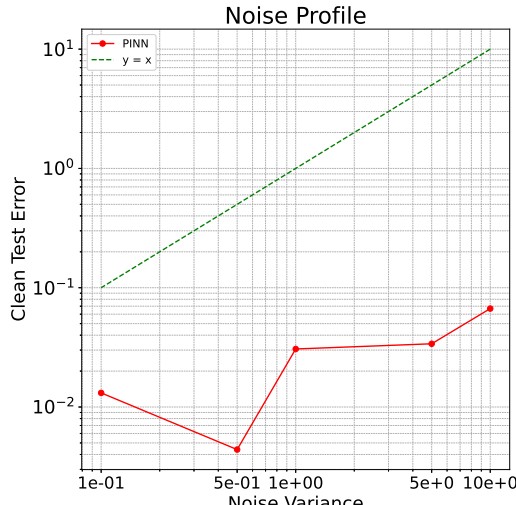

Figure 2: We again verified our findings using PDE with solution of low regularity at the origin. The noise profile of Physics-informed interpolator exhibits benign overfitting, unlike the regression interpolator.

test loss, we use $\frac{1}{n} \sum_{i=1}^{n} (\hat{f}(x_i, \theta) - f(x_i))^2$ to match the definition of excess risk, where $\{(x_i, y_i)\}$ is re-sampled from $\Omega$.

In all experiments, we use Adam optimizer with learning rate 5e-3 for regression problem, and 1e-4 for PINN problem where both are optimally tuned. Weight decay is set as 1e-4, and learning rate schedule is StepLR with step size 3000 and gamma 0.8. In both experiments we train for 100000 iterations to allow convergence. All models considered are sufficiently over-parametrized.

For the experiment verifying the effect of smoothness of the inductive bias, we uses the one-layer wide neural network with width 10000 (we choose one-layer here to avoid explosion of output due to ReLU$^4$), and vary different activation functions ReLU,ReLU$^2$,ReLU$^3$ and ReLU$^4$. Noise level $\sigma^2$ is set as 0.1. We vary sample size 50, 100, 500, 1000 and plot the convergence rate using different activation functions.

For the experiment verifying benign over-fitting of Physics-Informed interpolator, we train sufficient iterations to ensure interpolation into the noise. The used learning model here is a two-layer wide neural network with hidden size 1024, with sample size 500, using ReLU as activation function. We vary noise variance 1e-1, 3e-1, 5e-1, 1e+0, 3e+0, 5e+0, and plot the clean test loss against noise variance.

For the figure of visualizing landscape, we use a two-layer wide neural network with hidden size 1024, with sample size 500, using ReLU as activation function and with noise variance 5 and train it until it interpolates into the noise. We using the 100x100 grid on $[0, 2]^2$ to display landscape of ground truth $f$ and model output $\hat{f}$, also we display $\Delta f$ and $\Delta \hat{f}$, where red dots are the training set points.

**Verifying the Benign Overfitting Beyond Co-diagonalization Assumption**   We provide additional experiments on the PDE

$$-\nabla \cdot (|x| \nabla u) = f \quad \text{for } x \in \Omega \text{ and } u = 0 \text{ for } x \in \partial\Omega$$

where the commutative assumption no longer holds. Our result demonstrates that it still verifies our two findings. Here we consider solving a solution $u(x) = \sin(2\pi(1 - |x|))$ defined on $\Omega = \{x : |x| < 1\}$. $\hat{u}(x; \theta) = (1 - |x|) u_{\text{NN}}(x; \theta)$ to automatically satisfy the boundary condition, where $u_{\text{NN}}$ is the neural network. We maintain the same configurations as previous experiments.

