# OpenReview forum: "Physics-Informed Interpolator Generalizes Well in Fixed Dimension: Inductive Bias and Benign Overfitting"
_ICLR.cc/2025/Conference — Submitted to ICLR 2025_

### Official Review · Reviewer_fsKN · 2024-11-02

**Soundness:** 2
**Presentation:** 3
**Contribution:** 2
**Rating:** 5
**Confidence:** 2

**Summary:**

This paper examines the statistical performance of kernel-based methods for linear elliptic Partial Differential Equation (PDE) inverse problems. For the regularized least squares estimator with a reproducing kernel Sobolev space (RKSS) norm as the penalty term, and the minimum norm interpolation estimator, this paper derives learning curves in terms of the RKSS norm for both estimators. Utilizing the bias-variance decomposition from an explicit representation theorem of the estimators, the paper demonstrates that elliptic operators, under certain capacity assumptions, can exhibit benign-overfitting behavior for fixed-dimensional problems. Furthermore, the results indicate that all regularized least squares estimators, with appropriately chosen regularization parameters, can achieve an optimal convergence rate. This convergence rate is independent of the choice of inductive bias, provided that the inductive bias is sufficiently smooth.

**Strengths:**

This paper introduces novel methods to search for solutions of PDE problems. Distinct from prior works, this paper uses RKSS norm as penalty term instead of RKHS norm, and demonstrate firm proofs for the convergence rate of the models. Impressively, the convergence rate is actually optimal in certain circumstances. The result of independence of convergence rate on inductive bias also provides guidence to determine suitable kernel functions, inductive bias and regularization parameters.

**Weaknesses:**

The assumptions adopted in this paper are strong, especially the capacity assumption (Assumption 2.2(d)). This paper needs to demonstrate more detailed and nontrivial examples where the assumptions are satisfied and the models are computed explicitly.
Besides, both the regularized least square loss function in the problem settings (1)(2) and the representer provided in Lemma 3.1 are involved with the inductive bias and the RKSS norm, hence the numerical computations are neither trivial nor straightforward. This paper lacks numerical experiments to demonstrate the practical performance of the algorithms mentioned.

**Questions:**

First, I prefer to call the phenomena shown in the main theorems (Theorem 4.1 and Theorem 4.2) ‘generalization ability’ of the models, and I am confused with the term ‘benigning overfitting’ in the title. Can you explain more how the models are overfitting?
Besides, the capacity condition on elliptic operators (Assumption 2.2(d)) seems too strong, making the problem setting of regularized least square (1) very similar with the setting of kernel ridge regression. Except the RKSS norm penalty instead of RKHS norm penalty, is there any other essential difference between kernel ridge regression and the regulized least square model in this paper?

---

> ### Author Response · Authors · 2024-11-24
>
> We sincerely thank reviewer fsKN for their efforts and their acknowledgement of the strengths of our paper.
>
> > Regards Empirical Validation
>
> We added two numerical experiments to verify the two main discoveries in our paper. We run a PINN to solve the Poisson equation $u = \Delta f$ with zero Dirichlet  boundary condition. We trained neural network with Relu, $\text{Relu}^2$, $\text{Relu}^3$, $\text{Relu}^4$activation to test how the inductive bias of smoothness changes the convergence. As a smoother activation function is used,  we expect the resulting NTK to have faster decay [1]. Our results show that once the activation function is smooth enough, increasing smoothness would no longer help matches our theoretical claim. Secondly, we plotted the noise profile (how the asymptotic risk varies with the variance of additive Gaussian noise) following [2] to examine the beginning overfitting effect.  We show that the standard regression interpolator performs worse under stronger noise level. Instead, the test risk of the Physics-Informed interpolator does not change too much at various noise levels.
>
> Our intuition behind the benign overfitting behavior for inverse problems differs from that of plain regression because we are predicting \( \Delta^{-1} u \), not \( u \) itself. The operator \( \Delta^{-1} \) acts like a kernel smoothing operation (with the Green's function as the kernel), which smooths the predictions and underweights errors in the high-frequency components—these being the main contributors to the overall error. For general PDEs governing physical laws, most behave like differential operators, where the forward problem amplifies high-frequency components. Consequently, solving the inverse problem tends to attenuate these high-frequency components, resulting in a similar noise stabilization effect. This is what we called the “variance stabilisation” effect in our main text. We refined our main text to make the explanation clearer.
>
> We have already updated our submission, see Section 5 for details.
>
> We believe our new numerical experiments will extend our findings beyond kernel methods, offering valuable insights and practical guidance to PINN users. We hope the reviewer can reevaluate our paper accordingly. If the reviewer has any further concerns, we would be happy to address them.
>
> > Regards co-diagonal assumption
>
>
> We admit that the co-diagonal assumption is strong. However, when data is uniform distribution over the sphere (which is a common case in PINN learning since the data is often uniformly sampled from the domain), the co-diagonal assumption holds for many differential operators, e.g. the laplace operator $\Delta$ and $\Delta^k$.
>
>  **To our best knowledge, this remains an open question of how to avoid such assumptions.** So using such assumptions is widely used in the literature, including references [7, 8, 10] for inverse problems, [9] for operator learning, and [11] for pre-training and fine-tuning.  We added a numerical experiment to see if our insights still hold beyond the assumption. See our answer in regards to numerical experiments.

---

> ### Author Response · Authors · 2024-11-24
>
> > Regards computation in RKSS
>
> The RKSS model we employ here is not chosen for computational reasons but is motivated by two primary considerations for analyzing RKSS rather than RKHS. First, a substantial body of literature, from [3] to [4], has explored the use of spectrally transformed (smoother) kernels and suggested that they can enhance learning. However, none of these works have theoretically examined whether these methods improve the learning rate or quantified the extent of such improvement. Second, we use this model as a theoretical framework to understand how the inductive bias in (physics-informed) learning should be designed. Our findings reveal a non-trivial result: while smoothing can aid convergence, there is a point where a sufficiently smooth inductive bias leads to optimal convergence, beyond which further smoothing provides no additional benefit. To the best of our knowledge, this is the first work to theoretically quantify the benefits in terms of convergence rate of a smooth inductive bias.
>
>
> > Regards benign overfitting
>
> Benign overfitting is a terminology refers to the phenomenon where an over-parameterized interpolating estimator achieves non-trivial generalization [5], [6]. Traditionally, if an estimator can interpolate any data, it cannot generalize. In our paper, we analyze interpolating estimators and demonstrate a benign overfitting result in fixed dimensions for physics-informed learning. This behavior, which contrasts with typical regression scenarios, highlights a distinct advantage of physics-informed learning.
>
> -----
>
> We believe our new numerical experiments verifies our findings and  extend them beyond kernel methods, offering valuable insights and practical guidance to PINN users. We hope reviewer can reevaluate our paper accordingly. If the reviewer has any further concerns, we would be happy to address them.
>
> ----
>
> [1] Bietti A, Bach F. Deep equals shallow for ReLU networks in kernel regimes. arXiv preprint arXiv:2009.14397, 2020.
>
> [2] Mallinar N, Simon J, Abedsoltan A, et al. Benign, tempered, or catastrophic: Toward a refined taxonomy of overfitting. Advances in Neural Information Processing Systems, 2022, 35: 1182-1195.
>
> [3] Zhou X, Belkin M. Semi-supervised learning by higher order regularization AISTATS. 2011.
>
> [4] Zhai R, Pukdee R, Jin R, et al. Spectrally Transformed Kernel Regression. ICLR 2024 oral
>
> [5] Bartlett P L, Long P M, Lugosi G, et al. Benign overfitting in linear regression[J]. Proceedings of the National Academy of Sciences, 2020, 117(48): 30063-30070.
>
> [6] Mallinar N, Simon J, Abedsoltan A, et al. Benign, tempered, or catastrophic: Toward a refined taxonomy of overfitting[J]. Advances in Neural Information Processing Systems, 2022, 35: 1182-1195.
>
> [7] Bartek T Knapik, Aad W Van Der Vaart, and J Harry van Zanten. Bayesian inverse problems with gaussian priors. 2011.
>
> [8] Vivien Cabannes, Loucas Pillaud-Vivien, Francis Bach, and Alessandro Rudi. Overcoming the curse of dimensionality with laplacian regularization in semi-supervised learning. In Thirty-Fifth Conference on Neural Information Processing Systems, 2021.
>
> [9] Maarten V de Hoop, Nikola B Kovachki, Nicholas H Nelsen, and Andrew M Stuart. Convergence rates for learning linear operators from noisy data. arXiv preprint arXiv:2108.12515, 2021.
>
> [10] Yiping Lu, Jose Blanchet, and Lexing Ying. Sobolev acceleration and statistical optimality for learning elliptic equations via gradient descent. Advances in Neural Information Processing Systems, 35:33233–33247, 2022.
>
> [11] Wu J, Zou D, Braverman V, Gu Q and Kakade S. The power and limitation of pretraining-finetuning for linear regression under covariate shift. Advances in Neural Information Processing Systems, 2022, 35: 33041-33053.

---

> ### Author Response · Authors · 2024-11-25
>
> As the discussion deadline is tomorrow, we have already provided **empirical validation** and **explanations on co-diagonalization assumption** on our theory, we also **add explanations on "benign overfitting" and RKSS**. We will be more than happy to provide additional clarifications if you have any further specific technical questions that still concern you. We hope our response has addressed your concerns and we would be grateful if you would consider reevaluating your assessment of our work.

---

### Official Review · Reviewer_zUQ1 · 2024-11-03

**Soundness:** 4
**Presentation:** 2
**Contribution:** 3
**Rating:** 6
**Confidence:** 3

**Summary:**

The authors show that certain design of norms (called Kernel Soboleve norm) can lead to more stability of the variance and even benign overfitting for fixed-dimension problems. (Previous work suggests that common kernels might need high dimensionality to do that) Furthermore, they show that inductive biases don't affect convergence rate, with optimal achieved with appropriate parameter chosen. Also some surprising connection in Bayesian settings are discovered.

**Strengths:**

Paper solid. Handles sophisticated mathematical theories.

The authors show a refreshing story that even for fixed dimension one can tune the kernel to achieve benign overfitting. This complements the storyline of kernel ridgeless ridge very well.

The authors present rich results with depth on the subject.

**Weaknesses:**

The title "Physics-Informed" is not 100% appropriate. Physics seems to be indirectly connected to the theme of the paper. Physics -> PDE -> Sobolev norm -> Kernel Sobolev norm. Sobolev is quite a general notion, a natural choice for quantifying smoothness. Many different settings could lead to Sobolev, not just Physics. It would have been more justified, if many empirical experiments related to Physics were done in the paper. For me personally, "Physics-informed" is misleading and gave the wrong impression before I fully read the paper.

The essence of the proof and theory is not explained clearly. Essence is the core mechanism that makes a difference, an intuitive explanation of which makes it easy to understand why things work before eating all technical details and to credit the paper for its theoretical contribution. The authors show a phenomenon very different from previous work on the subject. Why the kernel proposed in the paper can work for benign overfitters even for fix dimensionality? My understanding is that, the kernels in the paper have more flexible spectrums, making it possible to simultaneously making both biases and variances small.

The authors show that "the choice of smooth enough inductive bias also does not affect convergence speed". To make this claim sounder, some empirical experiments are needed. Otherwise, it looks like it might just be theories are too intractable to lead to a difference.

Some statements are confusing. It's said with emphasis that "Our results show that the PDE operators in inverse problems possess the capability to stabilize variance and remarkably behave benign overfitting ...". However Gaussian kernels are the inverse of laplacian operators and previous work has proven Gaussian kernels for any bandwidth has a nonzero lower bound on error rate for fixed dimension. And it's natural to think Laplacian operators as suitable for inverse problems (although I'm not familiar with that). It would be better if there are more explanation on the difference between these kernels and previous ones.

**Questions:**

Included in weakness.

---

> ### Author Response · Authors · 2024-11-24
>
> We sincerely thank reviewer zUQ1 for their efforts and their acknowledgement of the strengths of our paper.
>
> > Regards Empirical Validation
>
> We added two numerical experiments to verify the two main discoveries in our paper. We run a PINN to solve the Poisson equation $u = \Delta f$ with zero Dirichlet  boundary condition. We trained neural network with Relu, $\text{Relu}^2$, $\text{Relu}^3$, $\text{Relu}^4$activation to test how the inductive bias of smoothness changes the convergence. As a smoother activation function is used,  we expect the resulting NTK to have faster decay [1]. Our results show that once the activation function is smooth enough, increasing smoothness would no longer help matches our theoretical claim. Secondly, we plotted the noise profile (how the asymptotic risk varies with the variance of additive Gaussian noise) following [2] to examine the beginning overfitting effect.  We show that the standard regression interpolator performs worse under stronger noise level. Instead, the test risk of the Physics-Informed interpolator does not change too much at various noise levels.
>
> Our intuition behind the benign overfitting behavior for inverse problems differs from that of plain regression because we are predicting \( \Delta^{-1} u \), not \( u \) itself. The operator \( \Delta^{-1} \) acts like a kernel smoothing operation (with the Green's function as the kernel), which smooths the predictions and underweights errors in the high-frequency components—these being the main contributors to the overall error. For general PDEs governing physical laws, most behave like differential operators, where the forward problem amplifies high-frequency components. Consequently, solving the inverse problem tends to attenuate these high-frequency components, resulting in a similar noise stabilization effect. This is what we called the “variance stabilisation” effect in our main text. We refined our main text to make the explanation clearer.
>
> We have already updated our submission, see Section 5 for details.
>
>
> > Regards Physics-Informed in the Title
>
> We fully understand the confusion made by world physics-informed, not all the physics leads to a PDE. We use the word “physics-inform” that follows previous literature [3]: using neural network to solve PDE is named as physics-informed machine learning.
>
>
> > Regards intuition behind the theory
>
> The flexibility of selecting the kernel spectrum is well-established, as demonstrated in previous works [1,2]. Initially, we shared this perspective. However, after conducting detailed theoretical computations, our findings were surprising: unless the kernel spectrum decays sufficiently fast (indicating smoothness in the spatial domain), it does not influence the convergence rate. Consequently, we treat the problem's hardness and the classifier's capacity as flexible parameters and complete the convergence rate analysis accordingly.
>
> After we complete the whole theory, we find out that the intuition behind the benign overfitting behavior for inverse problems differs from that of plain regression because we are predicting \( \Delta^{-1} u \), not \( u \) itself. The operator \( \Delta^{-1} \) acts like a kernel smoothing operation (with the Green's function as the kernel), which smooths the predictions and underweights errors in the high-frequency components—these being the main contributors to the overall error. For general PDEs governing physical laws, most behave like differential operators, where the forward problem amplifies high-frequency components. Consequently, solving the inverse problem tends to attenuate these high-frequency components, resulting in a similar noise stabilization effect. This is what we called the “variance stabilization” effect in our main text and this is exactly the benefit brought by the PDE (the physics) but not brought by the flexible design of the kernel. We refined our main text to make the explanation clearer.
>
> We have already updated our submission, see Section 5 for details.
>
> ----
>
> We believe our new numerical experiments verify our finds and extend our findings beyond kernel methods, offering valuable insights and practical guidance to PINN users. We hope the reviewer can reevaluate our paper accordingly. If the reviewer has any further concerns, we would be happy to address them.
>
> ----
>
>
> [1] Bietti A, Bach F. Deep equals shallow for ReLU networks in kernel regimes. arXiv preprint arXiv:2009.14397, 2020.
>
> [2] Mallinar N, Simon J, Abedsoltan A, et al. Benign, tempered, or catastrophic: Toward a refined taxonomy of overfitting. Advances in Neural Information Processing Systems, 2022, 35: 1182-1195.
>
> [3] Karniadakis G E, Kevrekidis I G, Lu L, et al. Physics-informed machine learning. Nature Reviews Physics, 2021, 3(6): 422-440.

---

> ### Author Response · Authors · 2024-11-25
>
> As the discussion deadline is tomorrow, we have already provided **empirical validation** and **intuitive explanations** on our theory, we will be more than happy to provide additional clarifications if you have any further specific technical questions that still concern you. We hope our response has addressed your concerns and we would be grateful if you would consider reevaluating your assessment of our work.

---

> ### Author Response · Authors · 2024-11-30
>
> We hope that our previous response has fully addressed your concerns, and as the discussion deadline is fastly approaching, we are looking forward to answering questions if the reviewer has other concerns about the paper. We hope that the reviewer can reevaluate our work and adjust the score.

---

### Official Review · Reviewer_eSS5 · 2024-11-03

**Soundness:** 3
**Presentation:** 3
**Contribution:** 2
**Rating:** 5
**Confidence:** 3

**Summary:**

This paper studies performance of solving inverse problems using kernel based methods. Considering the least square problem regularized by kernel Sobolev norm or the interpolator, under certain assumptions, this paper shows the generalization error in terms of bias-variance decomposition. This paper further discusses the implications of the results that how the inductive bias from the chosen kernel Sobolev space can affect the generalization.

**Strengths:**

- This paper is well-organized and accessible, with clear presentation of the main results even for those unfamiliar with complex notations and quantities.
- This paper studies the solution of inverse problems using least squares regression in RKHS, establishing upper bounds. This contributes to the theory of physics-informed machine learning.

- The finding that "the PDE operator in the inverse problem enables benign overfitting even in fixed-dimensional settings" is interesting and provides new insights into physics-informed machine learning.

**Weaknesses:**

- Problem Setting: The paper does not consider boundary conditions, which are essential for inverse problems, which limits the applicability and the implications of the results.

- Strong Assumptions: Assumption 2.2 (d) is particularly strong, requiring the kernel's orthonormal basis to diagonalize the operator.  Additionally, with operator  diagonalized by the kernel's basis and without boundary conditions, the analysis closely follows that of kernel ridge regression as presented in Barzilai & Shamir (2023), making the techniques used less novel.

- Lack of Empirical Validation: There is no empirical validation to verify the effectiveness of the method. Experiments are necessary to convincingly demonstrate the practical utility of the proposed approach across different types of inverse problems, including those with boundary conditions or in high-dimensional settings.

**Questions:**

1. In Theorem 4.2, there is a term $O(1/\log n)$ in the probability bound, making it almost vanishing. Can this bound be improved?

2. I would like the authors to discuss in detail how $ \rho_{k,n}$ behaves in concrete and realistic settings. It seems that I cannot find a thorough analysis in the appendix as suggested in Remark 6.

3. For real-world applications, how can we choose the hyperparameters in the proposed method empirically?

4. How does the dimensionality of $\mathcal{X}$ affect the results, particularly in terms of generalization and variance stabilization?

---

> ### Author Response · Authors · 2024-11-24
>
> We sincerely thank reviewers eSS5 for their efforts and their acknowledgement of the strengths of our paper.
>
> > Regards Empirical Validation
>
> We added two numerical experiments to verify the two main discoveries in our paper. We run a PINN to solve the Poisson equation $u = \Delta f$ with zero Dirichlet  boundary condition. We trained neural network with Relu, $\text{Relu}^2$, $\text{Relu}^3$, $\text{Relu}^4$activation to test how the inductive bias of smoothness changes the convergence. As a smoother activation function is used,  we expect the resulting NTK to have faster decay [1]. Our results show that once the activation function is smooth enough, increasing smoothness would no longer help matches our theoretical claim. Secondly, we plotted the noise profile (how the asymptotic risk varies with the variance of additive Gaussian noise) following [2] to examine the beginning overfitting effect.  We show that the standard regression interpolator performs worse under stronger noise level. Instead, the test risk of the Physics-Informed interpolator does not change too much at various noise levels.
>
> Our intuition behind the benign overfitting behavior for inverse problems differs from that of plain regression because we are predicting ( $\Delta^{-1} u$ ), not \( u \) itself. The operator \( $\Delta^{-1}$ \) acts like a kernel smoothing operation (with the Green's function as the kernel), which smooths the predictions and underweights errors in the high-frequency components—these being the main contributors to the overall error. For general PDEs governing physical laws, most behave like differential operators, where the forward problem amplifies high-frequency components. Consequently, solving the inverse problem tends to attenuate these high-frequency components, resulting in a similar noise stabilization effect. This is what we called the “variance stabilisation” effect in our main text. We refined our main text to make the explanation clearer.
>
> We have already updated our submission, see Section 5 for details.
>
>
>
> > Regards boundary condition
>
> Boundary conditions present significant challenges for incorporation into the kernel framework, with recent breakthrough work [3] being one of the few known approaches. Additionally, they pose difficulties for achieving optimal statistical analysis [4]. Despite their importance and inherent interest, addressing boundary conditions within the kernel framework and their statistical analysis is beyond the scope of this work and is left as a direction for future research.
>
> > Regarding the co-diagonalization assumption
>
>
> We admit that the co-diagonal assumption is strong. However, when data is the uniform distribution over the sphere (which is a common case in PINN learning since the data is often uniformly sampled from the domain), the co-diagonal assumption holds for many differential operators, e.g. the laplace operator $\Delta$ and $\Delta^k$.
>
>  **To our knowledge, this remains an open question of how to avoid such assumptions.** So using such assumptions is widely used in the literature, including references [7, 8, 10] for inverse problems, [9] for operator learning, and [11] for pre-training and fine-tuning.  We added a numerical experiment to see if our insights still hold beyond the assumption. See our answer in regard to numerical experiments.

---

> ### Author Response · Authors · 2024-11-24
>
> >  Regards the probability $O(1/\log n)$ in Theorem 4.2
>
> The probability we achieve is weaker than [1] but comparable to [2]. The key distinction lies in the assumptions about the feature map: [1] assumes the feature map is sub-Gaussian, while neither [2] nor our analysis relies on this assumption. Consequently, detailed analysis of the self-regularization effect  $\mu_n(K_{\ge k})$ is needed.  The $\log(n)$ probability would appear when lower bounding the self-regularization effect  $\mu_n(K_{\ge k})$. The probability used here is tight in our proof.
>
>
> > Regards the behavior of $\rho_{k,n}$
>
> For well-behaved sub-Gaussian features, the concentration coefficients $\rho_{k,n} = \Theta(1)$ as shown in [5] while in our appendix we also proivde a worst case bound for the concertration coefficeints at $\tilde{O}(n^{2p + \beta \lambda - 1})$. It is shwon in (line1974-1997). We bounded both $\mu_1$ and $\mu_n$ which leads to the estimation of $\rho_{k,n}$
>
>
> > Regards the choice of the hyper-parameters
>
> The eigendecay of the kernel is dependent on the dimensionality of X (see below) and the parameter $r$ is dependent on the order of PDE. The parameter $\beta'$, it is determined based on the user's specific requirements regarding the desired quality of the learned solution, particularly in relation to whether gradient convergence is necessary for the downstream application. For the function smoothness, it's hard to determine hyperparameters, however, our theory showed that a fixed algorithm can be optimal to all function smoothness!
>
>
> > Regards the input dimension of $X$
>
> The dimensionality of X is hidden in the eigendecay speed of the kernel. The eigen decay speed is typically proportion to 1/d and leads to dimensionality dependency. To avoid the curse of dimensionality, one needs further smoothness assumption, which relates to our parameter $r$. We need a smoothness $r$ proportion to the dimension $d$ aligned with the kernel literature, e.g. [6].
>
> > Novelty of Our Analysis
>
> Our analysis primarily reweights different eigenspaces, particularly under the co-diagonal assumption. However, we also introduce an (implicit) spectral transformation regularization, which entangles the reweighting in a less straightforward manner (see line 931). Our main technical contribution lies in carefully organizing the placement of different weightings within the matrix structure to achieve an optimal convergence rate. Importantly, not all weighting strategies (in generalizing [5]'s proof) guarantee optimal performance. Additionally, our work provides theoretical insights and revisits several key aspects (e.g., the role of smooth inductive bias) in the context of physics-informed machine learning, as demonstrated in our experiments.
>
> ----
>
>
> We believe our new numerical experiments verify our findings and extend our findings beyond kernel methods, offering valuable insights and practical guidance to PINN users. We hope the reviewer can reevaluate our paper accordingly. If the reviewer has any further concerns, we would be happy to address them.
>
> ----
>
> [1] Bietti A, Bach F. Deep equals shallow for ReLU networks in kernel regimes. arXiv preprint arXiv:2009.14397, 2020.
>
> [2] Mallinar N, Simon J, Abedsoltan A, et al. Benign, tempered, or catastrophic: Toward a refined taxonomy of overfitting. Advances in Neural Information Processing Systems, 2022, 35: 1182-1195.
>
> [3] Ding L, Mak S, Wu C F. BdryGP: a new Gaussian process model for incorporating boundary information. arXiv preprint arXiv:1908.08868, 2019.
>
> [4] Lu Y, Chen H, Lu J, et al. Machine learning for elliptic PDEs: Fast rate generalization bound, neural scaling law and minimax optimality[J]. arXiv preprint arXiv:2110.06897, 2021.
>
> [5] Barzilai D, Shamir O. Generalization in kernel regression under realistic assumptions. arXiv preprint arXiv:2312.15995, 2023.
>
> [6] Schölpple M, Steinwart I. Which Spaces can be Embedded in Reproducing Kernel Hilbert Spaces?. arXiv preprint arXiv:2312.14711, 2023.
>
> [7] Bartek T Knapik, Aad W Van Der Vaart, and J Harry van Zanten. Bayesian inverse problems with gaussian priors. 2011.
>
> [8] Vivien Cabannes, Loucas Pillaud-Vivien, Francis Bach, and Alessandro Rudi. Overcoming the curse of dimensionality with laplacian regularization in semi-supervised learning. In Thirty-Fifth Conference on Neural Information Processing Systems, 2021.
>
> [9] Maarten V de Hoop, Nikola B Kovachki, Nicholas H Nelsen, and Andrew M Stuart. Convergence rates for learning linear operators from noisy data. arXiv preprint arXiv:2108.12515, 2021.
>
> [10] Yiping Lu, Jose Blanchet, and Lexing Ying. Sobolev acceleration and statistical optimality for learning elliptic equations via gradient descent. Advances in Neural Information Processing Systems, 35:33233–33247, 2022.
>
> [11] Wu J, Zou D, Braverman V, Gu Q and Kakade S. The power and limitation of pretraining-finetuning for linear regression under covariate shift. Advances in Neural Information Processing Systems, 2022, 35: 33041-33053.

---

> > ### Author Response · Authors · 2024-11-25
> >
> > As the discussion deadline is tomorrow, we have already provided **empirical validation** on our theory, **explained the co-diagonalization assumption**, and also addressed other concerns related to the proof. we will be more than happy to provide additional clarifications if you have any further specific technical questions that still concern you. We hope our response has addressed your concerns and we would be grateful if you would consider reevaluating your assessment of our work.

---

> > ### Comment · Reviewer_eSS5 · 2024-11-26
> >
> > Thank you for your explanation. However, I am still concerned about the strong assumptions made in this paper, particularly the co-diagonalization assumption. If the data is the uniform distribution over the sphere, the eigenbasis can be explicitly given by the spherical harmonics, which, I think, is a much simplified case.

---

> ### Author Response · Authors · 2024-11-28
>
> We sincerely thank reviewers eSS5 for responsing to improve our manuscript better, here's our improvement to the paper.
>
> > Use of the results beyond kernel methods
>
> We agree with the reviewer that our theory results center around kernel methods. However, our aim is to leverage kernel theory to provide insights into the behavior of overparameterized neural networks, a domain where kernel methods are widely used as a theoretical model [1,2]. In response to your suggestions, we have revised our manuscript to emphasize that our estimators are kernel-based. Furthermore, we highlight that the intuition behind our theoretical results on kernel estimators can also be applied to Physics-Informed Neural Networks (PINNs), where kernels serve as a theoretical framework for modeling and understanding the behavior of overparameterized neural networks.
>
>
> > Regards the hardness of moving the co-diagonalization assumption
>
> **To our knowledge, this remains an open question of how to avoid such assumptions. **So using such assumptions is widely used in the literature, including references [1, 2, 3] for inverse problems, [4] for operator learning, and [5] for pre-training and fine-tuning. We added a numerical experiment to see if our insights still hold beyond the assumption. See our answer in regard to numerical experiments. We believe it would be overly stringent to demand the removal of an assumption that is commonly adopted in most papers within the literature. To validate our understanding and findings, we present experiments in the appendix that demonstrate benign overfitting behavior beyond the co-diagonalization assumption. We believe our understanding and findings can be a contribution for the physic-informed machine learning community to understand the behavior of PINN better.
>
> > Regards Kernel examples that takes Fourier modes as eigenvectors
>
> Thanks for the reviewer for brining this to our attention. We have updated the remark on co-diagonalization assumption at Line 254 in the revised manuscript. The examples we highlighted in blue in the paper demonstrate kernels that use Fourier modes as eigenvectors. These examples include inner product kernels designed for uniform data. To be specfic, the Laplacian operator's operator are Fourier modes  \( $e^{i \omega x}$ \) (for $\Delta e^{i \omega x} = -\|\omega\|^2 e^{i \omega^T x}$).
>
> For a shift-invariant kernel \( K(x, y) = K(x, y) = k(x - y) \), by Bochner's theorem, $k(x-y)$ can also be expressed in terms of its Fourier modes  $k(x-y) = \frac{1}{(2\pi)^d} \int_{\mathbb{R}^d}{p(w) e^{i w^T (x-y)}} dw$ where $p(w)$ is the probability distribution over $w$. Therefore:
>    $$
>     \Sigma(f)(x) =  \frac{1}{(2\pi)^d} \int_{\mathbb{R}^d}  k(x-y) f(y) dy = \int_{R^d} \int_{\mathbb{R}^d}{p(w) e^{i w^T (x-y)}} f(y) \ dw \ dy
>    $$
>
> For $f(x)= e^{i \omega^T x}$,  we have $\int_{\mathbb{R}^d} \int_{\mathbb{R}^d}{p(w) e^{i w^T (x-y)}} f(y) \ dw \ dy = \frac{1}{(2\pi)^d} \int_{R^d} p(w) e^{i w^T x} \ dw = \frac{1}{(2\pi)^d} \mathbb{E}_{\omega} e^{i w^Tx}$. **Therefore, the covariance operator $\Sigma$ has the fourier modes $e^{i \omega^T x}$** means that it is co-diagonal with the Laplacian operators.
>
> To valid our finding for the empirical user,  **we have conducted additional experiments on PDE, where the co-diagonalization assumption no longer holds in the appendix. ** We plot the noise profile of solving nonlinear-PDE $-\nabla \cdot (|x| \nabla u) = f \quad \text{for } x \in \Omega \text{ and } u = 0 $. The noise profile of Physics-informed interpolator exhibits benign overfitting, unlike the regression interpolator, which verifies our theoritcal findings.
>
> -----
>
> We hope the reviewer understand our using of kenrel method as a therotical model to understand the behavior of neural network nd our new numerical experiments verify our findings and extend our findings beyond kernel methods, offering valuable insights and practical guidance to PINN users. We hope the reviewer can reevaluate our paper accordingly. If the reviewer has any further concerns, we would be happy to address them.
>
> -----
>
> [1] Shamir O. The implicit bias of benign overfitting Conference on Learning Theory. PMLR, 2022: 448-478.
>
> [2] Belkin M. Fit without fear: remarkable mathematical phenomena of deep learning through the prism of interpolation. Acta Numerica, 2021, 30: 203-248.

---

> ### Comment · Reviewer_fsKN · 2024-12-01
>
> We thank the authors sincerely for the detailed and thorough responses to our questions.
>
> With regard to the terminology “benigning overfitting”, we understand that the model (2) (minimum norm interpolation) does indeed provide an overparametrized interpolation estimator, and this paper shows that this interpolation estimator can surprisingly generalize. On the contrary, like kernel ridge regression, model (1) (regularized least square) uses a regularization penalty term in the loss function, hence does not in general interpolate the sample data perfectly, even though the minimizer of the penalized loss function of (1) can be actually computed explicitly (Lemma 3.1). This is why the terminology “benigning overfitting” is a little bit confusing for us.
>
> As for the relationship between model (1) and kernel ridge regression, the main results (Theorem 4.1 and 4.2) provides us with a guidance on how to choose a sufficiently great inductive bias to reach the optimal convergence rate, and reminds us of the sactuation effect in kernel ridge regression. First, the RKSS itself is an RKHS with a new kernel sharing the same eigenspaces with the original kernel, hence the loss function of (1) would coincide with the loss function of kernel ridge regression of the new kernel, if we pretended that the operator A were an identity map (which violates Assumption 2.2(d) of course), and the inductive bias characterizes the smoothness of this new kernel; second, the sactuation effect (see [1][2] for example) states that kernel ridge regression estimator cannot reach the optimal convergence rate if the relative smoothness of kernel function is too low compared with the target function f*. Thus, we are very curious about whether there is some deep relationship between these two settings of problems, and how to explain the role that the operator A plays.
>
> [1]F. Bauer, S. Pereverzyev, and L. Rosasco. On regularization algorithms in learning theory. Journal of complexity, 23(1):52–72, 2007.
> [2]Li, Y., Zhang, H., & Lin, Q. On the saturation effect of kernel ridge regression. arXiv preprint arXiv:2405.09362, 2024.

---

> ### Author Response · Authors · 2024-12-03
>
> We thank the reviewer for their review and comments and we value the opportunity to discuss and remain available to address any additional questions or concerns. In light of the clarifications, revisions, and additional experiments we have provided, we kindly encourage the reviewer to revisit their assessment and consider adjusting their final rating before the discussion period ends.

---

> ### Author Response · Authors · 2024-12-04
>
> We thank reviewer fsKN for the detailed comments!
>
> > Confusion on Benign overfitting
>
> Thanks for raising this point! Yes as you mentioned, **benign overfitting** in our paper mainly refers to the minimum norm interpolator, where we show that it surprisingly exhibits benign overfitting (because of the inverse problem operator $\mathcal{A}$). We will update our manuscript to emphasize that benign overfitting is for the interpolator only, to avoid misunderstanding.
>
> Besides, we remark that our techniques based on bias-variance decomposition hold for both regularized least squares and interpolator **under the same proof framework**. This differs from previous work requiring separate theoretical analysis techniques (regularized [1, 2], interpolator [3, 4]). Furthermore, our results on the regularized estimator can recover the minimax optimal rate in [5].
>
> > Relationship between model(1) and kernel ridge regression
>
> Thanks for raising this question!  In our opinion, taking $\mathcal{A} = I$ still satisfies Assumption 2.2(d) on co-diagonalization. We require such an assumption to ensure commutativity of $\mathcal{A}$ and $\Sigma$ which is required in our proofs, when $\mathcal{A} = I$, we have $\mathcal{A} \Sigma = I \Sigma = \Sigma I = \Sigma \mathcal{A}$.
>
> In this special case, you are right that model (1) concides with the kernel ridge regression, with penalized Sobolev norm that regulates the inductive bias on smoothness of the model. However, the key message of our paper is the role operator $\mathcal{A}$ play in our bound, which we explain in the next question.
>
>
> > Regarding Saturation effect of kernel and how this affects model (1)
>
> Thanks for mentioning this interesting phenomenon! In this case, we think similar saturation effect can be observed.
> However, we suggest that $\mathcal{A}$ stabilizes the variance, and makes the model match with the task intuitively. Hence, $\mathcal{A}$ may help to alleviate the saturation effect, which is worth further investigation. We will add a section to discuss the saturation effect in this setup in the camera-ready version if the paper gets accepted.
>
> We thank the reviewer again for helping to improve our work and we hope our response has addressed the concerns. It would be greatly appreciated if the reviewer is willing to reassess our work and modify the score accordingly.
>
> ---
>
>
> [1] Andrea Caponnetto and Ernesto De Vito. Optimal rates for the regularized least-squares algorithm. Foundations of Computational Mathematics, 7:331–368, 2007.
>
> [2] Ingo Steinwart, Don R Hush, Clint Scovel, et al. Optimal rates for regularized least squares regression. In COLT, pages 79–93, 2009.
>
> [3] Peter L Bartlett, Philip M Long, G´abor Lugosi, and Alexander Tsigler. Benign overfitting in linear regression. Proceedings of the National Academy of Sciences, 117(48):30063–30070, 2020.
>
> [4] Tengyuan Liang and Alexander Rakhlin. Just interpolate: Kernel “ridgeless” regression can generalize. 2020.
>
> [5] Yiping Lu, Jose Blanchet, and Lexing Ying. Sobolev acceleration and statistical optimality for learning elliptic equations via gradient descent. Advances in Neural Information Processing Systems, 35:33233–33247, 2022.

---

### Official Review · Reviewer_cUNr · 2024-11-09

**Soundness:** 3
**Presentation:** 3
**Contribution:** 3
**Rating:** 5
**Confidence:** 3

**Summary:**

The paper introduces an approach for the solution of inverse problems, where the forward problem is assumed to be given by a linear, self adjoint elliptic operator,  via a kernel-based fitting (least-squares of interpolation) regularized with a kernel Sobolev $\beta$ norm.
The focus is particularly on the role of the parameter $\beta$ in terms of the existence of a non-asymptotic
benign overfitting regime.

**Strengths:**

- The topic is of current interest, and the results are clearly presented, discussed, and put into perspective in the existing literature.
- The new results are interesting and informative w.r.t. the role of the various parameters.

**Weaknesses:**

- The results are presented and discussed in terms of general machine learning techniques (see e.g. the mention of PINN in "Takeaway to Practitioners" in Section 4.3), but it's not directly clear how the result of the paper can be applied beyond kernel-based methods.

- The results rely on a number of assumptions (Assumption 2.2) that are a bit hard to transfer to practical cases. For example, Assumption 2.2, point (d): The co-diagonalization assumption is quite restrictive, and does not generally hold if A is PDE operator and K is chosen freely.

- Similarly, Assumption 3.4 is hard to relate to actual kernels. In particular, although the examples in Remark 3 are correct, $\beta_k$ in Assumption 3.4 is usually $\infty$, as the first sum in the maximum can be diverging, see e.g. [2]. This is the case with many commonly used kernels. Especially, I believe the assumption $\beta_k=\Theta(1)$ is generally too restrictive.

[2] Ding-Xuan Zhou, The Covering Number in Learning Theory, Journal of Complexity (2002)

**Questions:**

Regarding the three points discussed above, I would suggest the following:
-  Discuss how the results could potentially be extended or applied to non-kernel methods.
-  Provide practical caes of PDEs and kernels that fit into Assumptions 2.2, or otherwise discuss how the results might be extended if the co-diagonalization assumption is relaxed.
- Provide practical examples of kernels (beyond Remark 3) that fit into Assumption 3.4, or discuss how the results might be extended if these hypotheses are relaxed.

Apart from the points discussed above, there are the following minor points:

- Example 2.3: I believe Bohner should be Bochner.
- in "Decomposition of signals", end of page 5: Is the map $\phi_{>k}^*$ well defined for any $\theta\in\mathbb{R}^\infty$?
- Assumption 3.5 is cited in Theorem 3.2, before being introduced.

---

> ### Author Response · Authors · 2024-11-24
>
> We sincerely thank reviewer cUNr for their efforts and their acknowledgement of the strengths of our paper. We address the reviewer's questions point by point as follows:
>
> > Regards Empirical Validation
>
> We added two numerical experiments to verify the two main discoveries in our paper. We run a PINN to solve the Poisson equation $u = \Delta f$ with zero Dirichlet  boundary condition. We trained neural network with Relu, $\text{Relu}^2$, $\text{Relu}^3$, $\text{Relu}^4$activation to test how the inductive bias of smoothness changes the convergence. As a smoother activation function is used,  we expect the resulting NTK to have faster decay [1]. Our results show that once the activation function is smooth enough, increasing smoothness would no longer help matches our theoretical claim. Secondly, we plotted the noise profile (how the asymptotic risk varies with the variance of additive Gaussian noise) following [2] to examine the beginning overfitting effect.  We show that the standard regression interpolator performs worse under stronger noise level. Instead, the test risk of the Physics-Informed interpolator does not change too much at various noise levels.
>
> Our intuition behind the benign overfitting behavior for inverse problems differs from that of plain regression because we are predicting \( $\Delta^{-1} u$ \), not \( $u$ \) itself. The operator \( $\Delta^{-1}$ \) acts like a kernel smoothing operation (with the Green's function as the kernel), which smooths the predictions and underweights errors in the high-frequency components—these being the main contributors to the overall error. For general PDEs governing physical laws, most behave like differential operators, where the forward problem amplifies high-frequency components. Consequently, solving the inverse problem tends to attenuate these high-frequency components, resulting in a similar noise stabilization effect. This is what we called the “variance stabilisation” effect in our main text. We refined our main text to make the explanation clearer.
>
> We have already updated our submission, see Section 5 for details.
>
>
> > Regards how to apply our results beyond kernel methods
>
>
> The used techniques for non-kernel methods largely differ from our results as the spectrum decomposition cannot be directly used. However, our results still share some similar spirit on kernel/non-kernel models. For example, source condition means the smoothness of the target function. Accordingly, our results can provide practical guidance to non-kernel methods. Thus we added two numerical experiments to verify the two main discoveries in our paper beyond non-kernel methods. We show that the impact of smooth inductive bias with the benign overfitting phenomenon still holds for non-kernel methods (neural networks). See our answer in regards to numerical experiments.
>
>
> > Regards the co-diagonalization assumption
>
>
> We admit that the co-diagonal assumption is strong. However, when data is uniform distribution over the sphere (which is a common case in PINN learning since the data is often uniformly sampled from the domain), the co-diagonal assumption holds for many differential operators, e.g. the laplace operator $\Delta$.
>
>  **To our best knowledge, this remains an open question of how to avoid such assumptions.** So using such assumptions is widely used in the literature, including references [3, 4, 6] for inverse problems, [5] for operator learning, and [7] for pre-training and fine-tuning.  We added a numerical experiment to see if our insights still hold beyond the assumption. See our answer in regards to numerical experiments.

---

> ### Author Response · Authors · 2024-11-24
>
> > Practical Examples of PDE that satisfies Assumption 2.2
>
> This can refer to Example 2.3 Schrodinger equation on a Hypercube, where the co-diagonalization assumption holds since both the Laplacian operator ∆ and the Kernel covariance operator have the Fourier modes as eigenfunction which is guaranteed by Bochner’s theorem. The famous Poisson equation $u = \Delta f$ on $\Omega$ and with Dirichlet condition on $\partial \Omega$ also satisfies the co-diagonalization assumption.
>
>
> > Regards Assumption 3.4
>
>
> Verifying assumption 3.4 needs to write down the mercer decomposition and use the explicit form of the feature map (check the boundedness of eigenfunctions, which is an active area in pure math [8]). However Dot-Product Kernels on sphere and  RBF/shift-invariant kernels satisfies our assumption which is shown both in our paper line 326-337 and [9, Appendix H] . When $\beta=O(1)$ for trace-class kernel, Assumption 3.5 would automatically satisfy.  Assumptions 3.4 and 3.5 are the weakest assumptions made in the benign overfitting literature as far as the authors know. If there are any other assumptions that could lead to potential improvements, we would be happy to make the necessary modifications.
>
> > Well-definedness of $\phi^*_{>k}$ at the end of Page 5
>
> Thanks for pointing this out,  we've changed the definition of $\phi^*_{>k}$ from $R^{\infty}\rightarrow \mathcal{H}$ to $\ell_2^{\infty}\rightarrow \mathcal{H}$. Sorry for the confusion.
>
> > Assumption 3.5 is cited in Theorem 3.2
>
> We’ve changed our order of section 3.2 concentration coefficients and section 3.1 excess risk. Sorry for the confusion.
>
>
> ------
>
>
> We believe our new numerical experiments verifies our findings and extend our findings beyond kernel methods, offering valuable insights and practical guidance to PINN users. We hope reviewer can reevaluate our paper accordingly. If the reviewer has any further concerns, we would be happy to address them.
>
> --------
> [1] Bietti A, Bach F. Deep equals shallow for ReLU networks in kernel regimes. arXiv preprint arXiv:2009.14397, 2020.
>
> [2] Mallinar N, Simon J, Abedsoltan A, et al. Benign, tempered, or catastrophic: Toward a refined taxonomy of overfitting. Advances in Neural Information Processing Systems, 2022, 35: 1182-1195.
>
> [3] Bartek T Knapik, Aad W Van Der Vaart, and J Harry van Zanten. Bayesian inverse problems with gaussian priors. 2011.
>
> [4] Vivien Cabannes, Loucas Pillaud-Vivien, Francis Bach, and Alessandro Rudi. Overcoming the curse of dimensionality with laplacian regularization in semi-supervised learning. In Thirty-Fifth Conference on Neural Information Processing Systems, 2021.
>
> [5] Maarten V de Hoop, Nikola B Kovachki, Nicholas H Nelsen, and Andrew M Stuart. Convergence rates for learning linear operators from noisy data. arXiv preprint arXiv:2108.12515, 2021.
>
> [6] Yiping Lu, Jose Blanchet, and Lexing Ying. Sobolev acceleration and statistical optimality for learning elliptic equations via gradient descent. Advances in Neural Information Processing Systems, 35:33233–33247, 2022.
>
> [7] Wu J, Zou D, Braverman V, Gu Q and Kakade S. The power and limitation of pretraining-finetuning for linear regression under covariate shift. Advances in Neural Information Processing Systems, 2022, 35: 33041-33053.
>
> [8] Steinerberger S. Quantum entanglement and the growth of Laplacian eigenfunctions. Communications in Partial Differential Equations, 2023, 48(4): 511-541.
>
> [9] Barzilai D, Shamir O. Generalization in kernel regression under realistic assumptions. arXiv preprint arXiv:2312.15995, 2023.

---

> > ### Author Response · Authors · 2024-11-25
> >
> > As the discussion deadline is tomorrow, we have already provided **empirical validation** on our theory, showing how our results generalize beyond kernel setting, and **explained the co-diagonalization assumption**, we will be more than happy to provide additional clarifications if you have any further specific technical questions that still concern you. We hope our response has addressed your concerns and we would be grateful if you would consider reevaluating your assessment of our work.

---

> > > ### Comment · Reviewer_cUNr · 2024-11-26
> > > **Response to the authors**
> > >
> > > I would like to thank the authors for responding to my questions and for their efforts at clarification. However, my main questions remain at least partially unanswered, namely
> > > - *Use of the results beyond kernel methods:* I appreciate the empirical experiment, but it does not address my main concern. Namely, I think that the theoretical results of the paper only apply to kernels (at the current state of the paper). This is not a problem at all, but it should be made clearer, and any reference to PINNs or other physics-informed methods should be carefully explained. Adding numerical experiments on PINNs adds to the confusion, in my opinion.
> > > - *Co-diagonalization assumption:* I agree that the assumption may hold for the Laplacian, but only for some specific kernels. What are these kernels? In general, my problem here is that there are no guidelines on how to choose the kernel so that the co-diagonalization assumption holds for a given problem. Furthermore, the authors state in their reply: "We admit that the co-diagonalization assumption is strong". However, I don't see this fact reflected anywhere in the (revised) pdf.
> > > - *Kernel examples:* I appreciate the additional detail on this issue. However, it is still unclear how these hypotheses interact with the co-diagonalization hypothesis. Namely: What are some relevant pairs $(\mathcal{A}, K)$ of operator-kernel that fit the framework?

---

> ### Author Response · Authors · 2024-11-28
>
> We sincerely thank reviewers cUNr for responsing to improve our manuscript better, here's our improvement to the paper.
>
> > Use of the results beyond kernel methods
>
> We agree with the reviewer that our theory results center around kernel methods. However, our aim is to leverage kernel theory to provide insights into the behavior of overparameterized neural networks, a domain where kernel methods are widely used as a theoretical model [1,2]. In response to your suggestions, we have revised our manuscript to emphasize that our estimators are kernel-based. Furthermore, we highlight that the intuition behind our theoretical results on kernel estimators can also be applied to Physics-Informed Neural Networks (PINNs), where kernels serve as a theoretical framework for modeling and understanding the behavior of overparameterized neural networks.
>
>
>
>
> > Regards Kernel examples that takes Fourier modes as eigen vectors
>
> Thanks for the reviewer for brining this to our attention. We have updated the remark on co-diagonalization assumption at Line 254 in the revised manuscript. The examples we highlighted in blue in the paper demonstrate kernels that use Fourier modes as eigenvectors. These examples include inner product kernels designed for uniform data. To be specfic, the Laplacian operator's operator are Fourier modes  \( $e^{i \omega x}$ \) (for $\Delta e^{i \omega x} = -\|\omega\|^2 e^{i \omega^T x}$).
>
> For a shift-invariant kernel \( K(x, y) = K(x, y) = k(x - y) \), by Bochner's theorem, $k(x-y)$ can also be expressed in terms of its Fourier modes  $k(x-y) = \frac{1}{(2\pi)^d} \int_{\mathbb{R}^d}{p(w) e^{i w^T (x-y)}} dw$ where $p(w)$ is the probability distribution over $w$. Therefore:
>    $$
>     \Sigma(f)(x) =  \frac{1}{(2\pi)^d} \int_{\mathbb{R}^d}  k(x-y) f(y) dy = \int_{R^d} \int_{\mathbb{R}^d}{p(w) e^{i w^T (x-y)}} f(y) \ dw \ dy
>    $$
>
> For $f(x)= e^{i \omega^T x}$,  we have $\int_{\mathbb{R}^d} \int_{\mathbb{R}^d}{p(w) e^{i w^T (x-y)}} f(y) \ dw \ dy = \frac{1}{(2\pi)^d} \int_{R^d} p(w) e^{i w^T x} \ dw = \frac{1}{(2\pi)^d} \mathbb{E}_{\omega} e^{i w^Tx}$. **Therefore, the covariance operator $\Sigma$ has the fourier modes $e^{i \omega^T x}$** means that it is co-diagonal with the Laplacian operators.
>
> To valid our finding for the empirical user,  **we have conducted additional experiments on PDE, where the co-diagonalization assumption no longer holds in the appendix. ** We plot the noise profile of solving nonlinear-PDE $-\nabla \cdot (|x| \nabla u) = f \quad \text{for } x \in \Omega \text{ and } u = 0 $. The noise profile of Physics-informed interpolator exhibits benign overfitting, unlike the regression interpolator, which verifies our theoritcal findings.
>
> -----
>
> We hope the reviewer understand our using of kenrel method as a therotical model to understand the behavior of neural network nd our new numerical experiments verify our findings and extend our findings beyond kernel methods, offering valuable insights and practical guidance to PINN users. We hope the reviewer can reevaluate our paper accordingly. If the reviewer has any further concerns, we would be happy to address them.
>
> -----
>
> [1] Shamir O. The implicit bias of benign overfitting Conference on Learning Theory. PMLR, 2022: 448-478.
>
> [2] Belkin M. Fit without fear: remarkable mathematical phenomena of deep learning through the prism of interpolation. Acta Numerica, 2021, 30: 203-248.

---

> ### Author Response · Authors · 2024-12-03
>
> We thank the reviewer for their review and comments and we value the opportunity to discuss and remain available to address any additional questions or concerns. In light of the clarifications, revisions, and additional experiments we have provided, we kindly encourage the reviewer to revisit their assessment and consider adjusting their final rating before the discussion period ends.

---

### Meta-Review · Area_Chair_AQUC · 2024-12-16

**Metareview:**

The paper studies the statistical performance of kernel-based methods for linear elliptic PDE inverse problems. It provides learning curves in the kernel Sobolev space (KSS) norm for the regularized least squares and minimum norm interpolation estimators. Using a bias-variance decomposition, it shows that elliptic operators can exhibit benign overfitting under certain conditions. The results also demonstrate that regularized least squares estimators achieve an optimal convergence rate, independent of the inductive bias, if the bias is sufficiently smooth.

Overall, the paper is well-written and presents an interesting set of ideas. However, most reviewers noted that the results rely on assumptions that could be considered overly restrictive. While these assumptions are used in some prior works, they raise concerns about the generality and broader impact of the results beyond the specific setting considered. The paper could be strengthened by either relaxing some of these assumptions or conducting extensive experimental studies to illustrate their validity and limitations in real-world settings.

**Additional Comments On Reviewer Discussion:**

Most reviewers found the assumptions underlying the results to be restrictive. While the authors argued that these assumptions have been employed in prior works, this does not fully address concerns about the generality and broader applicability of the results beyond the specific setting considered.

---

### Decision · Program_Chairs · 2025-01-22

Reject